# SUBSAMPLED ENSEMBLE CAN IMPROVE GENERALIZATION TAIL EXPONENTIALLY

## ABSTRACT

Ensemble learning is a popular technique to improve the accuracy of machine learning models. It hinges on the rationale that aggregating multiple weak models can lead to better models with lower variance and hence higher stability, especially for discontinuous base learners. In this paper, we provide a new perspective on ensembling. By selecting the best model trained on subsamples via majority voting, we can attain exponentially decaying tails for the excess risk, even if the base learner suffers from slow (i.e., polynomial) decay rates. This tail enhancement power of ensembling is agnostic to the underlying base learner and is stronger than variance reduction in the sense of exhibiting rate improvement. We demonstrate how our ensemble methods can substantially improve out-of-sample performances in a range of examples involving heavy-tailed data or intrinsically slow rates.

## 1 INTRODUCTION

Ensemble learning (Dietterich, 2000; Zhou, 2012) is a class of methods to improve the accuracy of machine learning models. It comprises repeated training of models (the "base learners"), which are then aggregated through averaging or majority vote. In the literature, the main justification for ensemble methods, such as bootstrap aggregating (bagging) (Breiman, 1996) and boosting (Freund et al., 1996), pertains to bias/variance reduction or higher stability. This justification has been shown to be particularly relevant for certain U-statistics (Buja & Stuetzle, 2006) and models with hard-thresholding rules such as decision trees (Breiman, 2001; Drucker & Cortes, 1995).

Contrary to the established understanding, in this paper we present a new view of ensembling in offering an arguably stronger power than variance reduction: By suitably selecting the best base learners trained on random subsamples, ensembling leads to exponentially decaying excess risk tails. In particular, for general stochastic optimization problems that suffer from a slow, namely polynomial, decay in excess risk tails, ensembling can reduce these tails to an exponential decay rate. Thus, instead of the typical constant factor of improvement exhibited by variance reduction, our ensemble method offers a rate improvement, and moreover, the improvement is substantial.

In the following, we will first qualify our claims above by discussing how slow convergence can arise generically in machine learning and more general data-driven decision-making problems under heavy-tailed data. We then give intuition on our new ensembling perspective, proposed procedures, and the technicality involved in a full analysis.

**Main results at a high level.** We begin by introducing a generic stochastic optimization problem

$$\min_{\theta \in \Theta} L(\theta) := \mathbb{E}\left[l(\theta, z)\right], \tag{1}$$

where $\theta$ is the decision variable on space $\Theta$, $z \in \mathcal{Z}$ denotes the randomness governed by a probability distribution, and $l$ is the cost function. $n$ i.i.d. samples $\{z_1, \ldots, z_n\}$ are available from the underlying distribution of $z$. In machine learning, $\theta$ corresponds to model parameters, $\{z_1, \ldots, z_n\}$ the training data, $l$ the loss function, and $L$ the population-level expected loss. More generally, (1) encapsulates data-driven decision-making problems, namely the integration of data on $z$ into a downstream optimization task with overall cost function $l$ and prescriptive decision $\theta$. These problems are increasingly prevalent in various industrial applications (Kamble et al., 2020; Bertsimas et al., 2023; Ghosal et al., 2024), such as in supply chain network design where $\theta$ may represent the decision to

open processing facilities, $z$ the uncertain supply and demand, and $l$ the total cost of processing and transportation.

Given the data, we can train the model or decision with a learning algorithm that maps the data to an element in $\Theta$. This encompasses a wide range of methods, including machine learning training algorithms and data-driven approaches like sample average approximation (SAA) (Shapiro et al., 2021) and distributionally robust optimization (DRO) (Mohajerin Esfahani & Kuhn, 2018) in stochastic optimization. Our proposal and theory described below are agnostic to the choice of learning algorithm.

We characterize the generalization performance of a solution to (1), denoted by $\hat{\theta}$, via the tail probability bound on the excess risk or regret $L(\hat{\theta}) - \min_{\theta \in \Theta} L(\theta)$, i.e., $\mathbb{P}(L(\hat{\theta}) > \min_{\theta \in \Theta} L(\theta) + \delta)$ for some fixed $\delta > 0$, where the probability is over both the data and training randomness. By a polynomially decaying generalization tail, we mean that

$$\mathbb{P}\Big( L(\hat{\theta}) > \min_{\theta \in \Theta} L(\theta) + \delta \Big) \leq C_1 n^{-\alpha} \tag{2}$$

for some $\alpha > 0$ and $C_1$ depends on $\delta$. Such bounds are common under heavy-tailed data distributions (Kaňková & Houda, 2015; Jiang et al., 2020; Jiang & Li, 2021) due to slow concentration, which frequently arises in machine learning applications such as large language models (e.g, Jalalzai et al. (2020); Zhang et al. (2020); Cutkosky & Mehta (2021) among others), finance (Mainik et al., 2015; Gilli & Këllezi, 2006) and physics (Fortin & Clusel, 2015; Michel & Chave, 2007), and are proved to be tight (Catoni, 2012) for empirical risk minimization (ERM) (Vapnik, 1991). As our key insight, our proposed ensembling methodology can improve the above to an exponential decay, i.e.,

$$\mathbb{P}\Big( L(\hat{\theta}) > \min_{\theta \in \Theta} L(\theta) + \delta \Big) \leq C_2 \gamma^{n/k}, \tag{3}$$

where $k$ is the subsampled data size and can be chosen at a slower rate in $n$, and $\gamma < 1$ depends on $k, \delta$ such that $\gamma \to 0$ as $k \to \infty$. Hence, when $k$ is properly chosen, the decay becomes exponential. This exponential acceleration is qualitatively different from the well-known variance reduction benefit of ensembling in several aspects. First, variance reduction refers to the smaller variability in predictions from models trained on independent data sets, which has a more direct impact on the expected regret than the tail decay rate. Second, the improvement by variance reduction is typical of a constant factor (e.g., Bühlmann & Yu (2002) reported a reduction factor of 3), thus affecting at best the constant $C_1$ in (2), whereas we obtain an order-of-magnitude improvement.

**Main intuition.** To facilitate our explanation, let us first focus on discrete space $\Theta$. Our ensembling methodology uses a majority-vote mechanism at the model level: After repeatedly running the learning algorithm on subsamples to generate many models, we output the model that occurs most frequently. This implicitly solves a surrogate optimization problem over the same decision space $\Theta$ as (1) that maximizes the probability of being output by the learning algorithm. This conversion of the original general objective in (1) to a probability objective is the key: As an expectation of a random indicator function, the latter is uniformly bounded even if the original objective is heavy-tailed. Together with a bootstrap argument that establishes the closeness between subsample and full data, this in turn results in exponentially decaying tails for the regret.

For more general problems with continuous space, we replace the simple majority vote with a vote based on the likelihood of being $\epsilon$-optimal among all the generated models when evaluated on a random subsample. This avoids the degeneracy issue of using a simple majority vote for continuous problems while retaining similar (in fact, even stronger as we will see) guarantees. Regardless of discrete or continuous model space, our main insight on turning (2) into (3) applies. Moreover, in the discrete case, it turns out that not only the tail bound but also the average-case regret improves exponentially. This also explains why our improvement is particularly significant for discrete-decision problems in the experiments.

The rest of the paper is organized as follows. Section 2 presents our ensemble methods and their finite-sample bounds. Section 3 presents experimental results, and Section 4 discusses related work. Section 5 discusses limitations and concludes the paper. A review of additional related work, technical proofs, and additional experimental results can be found in the appendix.

## 2 METHODOLOGY AND THEORETICAL GUARANTEES

To solve (1) using data, we consider the generic learning algorithm in the form of a mapping

$$\mathcal{A}(z_1, \ldots, z_n; \omega) : \mathcal{Z}^n \times \Omega \rightarrow \Theta$$

that takes in the training data $(z_1, \ldots, z_n)$ and outputs a model possibly under some algorithmic randomness $\omega$ that is independent of the data. Examples of $\omega$ include gradient sampling in stochastic first-order algorithms and feature/data subsampling in random forests. $\mathcal{A}(z_1, \ldots, z_n; \omega)$ serves as our base learner. For convenience, we omit $\omega$ to write $\mathcal{A}(z_1, \ldots, z_n)$ when no confusion arises.

### 2.1 A BASIC PROCEDURE

We first introduce a procedure called MoVE that applies to discrete solution or model space $\Theta$. MoVE, which is formally described in Algorithm 1, repeatedly draws a total of $B$ subsamples from the data without replacement, learns a model via $\mathcal{A}$ on each subsample, and finally conducts a majority vote to output the most frequently subsampled model. Tie-breaking can be done arbitrarily.

---

**Algorithm 1 M**ajority **V**ote **E**nsembling (MoVE)

1: **Input:** A base learning algorithm $\mathcal{A}$, $n$ i.i.d. observations $\mathbf{z}_{1:n} = (z_1, \ldots, z_n)$, subsample size $k < n$, and ensemble size $B$.
2: **for** $b = 1$ **to** $B$ **do**
3:     Randomly sample $\mathbf{z}_k^b = (z_1^b, \ldots, z_k^b)$ uniformly from $\mathbf{z}_{1:n}$ without replacement, and obtain $\hat{\theta}_k^b = \mathcal{A}(z_1^b, \ldots, z_k^b)$.
4: **end for**
5: **Output:** $\hat{\theta}_n \in \arg\max_{\theta \in \Theta} \sum_{b=1}^{B} \mathbb{1}(\theta = \hat{\theta}_k^b)$.

---

To understand MoVE, we consider an optimization associated with the base learner $\mathcal{A}$

$$\max_{\theta \in \Theta} \ p_k(\theta) := \mathbb{P}\left(\theta = \mathcal{A}(z_1, \ldots, z_k)\right), \tag{4}$$

which maximizes the probability of a model being output by the base learner on $k$ i.i.d. observations. Here the probability $\mathbb{P}$ is with respect to both the training data and the algorithmic randomness. If $B = \infty$, MoVE essentially maximizes an empirical approximation of (4), i.e.

$$\max_{\theta \in \Theta} \ \mathbb{P}_*\left(\theta = \mathcal{A}(z_1^*, \ldots, z_k^*)\right), \tag{5}$$

where $(z_1^*, \ldots, z_k^*)$ is a uniform random subsample from $(z_1, \ldots, z_n)$, and $\mathbb{P}_*$ denotes the probability with respect to the algorithmic randomness and the subsampling randomness conditioned on $(z_1, \ldots, z_n)$. With a finite $B < \infty$, extra Monte Carlo noises are introduced, leading to the following maximization problem

$$\max_{\theta \in \Theta} \ \frac{1}{B} \sum_{b=1}^{B} \mathbb{1}(\theta = \mathcal{A}(z_1^b, \ldots, z_k^b)), \tag{6}$$

which gives exactly the output of MoVE. In other words, MoVE is a *bootstrap approximation* to the solution of (4). The following result materializes the intuition explained in the introduction on the conversion of the original potentially heavy-tailed problem (1) into a probability maximization (6) that possesses exponential bounds:

**Theorem 1 (Finite-sample bound for Algorithm 1)** *Consider discrete decision space $\Theta$. Recall $p_k(\theta)$ defined in (4). Let $p_k^{\max} := \max_{\theta \in \Theta} p_k(\theta)$, $\mathcal{E}_{k,\delta} := \mathbb{P}(L(\mathcal{A}(z_1, \ldots, z_k)) > \min_{\theta \in \Theta} L(\theta) + \delta)$ be the excess risk tail of $\mathcal{A}$, and*

$$\eta_{k,\delta} := p_k^{\max} - \mathcal{E}_{k,\delta}. \tag{7}$$

*For every $k \leq n$ and $\delta \geq 0$ such that $\eta_{k,\delta} > 0$, the solution output by* MoVE *satisfies that*

$$\mathbb{P}\left( L(\hat{\theta}_n) > \min_{\theta \in \Theta} L(\theta) + \delta \right)$$

$$\leq |\Theta| \left[ \exp\left( -\frac{n}{2k} \cdot D_{\mathrm{KL}}\left( p_k^{\max} - \frac{3\eta_{k,\delta}}{4} \middle\| p_k^{\max} - \eta_{k,\delta} \right) \right) + 2 \exp\left( -\frac{n}{2k} \cdot D_{\mathrm{KL}}\left( p_k^{\max} - \frac{\eta_{k,\delta}}{4} \middle\| p_k^{\max} \right) \right) \right.$$

$$+ \exp\left( -\frac{B}{24} \cdot \frac{\eta_{k,\delta}^2}{\min\left( p_k^{\max}, 1 - p_k^{\max} \right) + 3\eta_{k,\delta}/4} \right)$$

$$\left. + \mathbb{1}\left( p_k^{\max} + \frac{\eta_{k,\delta}}{4} \leq 1 \right) \cdot \exp\left( -\frac{n}{2k} \cdot D_{\mathrm{KL}}\left( p_k^{\max} + \frac{\eta_{k,\delta}}{4} \middle\| p_k^{\max} \right) - \frac{B}{24} \cdot \frac{\eta_{k,\delta}^2}{1 - p_k^{\max} + \eta_{k,\delta}/4} \right) \right].$$

$$(8)$$

*In particular, if $\eta_{k,\delta} > 4/5$, (8) is further bounded by*

$$|\Theta| \left( 3 \min\left( e^{-2/5}, C_1 \max(1 - p_k^{\max}, \ \mathcal{E}_{k,\delta}) \right)^{\frac{n}{C_2 k}} + e^{-B/C_3} \right), \qquad (9)$$

*where $C_1, C_2, C_3 > 0$ are universal constants, $|\Theta|$ denotes the cardinality of $\Theta$, and $D_{\mathrm{KL}}(p\|q) := p \ln \frac{p}{q} + (1-p) \ln \frac{1-p}{1-q}$ is the Kullback–Leibler divergence between two Bernoulli distributions with means $p$ and $q$.*

Theorem 1 states that the excess risk tail of MoVE decays exponentially in the ratio $n/k$ and ensemble size $B$. The bound consists of three parts. The first part has two terms with the Kullback–Leibler (KL) divergences and arises from the bootstrap approximation of (4) with (5). The second part quantifies the Monte Carlo error in approximating (5) with a finite $B$. The third part comes from the interaction between the two sources of errors and is typically of higher order. The multiplier $|\Theta|$ in the bound is avoidable, e.g., via a space reduction as in our next algorithm.

The quantity $\eta_{k,\delta}$ plays two roles. First, it quantifies how suboptimality in the surrogate problem (4) propagates to the original problem (1) in that every $\eta_{k,\delta}$-optimal solution for (4) is $\delta$-optimal for (1). Second, $\eta_{k,\delta}$ is directly related to the excess risk tail $\mathcal{E}_{k,\delta}$ of the base learner, in addition to $p_k^{\max}$ that captures the concentration of the base learner on $\delta$-optimal solutions. Therefore, $\eta_{k,\delta}$ taking large values signals the situation where the base learner already generalizes well. In this case, (8) can be simplified to (9). The bound (9) suggests that our approach does not hurt the performance of an already high-performing base learner as its generalization power is inherited through the $\max(1 - p_k^{\max}, \ \mathcal{E}_{k,\delta})$ term in the bound. See Appendix B for a more detailed comparison.

The quantity $\eta_{k,\delta}$ also hints at how to choose the subsample size $k$. As long as $\eta_{k,\delta}$ is bounded away from 0, our bound decays exponentially fast. Therefore, $k$ can be chosen in such a way that the base learner outputs good models more often than bad ones in order for the exponential decay of our bound to take effect, but at the same time considerably smaller than $n$ to ensure the amount of acceleration. In the experiments, we choose $k = \max(10, n/200)$.

On the choice of $B$, note that the two KL divergences in the first part of the tail bound (8) are in general bounded below by $\mathcal{O}(\eta_{k,\delta}^2)$ and so is the $\eta_{k,\delta}^2/(\min(p_k^{\max}, 1 - p_k^{\max}) + 3\eta_{k,\delta}/4)$ in the second part as $\eta_{k,\delta}$ is no larger than 1. Therefore using an ensemble size of $B = \mathcal{O}(n/k)$ is sufficient to control the Monte Carlo error to a similar magnitude as the data error.

## 2.2 A MORE GENERAL PROCEDURE

We next present a more general procedure called ROVE that applies to continuous space where simple majority vote in Algorithm 1 can lead to degeneracy, i.e., all learned models appear exactly once in the pool. Moreover, this general procedure relaxes our dependence on $|\Theta|$ in the bound in Theorem 1.

ROVE, displayed in Algorithm 2, proceeds initially the same as MoVE in repeatedly subsampling data and training the model using $\mathcal{A}$. However, in the aggregation step, instead of using a simple majority vote, ROVE outputs, among all the trained models, the one that has the highest likelihood of being $\epsilon$-optimal. This $\epsilon$-optimality avoids the degeneracy of the majority vote and, moreover, since we have restricted our output to the collection of trained models, the corresponding likelihood

---

**Algorithm 2** **R**etrieval and $\epsilon$-**O**ptimality **V**ote **E**nsembling (ROVE / ROVEs)

---

**Input:** A base learning algorithm $\mathcal{A}$, $n$ i.i.d. observations $\mathbf{z}_{1:n} = (z_1, \ldots, z_n)$, subsample size $k_1, k_2 < n$ (if no split) or $n/2$ (if split), ensemble sizes $B_1$ and $B_2$.

**Phase I: Model Candidate Retrieval**
**for** $b = 1$ **to** $B_1$ **do**
  Randomly sample $\mathbf{z}_{k_1}^b = (z_1^b, \ldots, z_{k_1}^b)$ uniformly from $\mathbf{z}_{1:n}$ (if no split) or $\mathbf{z}_{1:\lfloor \frac{n}{2} \rfloor}$ (if split)
  without replacement, and obtain $\hat{\theta}_{k_1}^b = \mathcal{A}(z_1^b, \ldots, z_{k_1}^b)$.
**end for**
Let $\mathcal{S} := \{\hat{\theta}_{k_1}^b : b = 1, \ldots, B_1\}$ be the set of all retrieved models.

**Phase II: $\epsilon$-Optimality Vote**
Choose $\epsilon \geq 0$ using the data $\mathbf{z}_{1:n}$ (if no split) or $\mathbf{z}_{1:\lfloor \frac{n}{2} \rfloor}$ (if split).
**for** $b = 1$ **to** $B_2$ **do**
  Randomly sample $\mathbf{z}_{k_2}^b = (z_1^b, \ldots, z_{k_2}^b)$ uniformly from $\mathbf{z}_{1:n}$ (if no split) or $\mathbf{z}_{\lfloor \frac{n}{2} \rfloor + 1:n}$ (if split)
  without replacement, and calculate

$$\widehat{\Theta}_{k_2}^{\epsilon, b} := \left\{ \theta \in \mathcal{S} : \frac{1}{k_2} \sum_{i=1}^{k_2} l(\theta, z_i^b) \leq \min_{\theta' \in \mathcal{S}} \frac{1}{k_2} \sum_{i=1}^{k_2} l(\theta', z_i^b) + \epsilon \right\}.$$

**end for**
**Output:** $\hat{\theta}_n \in \arg\max_{\theta \in \mathcal{S}} \sum_{b=1}^{B_2} \mathbb{1}(\theta \in \widehat{\Theta}_{k_2}^{\epsilon, b})$.

---

maximization is readily doable by simple enumeration. In addition, it helps reduce competition for votes among the best models as each subsample can now vote for multiple candidates, ensuring a high vote count for each of the top models even when there are many of them. This makes ROVE more effective than MoVE in the case of multiple (near) optima as our experiments will show. We have the following theoretical guarantees for Algorithm 2:

**Theorem 2 (Finite-sample bound for Algorithm 2)** *Recall the tail $\mathcal{E}_{k,\delta}$ of the base excess risk from Theorem 1. Consider Algorithm 2 with data splitting, i.e., ROVEs. Let $T_k(\cdot) := \mathbb{P}(\sup_{\theta \in \Theta} |(1/k) \sum_{i=1}^{k} l(\theta, z_i) - L(\theta)| > \cdot)$ be the tail function of the maximum deviation of the empirical objective estimate. For every $\delta > 0$, if $\epsilon$ is chosen such that $\mathbb{P}(\epsilon \in [\underline{\epsilon}, \bar{\epsilon}]) = 1$ for some $0 < \underline{\epsilon} \leq \bar{\epsilon} < \delta$ and $T_{k_2}((\delta - \bar{\epsilon})/2) + T_{k_2}(\underline{\epsilon}/2) < 1/5$, then*

$$\mathbb{P}\left(L(\hat{\theta}_n) > \min_{\theta \in \Theta} L(\theta) + 2\delta\right) \leq B_1 \left[ 3 \min\left( e^{-2/5}, C_1 T_{k_2}\left( \frac{\min(\underline{\epsilon}, \delta - \bar{\epsilon})}{2} \right) \right)^{\frac{n}{2 C_2 k_2}} + e^{-B_2/C_3} \right] \tag{10}$$
$$+ \min\left( e^{-(1 - \mathcal{E}_{k_1, \delta})/C_4}, C_5 \mathcal{E}_{k_1, \delta} \right)^{\frac{n}{2 C_6 k_1}} + e^{-B_1(1 - \mathcal{E}_{k_1, \delta})/C_7},$$

*where $C_1, C_2, C_3$ are the same as those in Theorem 1, and $C_4, C_5, C_6, C_7$ are universal constants.*

*Consider Algorithm 2 without data splitting, i.e., ROVE, and discrete space $\Theta$. Assume $\lim_{k \to \infty} T_k(\delta) = 0$ for all $\delta > 0$. Then, for every fixed $\delta > 0$, we have $\lim_{n \to \infty} \mathbb{P}(L(\hat{\theta}_n) > \min_{\theta \in \Theta} L(\theta) + 2\delta) \to 0$, if $\limsup_{k \to \infty} \mathcal{E}_{k,\delta} < 1$, $\mathbb{P}(\epsilon > \delta/2) \to 0$, $k_1$ and $k_2 \to \infty$, $n/k_1$ and $n/k_2 \to \infty$, and $B_1, B_2 \to \infty$ as $n \to \infty$.*

Theorem 2 provides an exponential excess risk tail, regardless of discrete or continuous space. The first line in the bound (10) is inherited from the bound (9) for MoVE from majority to $\epsilon$-optimality vote. In particular, the multiplier $|\Theta|$ in (9) is now replaced by $B_1$, the number of retrieved models. The second line in (10) bounds the performance sacrifice due to the restriction to Phase I model candidates.

ROVE may be carried out with the data split between the two phases, in which case it's referred to as ROVEs. Data splitting makes the procedure theoretically more tractable by avoiding inter-dependency between the phases but sacrifices some statistical power from halving the data size. Empirically we find ROVE to be overall more effective.

The optimality threshold $\epsilon$ is allowed to be chosen in a data-driven way and the main goal guiding this choice is to be able to distinguish models of different qualities. In other words, $\epsilon$ should be chosen to

create enough variability in the likelihood of being $\epsilon$-optimal across models. In our experiments, we find it a good strategy to choose an $\epsilon$ that leads to a maximum likelihood around $1/2$.

Lastly, our main theoretical results, Theorems 1 and 2, are derived using several novel techniques. First, we develop a sharper concentration result for U-statistics with binary kernels, improving upon standard Bernstein-type inequalities (e.g., Arcones (1995); Peel et al. (2010)). This refinement ensures the correct order of the bound, particularly (9), which captures the convergence of both the bootstrap approximation and the base learner, offering insights into the robustness of our methods for fast-converging base learners. Second, we perform a sensitivity analysis on the regret for the original problem (1) relative to the surrogate optimization (4), translating the superior generalization in the surrogate problem into accelerated convergence for the original. Finally, to establish asymptotic consistency for Algorithm 2 without data splitting, we develop a uniform law of large numbers (LLN) for the class of events of being $\epsilon$-optimal, using direct analysis of the second moment of the maximum deviation. Uniform LLNs are particularly challenging here because, unlike fixed classes in standard settings, this class dynamically depends on subsample size $k_2$ as $n \to \infty$.

## 3 NUMERICAL EXPERIMENTS

In this section, we numerically test Algorithm 1 (MoVE), Algorithm 2 with (ROVEs) and without (ROVE) data splitting in training neural networks for regression problems and solving stochastic programs. Additional experimental results are provided in Appendix D due to space constraints. The code is available at: https://anonymous.4open.science/r/vote_ensemble.

To empirically determine well-performing configurations for general use, we performed a comprehensive hyperparameter profiling of our algorithms in Appendix D.3. Below, we summarize the recommended configurations used in all experiments presented in this section (except Figure 4): 1) For discrete space $\Theta$, use $k = \max(10, n/200), B = 200$ for MoVE, and $k_1 = k_2 = \max(10, n/200), B_1 = 20, B_2 = 200$ for ROVE and ROVEs; 2) For continuous space $\Theta$, use $k_1 = \max(30, n/2), k_2 = \max(30, n/200), B_1 = 50, B_2 = 200$ for ROVE and ROVEs; 3) The $\epsilon$ in ROVE and ROVEs is selected such that $\max_{\theta \in \mathcal{S}} (1/B_2) \sum_{b=1}^{B_2} \mathbb{1}(\theta \in \widehat{\Theta}_{k_2}^{\epsilon,b}) \approx 1/2$.

### 3.1 NEURAL NETWORKS FOR REGRESSION

We consider regression problems with multilayer perceptrons (MLPs) on both synthetic and real data. The base learning algorithm splits the data into training (70%) and validation (30%), and uses Adam to minimize mean squared error (MSE), with early stopping triggered when the validation improvement falls below 3% between epochs. The architecture details of the MLPs are provided in Appendix D.1. Note that MoVE is not included in this comparison as it's applicable to discrete problems only.

**Setup for Synthetic Data** Input-output pairs $(X, Y)$ are generated as $Y = (1/50) \cdot \sum_{j=1}^{50} \log(X_j + 1) + \varepsilon$, where each $X_j$ is drawn independently from $\mathrm{Unif}(0, 2 + 198(j-1)/49)$, and the noise $\varepsilon$ is independent of $X$ with zero mean. We consider both standard Gaussian noise and Pareto noise $\varepsilon = \varepsilon_1 - \varepsilon_2$, where each $\varepsilon_i \sim \mathrm{Pareto}(2.1)$. The out-of-sample performance is estimated on a common test set of one million samples. Each algorithm is repeatedly applied to 200 independently generated datasets to assess the average and tail performance.

**Setup for Real Data** We use six datasets from the UCI Machine Learning Repository (Blake, 1998): *Wine Quality* (Cortez et al., 2009), *Bike Sharing* (Fanaee-T, 2013), *Online News* (Fernandes et al., 2015), *Appliances Energy* (Candanedo, 2017), *Superconductivity* (Hamidieh, 2018), and *Gas Turbine Emission* (gas, 2019). Each dataset is standardized (zero mean, unit variance). To evaluate the average and tail performance, we permute each dataset 100 times, and each time use the first half for training and the second for testing.

**Result.** As shown in Figure 1, in heavy-tailed noise settings (Figures 1a–1c), both ROVE and ROVEs significantly outperform the base algorithm in terms of both expected out-of-sample MSE and tail performance under all sample sizes $n$. Notably, the performance improvement becomes more

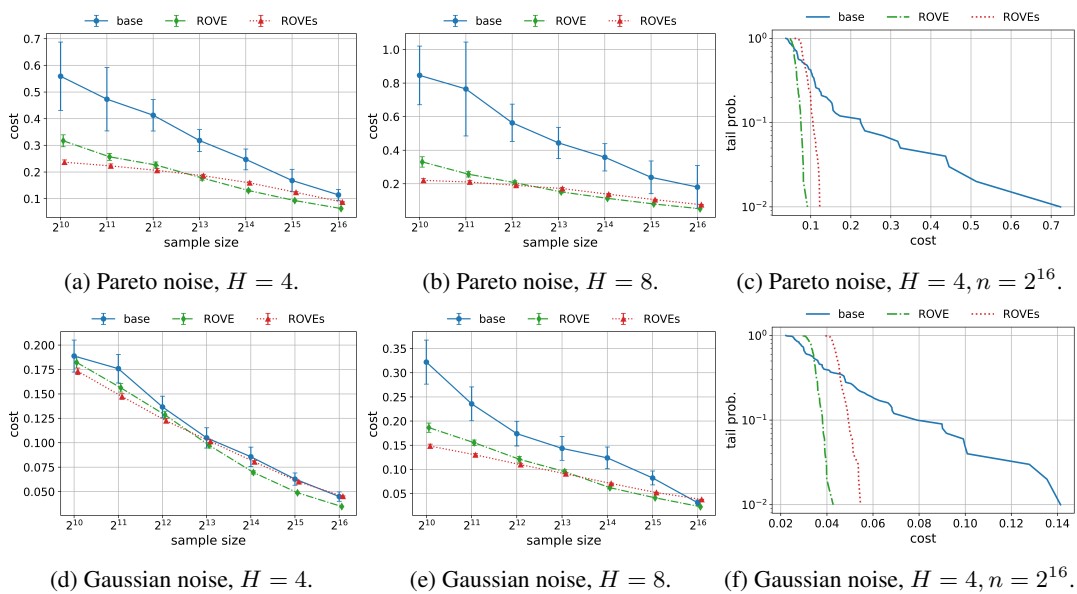

Figure 1: Results of neural networks on synthetic data. (a)(b)(d)(e): Expected out-of-sample costs (MSE) with $95\%$ confidence intervals under different noise distributions and varying numbers of hidden layers ($H$). (c) and (f): Tail probabilities of out-of-sample costs.

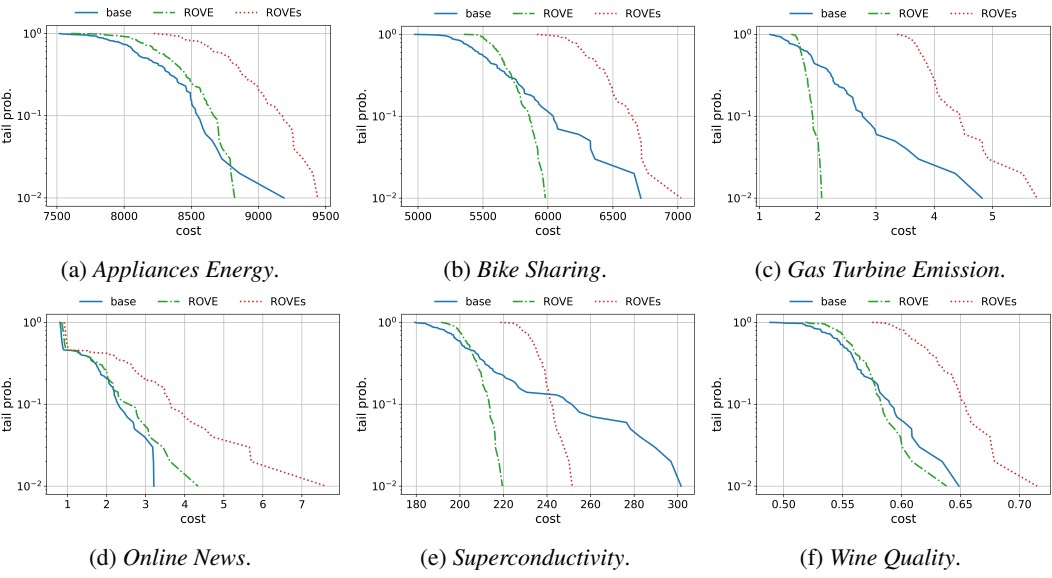

Figure 2: Results of neural networks with $4$ hidden layers on six real datasets, in terms of tail probabilities of out-of-sample costs (MSE).

pronounced with deeper networks ($H = 8$), indicating that the benefits of ROVE and ROVEs are more apparent in models with higher expressiveness and lower bias.

In light-tailed settings (Figures 1d–1f), ROVE and ROVEs show comparable expected out-of-sample performance to the base when $H = 4$, but outperform the base as $H$ increases. Additionally, ROVE and ROVEs outperform the base in tail probabilities even when $H = 4$. This indicates that ROVE and ROVEs provide better generalization as the model complexity grows even for light-tailed problems. Similar results for MLPs with $2$ and $6$ hidden layers can be found in Appendix D.4, where results on least squares regression and Ridge regression are also provided.

On real datasets (Figure 2), ROVE exhibits much lighter tails compared to the base on three out of six datasets, and similar tail behavior on the other three. ROVEs, however, underperforms the base in these real-world scenarios, potentially due to the data split that compromises its statistical power.

## 3.2 STOCHASTIC PROGRAMS

**Setup.** We consider four discrete stochastic programs: resource allocation, supply chain network design, maximum weight matching, and stochastic linear programming, alongside one continuous mean-variance portfolio optimization. All problems are designed to possess heavy-tailed uncertainties. For the stochastic linear program, instances with varying tail heaviness are explored to study its impact on algorithm performance. The base learning algorithm for all the problems is the SAA. Detailed descriptions of the problems are deferred to Appendix D.2 and results using DRO as the base algorithm are provided in Appendix D.4.

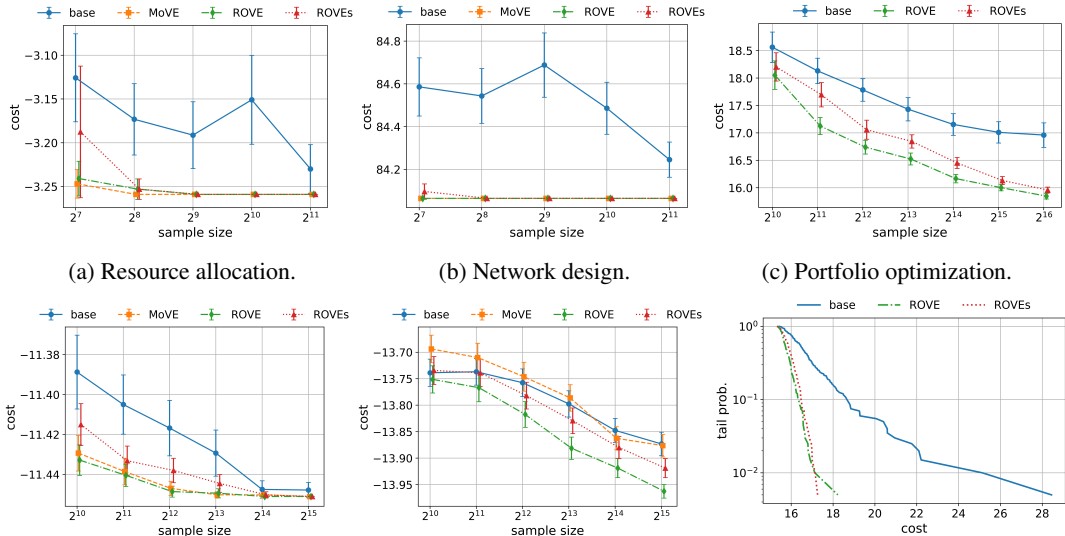

(a) Resource allocation.   (b) Network design.   (c) Portfolio optimization.

(d) Maximum weight matching.   (e) Linear program (multiple optima).   (f) Tail of portfolio opt., $n = 2^{16}$.

Figure 3: Results for stochastic programs. (a)-(e): Expected out-of-sample costs with $95\%$ confidence intervals. (f): Tail probabilities of out-of-sample costs for mean-variance portfolio optimization. All maximization problems are converted to minimization by negating their objectives, and the generic term "cost" refers to the minimizing objective.

**Result.** Figure 3 shows that our ensembling methods generally outperform the base algorithm in all cases, except for the linear program case (Figure 3e). Notably, ROVE still outperforms the base in the linear program case, demonstrating its robustness, while MoVE performs slightly worse than the base under small sample sizes. Comparing ROVE and ROVEs, ROVE consistently exhibits superior performance than ROVEs in all cases.

When there is a unique optimal solution, MoVE and ROVE perform similarly, both generally better than ROVEs, as seen in Figures 3a-3d. However, in cases with multiple optima (Figures 3e and 4a), the performance of MoVE deteriorates while ROVE and ROVEs stay strong. This is in accordance with our discussion on the advantage of $\epsilon$-optimality vote in Section 2.2. Additional results in Appendix D.4 shall further explain that optima multiplicity weakens the base learner for MoVE in the sense of decreasing the $\eta_{k,\delta}$ and hence inflating the tail bound in Theorem 1.

As shown in Figure 4a, the performance gap between ROVE, ROVEs, and the base algorithm becomes increasingly significant as the tail of the uncertainty becomes heavier. This supports the effectiveness of ROVE and ROVEs in handling heavy-tailed uncertainty, where the base algorithm's performance suffers. Note that here MoVE behaves similarly as the base due to optima multiplicity.

The running time comparison in Figure 4b shows that, despite requiring multiple runs on subsamples, our ensembling methods do not introduce a significantly higher computational burden compared to

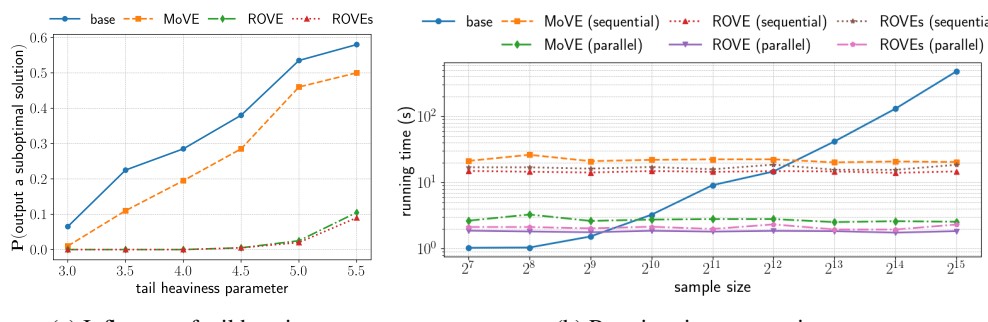

(a) Influence of tail heaviness.

(b) Running time comparison.

Figure 4: (a): Influence of tail heaviness in the stochastic linear program with multiple optima with $n = 10^6$. Hyperparameters: $k = 50, B = 2000$ for MoVE, $k_1 = k_2 = 50, B_1 = 200, B_2 = 5000$ for ROVE and ROVEs. The tail heaviness parameter corresponds to the mean of the Pareto random coefficient. (b): Running time for supply chain network design. Hyperparameters: $k = 10, B = 200$ for MoVE, $k_1 = k_2 = 10, B_1 = 20, B_2 = 200$ for ROVE and ROVEs. "Sequential" refers to sequential processing of the subsamples; "Parallel" refers to parallel processing with 8 CPU cores.

running the base algorithm on the full sample, and can even be advantageous under large sample sizes. This is because, in problems like DRO (Ben-Tal et al., 2013; Mohajerin Esfahani & Kuhn, 2018) and two-stage stochastic programming, solving the optimization on the full sample often leads to a substantial increase in problem size, as the decision space and constraints grow at least linearly with the sample size. Subsampled optimizations, as performed in our approach, result in smaller, more manageable problems that can be solved more efficiently. Moreover, our theory indicates that solving more than $\mathcal{O}(n/k)$ subsamples does not further improve generalization performance, ensuring that computational efficiency is maintained. Additionally, parallel processing of subsamples further reduces computational time.

Finally, among the three proposed ensemble methods, ROVE is the preferred choice over MoVE and ROVEs for general use as it's applicable to both discrete and continuous problems and consistently delivers superior and stable performance across all scenarios.

## 4 RELATED WORK

This work is closely connected to various topics in optimization and machine learning, and we only review the most relevant ones. See Appendix A for additional literature review.

**Ensemble learning.** Ensemble learning (Dietterich, 2000; Zhou, 2012; Sagi & Rokach, 2018) has been widely studied for improving model performance by combining multiple weak learners into strong ones. Popular ensemble methods include bagging (Breiman, 1996), boosting (Freund et al., 1996) and stacking (Wolpert, 1992; Džeroski & Ženko, 2004). Bagging enhances model stability by training models on different bootstrap samples and combining their predictions through majority voting or averaging, effectively reducing variance, especially for unstable learners like decision trees that underpin random forests (Breiman, 2001). Subagging (Bühlmann & Yu, 2002) is a variant of bagging that constructs the ensemble from subsamples in place of bootstrap samples. Boosting is a sequential process where each subsequent model corrects its predecessors' errors, reducing both bias and variance (Ibragimov & Gusev, 2019; Ghosal & Hooker, 2020). Prominent boosting methods include AdaBoost (Freund et al., 2003), Stochastic Gradient Boosting (SGB) (Friedman, 2001; 2002), and Extreme Gradient Boosting (XGB) (Friedman et al., 2000) which differ in their approaches to weighting training data and hypotheses. Boosting is commonly used with decision trees as Gradient Boosted Decision Trees (GBDT), including XGBoost (Chen & Guestrin, 2016), LightGBM (Ke et al., 2017), and CatBoost (Hancock & Khoshgoftaar, 2020). Instead of using simple aggregation like weighted averaging or majority voting, stacking trains a model to combine base predictions in a more sophisticated way, further improving performance. A key procedural difference of our approach from these ensemble methods is that we perform majority voting at the model level, rather than at the prediction level, to select a single best model from the ensemble. As a result, our method consistently

outputs models within the same space as the base learner, making it applicable to general stochastic optimization problems. In contrast, most existing ensemble methods yield aggregated models outside the base space. Additionally, compared to the bias/variance reduction of typical ensembles, our approach guarantees exponentially decaying excess risk tails and hence is particularly effective in settings with heavy-tailed noise.

**Optimization and learning with heavy tails.** Optimization with heavy-tailed noises has garnered significant attention due to its relevance in traditional fields such as portfolio management (Mainik et al., 2015) and scheduling (Im et al., 2015), as well as emerging domains like large language models (Brown et al., 2020; Achiam et al., 2023). Tail bounds of most existing algorithms are guaranteed to decay exponentially under sub-Gaussian or uniformly bounded costs but deteriorate to a slow polynomial decay under heavy-tailedness (Kaňková & Houda, 2015; Jiang et al., 2020; Jiang & Li, 2021; Oliveira & Thompson, 2023). For SAA or ERM, faster rates are possible under the small-ball (Mendelson, 2018; 2015; Roy et al., 2021) or Bernstein's condition (Dinh et al., 2016) on the function class, while our approach is free from such conditions. Considerable effort has been made to mitigate the adverse effects of heavy-tailedness with robust procedures among which the geometric median (Minsker, 2015), or more generally, median-of-means (MOM) (Lugosi & Mendelson, 2019a;c) approach is most similar to ours. The basic idea there is to estimate a true mean by dividing the data into disjoint subsamples, computing an estimate on each, and then taking the median. Lecué & Lerasle (2019); Lugosi & Mendelson (2019b); Lecué & Lerasle (2020) use MOM in estimating the expected cost and establish exponential tail bounds for the mean squared loss and convex function classes. Hsu & Sabato (2016; 2014) apply MOM directly on the solution level for continuous problems and require strong convexity from the cost to establish generalization bounds. Besides MOM, another approach estimates the expected cost via truncation (Catoni, 2012) and allows heavy tails for linear regression (Audibert & Catoni, 2011; Zhang & Zhou, 2018) or problems with uniformly bounded function classes (Brownlees et al., 2015), but is computationally intractable due to the truncation and thus more of theoretical interest. In contrast, our ensemble approach is a meta algorithm that acts on any learning algorithm to provide exponential tail bounds regardless of the underlying problem characteristics. Relatedly, various techniques such as gradient clipping (Cutkosky & Mehta, 2021; Gorbunov et al., 2020) and MOM (Puchkin et al., 2024) have been adopted in stochastic gradient descent (SGD) algorithms for handling heavy-tailed gradient noises, but their focus is the faster convergence of SGD rather than generalization.

**Machine learning for optimization.** Learning to optimize (L2O) studies the use of machine learning in accelerating existing or discovering novel optimization algorithms. Much effort has been in training models via supervised or reinforcement learning to make critical algorithmic decisions such as cut selection (e.g., Deza & Khalil (2023); Tang et al. (2020)), search strategies (e.g., Khalil et al. (2016); He et al. (2014); Scavuzzo et al. (2022)), scaling (Berthold & Hendel, 2021), and primal heuristics (Shen et al., 2021) in mixed-integer optimization, or even directly generate high-quality solutions (e.g., neural combinatorial optimization pioneered by Bello et al. (2016)). See Chen et al. (2022; 2024); Bengio et al. (2021); Zhang et al. (2023) for comprehensive surveys on L2O. This line of research is orthogonal to our goal, and L2O techniques can work as part of or directly serve as the base learning algorithm within our framework.

## 5 CONCLUSION AND LIMITATION

This paper introduces a novel ensemble technique that significantly improves generalization by aggregating base learners via majority voting. In particular, our approach converts polynomially decaying generalization tails into exponential decay, thus providing order-of-magnitude improvements as opposed to constant factor improvements exhibited by variance reduction. Extensive numerical experiments in both machine learning and stochastic programming validate its effectiveness, especially for scenarios with heavy-tailed data and slow convergence rates. This work underscores the powerful potential of our new ensemble approach across a broad range of machine learning applications.

While our method accelerates tail convergence, it may increase model bias, similar to other subsampling-based techniques like subagging (Bühlmann & Yu, 2002). This makes it best suited for applications with relatively low bias, e.g., when the model is sufficiently expressive.

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

# Supplemental materials

The appendices are organized as follows. In Appendix A, we review additional related work. Appendix B presents additional technical discussion for Theorem 1. Next, in Appendix C, we document the proofs of theoretical results in our paper. Specifically, we introduce some preliminary definitions and lemmas in Appendix C.1. Then, the proof of Theorem 1 can be found in Appendix C.2, and the proof of Theorem 2 can be found in Appendix C.3. Finally, we provide additional numerical experiments in Appendix D.

## APPENDIX A  ADDITIONAL RELATED WORK

**Bagging for stochastic optimization.**  Bagging has been adopted in stochastic optimization for various purposes. The most relevant line of works (Biggs et al., 2023; Perakis & Thayaparan, 2021; Wang et al., 2021; Biggs & Perakis, 2023) study mixed integer reformulations for stochastic optimization with bagging approximated objectives such as random forests and ensembles of neural networks with the ReLU activation. These works focus on computational tractability instead of generalization performance. Anderson & Nguyen (2020) empirically evaluates several statistical techniques including bagging against the plain SAA and finds bagging advantageous for portfolio optimization problems. Birge (2023) investigates a batch mean approach for continuous optimization that creates subsamples by dividing the data set into non-overlapping batches instead of resampling and aggregates SAA solutions on the subsamples via averaging, which is empirically demonstrated to reduce solution errors for constrained and high-dimensional problems. Another related batch of works (Lam & Qian, 2018a;b; Chen & Woodruff, 2024; 2023; Eichhorn & Römisch, 2007) concern the use of bagging for constructing confidence bounds for generalization errors of data-driven solutions, but they do not attempt to improve generalization. Related to bagging, bootstrap has been utilized to quantify algorithmic uncertainties for randomized algorithms such as randomized least-squares algorithms (Lopes et al., 2018), randomized Newton methods (Chen & Lopes, 2020), and stochastic gradient descent (Fang et al., 2018; Zhong et al., 2023), which is orthogonal to our focus on generalization performance.

## APPENDIX B  IMPLICATIONS OF THEOREM 1 FOR STRONG BASE LEARNERS

We provide a brief discussion of Theorem 1 applied to fast convergent base learners. Based on Theorem 1, the way $p_k^{\max}$ and $\mathcal{E}_{k,\delta}$ enter into (9) reflects how the generalization performance of the base learning algorithm is inherited by our framework. To explain, large $p_k^{\max}$ and small $\mathcal{E}_{k,\delta}$ correspond to better generalization of the base learning algorithm. This can be exploited by the bound (9) with the presence of $\max(1 - p_k^{\max}, \ \mathcal{E}_{k,\delta})$, which is captured with our sharper concentration of U-statistics with binary kernels. In particular, for base learning algorithms with fast generalization convergence, say $1 - p_k^{\max} = \mathcal{O}(e^{-k})$ and $\mathcal{E}_{k,\delta} = \mathcal{O}(e^{-k})$ for simplicity, we have $C_1 \max(1 - p_k^{\max}, \ \mathcal{E}_{k,\delta}) = \mathcal{O}(e^{-k})$ and hence the first term in (9) becomes $\mathcal{O}(e^{-n})$ which matches the error of the base learning algorithm applied directly to the full data set.

## APPENDIX C  TECHNICAL PROOFS

### C.1  PRELIMINARIES

An important tool in the development of our theories is the U-statistic that naturally arises in subsampling without replacement. We first present the definition of U-statistic and its concentration properties.

**Definition 1** *Given the i.i.d. data set $\{z_1, \ldots, z_n\} \subset \mathcal{Z}$ and a (not necessarily symmetric) kernel of order $k \leq n$ is a function $\kappa : \mathcal{Z}^k \to \mathbb{R}$ such that $\mathbb{E}\left[|\kappa(z_1, \ldots, z_k)|\right] < \infty$, the U-statistic associated with the kernel $\kappa$ is*

$$U(z_1, \ldots, z_n) = \frac{1}{n(n-1)\cdots(n-k+1)} \sum_{1 \leq i_1, i_2, \cdots, i_k \leq n \text{ s.t. } i_s \neq i_t \ \forall 1 \leq s < t \leq k} \kappa(z_{i_1}, \ldots, z_{i_k}).$$

**Lemma 1 (MGF dominance of U-statistics from Hoeffding (1963))** *For any integer $0 < k \leq n$ and any kernel $\kappa(z_1, \ldots, z_k)$, let $U(z_1, \ldots, z_n)$ be the corresponding U-statistic defined in Definition 1, and*

$$\bar{\kappa}(z_1, \ldots, z_n) = \frac{1}{\lfloor n/k \rfloor} \sum_{i=1}^{\lfloor n/k \rfloor} \kappa(z_{k(i-1)+1}, \ldots, z_{ki}) \tag{11}$$

*be the average of the kernel across the first $\lfloor n/k \rfloor k$ data. Then, for every $t \in \mathbb{R}$, it holds that*

$$\mathbb{E}\left[\exp(tU)\right] \leq \mathbb{E}\left[\exp(t\bar{\kappa})\right].$$

*Proof of Lemma 1.* By symmetry, we have that

$$U(z_1, \ldots, z_n) = \frac{1}{n!} \sum_{\text{bijection } \pi:[n]\to[n]} \bar{\kappa}(z_{\pi(1)}, \ldots, z_{\pi(n)}),$$

where we denote $[n] := \{1, \ldots, n\}$. Then, by the convexity of the exponential function and Jensen's inequality, we have that

$$
\begin{aligned}
\mathbb{E}\left[\exp(tU)\right] &= \mathbb{E}\left[\exp\left(t \cdot \frac{1}{n!} \sum_{\text{bijection } \pi:[n]\to[n]} \bar{\kappa}(z_{\pi(1)}, \ldots, z_{\pi(n)})\right)\right] \\
&\leq \mathbb{E}\left[\frac{1}{n!} \sum_{\text{bijection } \pi:[n]\to[n]} \exp\left(t \cdot \bar{\kappa}(z_{\pi(1)}, \ldots, z_{\pi(n)})\right)\right] \\
&= \mathbb{E}\left[\exp\left(t \cdot \bar{\kappa}(z_1, \ldots, z_n)\right)\right].
\end{aligned}
$$

This completes the proof. $\qquad\qquad\square$

Next, we present our sharper concentration bound for U-statistics with binary kernels:

**Lemma 2 (Concentration bound for U-statistics with binary kernels)** *Let $\kappa(z_1, \ldots, z_k; \omega)$ be a $\{0, 1\}$-valued kernel of order $k \leq n$ that possibly depends on additional randomness $\omega$ that is independent of the data $\{z_1, \ldots, z_n\}$, $\kappa^*(z_1, \ldots, z_k) := \mathbb{E}\left[\kappa(z_1, \ldots, z_k; \omega) | z_1, \ldots, z_k\right]$, and $U(z_1, \ldots, z_n)$ be the U-statistic associated with $\kappa^*$. Then, it holds that*

$$\mathbb{P}\left(U - \mathbb{E}\left[\kappa\right] \geq \epsilon\right) \leq \exp\left(-\frac{n}{2k} \cdot D_{\mathrm{KL}}\left(\mathbb{E}\left[\kappa\right] + \epsilon \| \mathbb{E}\left[\kappa\right]\right)\right),$$

$$\mathbb{P}\left(U - \mathbb{E}\left[\kappa\right] \leq -\epsilon\right) \leq \exp\left(-\frac{n}{2k} \cdot D_{\mathrm{KL}}\left(\mathbb{E}\left[\kappa\right] - \epsilon \| \mathbb{E}\left[\kappa\right]\right)\right),$$

*where $D_{\mathrm{KL}}(p\|q) := p \ln \frac{p}{q} + (1-p) \ln \frac{1-p}{1-q}$ is the KL-divergence between two Bernoulli random variables with parameters $p$ and $q$, respectively.*

*Proof of Lemma 2.* We first consider the direction $U - \mathbb{E}\left[\kappa\right] \geq \epsilon$. Let

$$\tilde{\kappa}^* := \frac{1}{\hat{n}} \sum_{i=1}^{\hat{n}} \kappa^*(z_{k(i-1)+1}, \ldots, z_{ki}),$$

and

$$\tilde{\kappa} := \frac{1}{\hat{n}} \sum_{i=1}^{\hat{n}} \kappa(z_{k(i-1)+1}, \ldots, z_{ki}; \omega_i),$$

where we use the shorthand notation $\hat{n} := \lfloor \frac{n}{k} \rfloor$, and $\omega_i$'s are mutually independent and also independent from $\{z_1, \ldots, z_n\}$. Then, since $\mathbb{E}\left[\kappa\right] = \mathbb{E}\left[\kappa^*\right]$, for all $t > 0$ it holds that

$$
\begin{aligned}
\mathbb{P}\left(U - \mathbb{E}\left[\kappa\right] \geq \epsilon\right) &= \mathbb{P}\left(\exp\left(tU\right) \geq \exp\left(t\left(\mathbb{E}\left[\kappa\right] + \epsilon\right)\right)\right) \\
&\overset{(i)}{\leq} \exp\left(-t\left(\mathbb{E}\left[\kappa\right] + \epsilon\right)\right) \cdot \mathbb{E}\left[\exp\left(tU\right)\right] \\
&\overset{(ii)}{\leq} \exp\left(-t\left(\mathbb{E}\left[\kappa\right] + \epsilon\right)\right) \cdot \mathbb{E}\left[\exp\left(t\tilde{\kappa}^*\right)\right] \\
&\overset{(iii)}{\leq} \exp\left(-t\left(\mathbb{E}\left[\kappa\right] + \epsilon\right)\right) \cdot \mathbb{E}\left[\exp\left(t\tilde{\kappa}\right)\right],
\end{aligned} \tag{12}
$$

where we apply the Markov inequality in $(i)$, step $(ii)$ is due to Lemma 1, and step $(iii)$ uses Jensen's inequality and the convexity of the exponential function. Due to independence, $\tilde{\kappa}$ can be viewed as the sample average of $\hat{n}$ i.i.d. Bernoulli random variables, i.e., $\tilde{\kappa} \sim \frac{1}{\hat{n}} \sum_{i=1}^{\hat{n}} \text{Bernoulli}(\mathbb{E}[\kappa])$. Hence, we have that

$$
\begin{aligned}
\mathbb{E}\left[\exp(t\tilde{\kappa})\right] &= \mathbb{E}\left[\exp\left(\frac{t}{\hat{n}} \sum_{i=1}^{\hat{n}} \text{Bernoulli}(\mathbb{E}[\kappa])\right)\right] \\
&= \left(\mathbb{E}\left[\exp\left(\frac{t}{\hat{n}} \text{Bernoulli}(\mathbb{E}[\kappa])\right)\right]\right)^{\hat{n}} \\
&= \left[(1 - \mathbb{E}[\kappa]) + \mathbb{E}[\kappa] \cdot \exp\left(\frac{t}{\hat{n}}\right)\right]^{\hat{n}},
\end{aligned}
\tag{13}
$$

where we use the moment-generating function of Bernoulli random variables in the last line. Substituting (13) into (12), we have that

$$
\mathbb{P}(U - \mathbb{E}[\kappa] \geq \epsilon) \leq \exp(-t(\mathbb{E}[\kappa] + \epsilon)) \cdot \left[(1 - \mathbb{E}[\kappa]) + \mathbb{E}[\kappa] \cdot \exp\left(\frac{t}{\hat{n}}\right)\right]^{\hat{n}} =: f(t). \tag{14}
$$

Now, we consider minimizing $f(t)$ for $t > 0$. Let $g(t) = \log f(t)$, then it holds that

$$
g'(t) = -(\mathbb{E}[\kappa] + \epsilon) + \frac{\mathbb{E}[\kappa] \cdot \exp\left(\frac{t}{\hat{n}}\right)}{(1 - \mathbb{E}[\kappa]) + \mathbb{E}[\kappa] \cdot \exp\left(\frac{t}{\hat{n}}\right)}.
$$

By setting $g'(t) = 0$, it is easy to verify that the minimum point of $f(t)$, denoted by $t^\star$, satisfies that

$$
\begin{aligned}
\mathbb{E}[\kappa] \cdot \exp\left(\frac{t}{\hat{n}}\right) \cdot (1 - \mathbb{E}[\kappa] - \epsilon) &= (1 - \mathbb{E}[\kappa]) \cdot (\mathbb{E}[\kappa] + \epsilon) \\
\Leftrightarrow \quad \exp(t) &= \left[\frac{(1 - \mathbb{E}[\kappa]) \cdot (\mathbb{E}[\kappa] + \epsilon)}{\mathbb{E}[\kappa] \cdot (1 - \mathbb{E}[\kappa] - \epsilon)}\right]^{\hat{n}}.
\end{aligned}
\tag{15}
$$

Substituting (15) into (14) gives

$$
\begin{aligned}
\mathbb{P}(U - \mathbb{E}[\kappa] \geq \epsilon) &\leq \left(\frac{1 - \mathbb{E}[\kappa]}{1 - \mathbb{E}[\kappa] - \epsilon}\right)^{\hat{n}} \cdot \left[\frac{\mathbb{E}[\kappa] \cdot (1 - \mathbb{E}[\kappa] - \epsilon)}{(1 - \mathbb{E}[\kappa])(\mathbb{E}[\kappa] + \epsilon)}\right]^{\hat{n}(\mathbb{E}[\kappa] + \epsilon)} \\
&= \left[\left(\frac{1 - \mathbb{E}[\kappa]}{1 - \mathbb{E}[\kappa] - \epsilon}\right)^{1 - \mathbb{E}[\kappa] - \epsilon} \cdot \left(\frac{\mathbb{E}[\kappa]}{\mathbb{E}[\kappa] + \epsilon}\right)^{\mathbb{E}[\kappa] + \epsilon}\right]^{\hat{n}} \\
&= \exp(-\hat{n} \cdot D_{\text{KL}}(\mathbb{E}[\kappa] + \epsilon \| \mathbb{E}[\kappa])).
\end{aligned}
\tag{16}
$$

Since $n/k \leq 2\hat{n}$, the first bound immediately follows from (16).

Since $D_{\text{KL}}(p\|q) = D_{\text{KL}}(1 - p\|1 - q)$, the bound for the reverse side $U - \mathbb{E}[\kappa] \leq -\epsilon$ then follows by applying the first bound to the flipped binary kernel $1 - \kappa$ and $1 - U$. This completes the proof of Lemma 2. $\qquad\square$

Next lemma gives lower bounds for KL divergences which help analyze the bounds in Lemma 2:

**Lemma 3** *Let* $D_{\text{KL}}(p\|q) := p \ln \frac{p}{q} + (1 - p) \ln \frac{1-p}{1-q}$ *be the KL-divergence between two Bernoulli random variables with parameters* $p$ *and* $q$, *respectively. Then, it holds that*

$$
D_{\text{KL}}(p\|q) \geq p \ln \frac{p}{q} + q - p. \tag{17}
$$

*If* $p \in [\gamma, 1 - \gamma]$ *for some* $\gamma \in (0, \frac{1}{2}]$, *it also holds that*

$$
D_{\text{KL}}(p\|q) \geq -\ln(2(q(1 - q))^\gamma). \tag{18}
$$

*Proof of Lemma 3.* To show (17), some basic calculus shows that for any fixed $q$, the function $g(p) := (1 - p) \ln \frac{1-p}{1-q}$ is convex in $p$, and we have that

$$
g(q) = 0, g'(q) = -1.
$$

Therefore $g(p) \geq g(q) + g'(q)(p - q) = q - p$, which implies (17) immediately.

The lower bound (18) follows from

$$
\begin{aligned}
D_{\mathrm{KL}}(p\|q) &\geq -p \ln q - (1-p)\ln(1-q) + \min_{p \in [\gamma, 1-\gamma]} \{p \ln p + (1-p)\ln(1-p)\} \\
&\geq -\gamma \ln q - \gamma \ln(1-q) - \ln 2 = -\ln(2(q(1-q))^\gamma).
\end{aligned}
$$

This completes the proof of Lemma 3. $\qquad\square$

To incorporate all the proposed algorithms in a unified theoretical framework, we consider a set-valued mapping

$$
\mathbb{A}(z_1, \ldots, z_k; \omega) : \mathcal{Z}^k \times \Omega \to 2^\Theta \tag{19}
$$

where $\omega$ denotes algorithmic randomness that is independent of the data $\{z_1, \ldots, z_k\}$. Each of our proposed algorithms attempts to solve the probability-maximization problem

$$
\max_{\theta \in \Theta} \hat{p}_k(\theta) := \mathbb{P}_* \left(\theta \in \mathbb{A}(z_1^*, \ldots, z_k^*; \omega)\right), \tag{20}
$$

for a certain choice of $\mathbb{A}$, where $\{z_1^*, \ldots, z_k^*\}$ is subsampled from the i.i.d. data $\{z_1, \ldots, z_n\}$ uniformly without replacement, and $\mathbb{P}_*$ denotes the probability with respect to the algorithmic randomness $\omega$ and the subsampling randomness conditioned on the data. Note that this problem is an empirical approximation of the problem

$$
\max_{\theta \in \Theta} p_k(\theta) := \mathbb{P}\left(\theta \in \mathbb{A}(z_1, \ldots, z_k; \omega)\right). \tag{21}
$$

The problem actually solved with a finite number of subsamples is

$$
\max_{\theta \in \Theta} \bar{p}_k(\theta) := \frac{1}{B} \sum_{b=1}^{B} \mathbb{1}(\theta \in \mathbb{A}(z_1^b, \ldots, z_k^b; \omega_b)). \tag{22}
$$

Specifically, Algorithm 1 uses

$$
\mathbb{A}(z_1^*, \ldots, z_k^*; \omega) = \{\mathcal{A}(z_1^*, \ldots, z_k^*; \omega)\} \tag{23}
$$

where $\mathcal{A}$ denotes the base learning algorithm, and Algorithm 2 uses

$$
\mathbb{A}(z_1^*, \ldots, z_{k_2}^*; \omega) = \left\{ \theta \in \mathcal{S} : \frac{1}{k_2} \sum_{i=1}^{k_2} l(\theta, z_i^*) \leq \min_{\theta' \in \mathcal{S}} \frac{1}{k_2} \sum_{i=1}^{k_2} l(\theta', z_i^*) + \epsilon \right\} \tag{24}
$$

conditioned on the solution set $\mathcal{S}$ retrieved in Phase I. Note that no algorithmic randomness is involved in (24) once the set $\mathcal{S}$ is given. We define:

**Definition 2** *For any $\delta \in [0, 1]$, let*

$$
\mathcal{P}_k^\delta := \{\theta \in \Theta : p_k(\theta) \geq \max_{\theta' \in \Theta} p_k(\theta') - \delta\} \tag{25}
$$

*be the set of $\delta$-optimal solutions of problem (21). Let*

$$
\theta_k^{\max} \in \arg\max_{\theta \in \Theta} p_k(\theta)
$$

*be a solution with maximum probability that is chosen in a unique manner if there are multiple such solutions. Let*

$$
\widehat{\mathcal{P}}_k^\delta := \{\theta \in \Theta : \hat{p}_k(\theta) \geq \hat{p}_k(\theta_k^{\max}) - \delta\} \tag{26}
$$

*be the set of $\delta$-optimal solutions relative to $\theta_k^{\max}$ for problem (20).*

and

**Definition 3** *Let*

$$
\Theta^\delta := \left\{\theta \in \Theta : L(\theta) \leq \min_{\theta' \in \Theta} L(\theta') + \delta\right\} \tag{27}
$$

*be the set of $\delta$-optimal solutions of problem (1). In particular, $\Theta^0$ represents the set of optimal solutions. Let*

$$
\widehat{\Theta}_k^\delta := \left\{\theta \in \Theta : \frac{1}{k} \sum_{i=1}^{k} l(\theta, z_i) \leq \min_{\theta' \in \Theta} \frac{1}{k} \sum_{i=1}^{k} l(\theta', z_i) + \delta\right\} \tag{28}
$$

*be the set of $\delta$-optimal solutions of the SAA with i.i.d. data $(z_1, \ldots, z_k)$.*

## C.2 PROOF FOR THEOREM 1

We consider Algorithm 3, a more general version of Algorithm 1 that operates on the set-valued learning algorithm $\mathbb{A}$ in (19) and reduces to exactly Algorithm 1 in the special case (23). Again we omit the algorithmic randomness $\omega$ in $\mathbb{A}$ for convenience.

---

**Algorithm 3** Majority Vote Ensembling for Set-Valued Learning Algorithms

---

1: Input: A set-valued learning algorithm $\mathbb{A}$, $n$ i.i.d. observations $\mathbf{z}_{1:n} = (z_1, \ldots, z_n)$, positive integers $k < n$, and ensemble size $B$.
2: **for** $b = 1$ **to** $B$ **do**
3:     Randomly sample $\mathbf{z}_k^b = (z_1^b, \ldots, z_k^b)$ uniformly from $\mathbf{z}_{1:n}$ without replacement, and obtain $\Theta_k^b = \mathbb{A}(z_1^b, \ldots, z_k^b)$
4: **end for**
5: Output $\hat{\theta}_n \in \arg\max_{\theta \in \Theta} \sum_{b=1}^{B} \mathbb{1}(\theta \in \Theta_k^b)$.

---

We have the following finite-sample result for Algorithm 3:

**Theorem 3 (Finite-sample bound for Algorithm 3)** *Consider discrete decision space $\Theta$. Recall $p_k(\theta)$ defined in (21). Let $p_k^{\max} := \max_{\theta \in \Theta} p_k(\theta)$ and*

$$\bar{\eta}_{k,\delta} := p_k^{\max} - \max_{\theta \in \Theta \setminus \Theta^\delta} p_k(\theta), \tag{29}$$

*where $\max_{\theta \in \Theta \setminus \Theta^\delta} p_k(\theta)$ evaluates to 0 if $\Theta \setminus \Theta^\delta$ is empty. For every $k \leq n$ and $\delta \geq 0$ such that $\bar{\eta}_{k,\delta} > 0$, the solution output by Algorithm 3 satisfies that*

$$\mathbb{P}\left( L(\hat{\theta}_n) > \min_{\theta \in \Theta} L(\theta) + \delta \right)$$

$$\leq |\Theta| \left[ \exp\left( -\frac{n}{2k} \cdot D_{\mathrm{KL}}\left( p_k^{\max} - \frac{3\eta}{4} \middle\| p_k^{\max} - \eta \right) \right) + 2\exp\left( -\frac{n}{2k} \cdot D_{\mathrm{KL}}\left( p_k^{\max} - \frac{\eta}{4} \middle\| p_k^{\max} \right) \right) + \right.$$

$$\exp\left( -\frac{B}{24} \cdot \frac{\eta^2}{\min\left( p_k^{\max}, 1 - p_k^{\max} \right) + 3\eta/4} \right) +$$

$$\left. \mathbb{1}\left( p_k^{\max} + \frac{\eta}{4} \leq 1 \right) \cdot \exp\left( -\frac{n}{2k} \cdot D_{\mathrm{KL}}\left( p_k^{\max} + \frac{\eta}{4} \middle\| p_k^{\max} \right) - \frac{B}{24} \cdot \frac{\eta^2}{1 - p_k^{\max} + \eta/4} \right) \right]$$

$$\tag{30}$$

*for every $\eta \in (0, \bar{\eta}_{k,\delta}]$. In particular, if $\bar{\eta}_{k,\delta} > 4/5$, (30) is further bounded by*

$$|\Theta| \left( 3 \min\left( e^{-2/5}, C_1 \max(1 - p_k^{\max}, \max_{\theta \in \Theta \setminus \Theta^\delta} p_k(\theta)) \right)^{\frac{n}{C_2 k}} + \exp\left( -\frac{B}{C_3} \right) \right), \tag{31}$$

*where $C_1, C_2, C_3 > 0$ are universal constants, and $D_{\mathrm{KL}}(p \| q) := p \ln \frac{p}{q} + (1 - p) \ln \frac{1-p}{1-q}$ is the Kullback–Leibler divergence between two Bernoulli distributions with means $p$ and $q$.*

*Proof of Theorem 3.* We first prove excess risk tail bounds for the problem (21), split into two lemmas, Lemmas 4 and 5 below.

**Lemma 4** *Consider discrete decision space $\Theta$. Recall from Definition 2 that $p_k^{\max} = p_k(\theta_k^{\max})$ holds for $\theta_k^{\max}$. For every $0 \leq \epsilon \leq \delta \leq p_k^{\max}$, it holds that*

$$\mathbb{P}\left( \widehat{\mathcal{P}}_k^\epsilon \not\subseteq \mathcal{P}_k^\delta \right) \leq |\Theta| \left[ \exp\left( -\frac{n}{2k} \cdot D_{\mathrm{KL}}\left( p_k^{\max} - \frac{\delta + \epsilon}{2} \middle\| p_k^{\max} - \delta \right) \right) \right.$$

$$\left. + \exp\left( -\frac{n}{2k} \cdot D_{\mathrm{KL}}\left( p_k^{\max} - \frac{\delta - \epsilon}{2} \middle\| p_k^{\max} \right) \right) \right].$$

*Proof of Lemma 4.* By Definition 2, we observe the following equivalence

$$\left\{ \widehat{\mathcal{P}}_k^\epsilon \not\subseteq \mathcal{P}_k^\delta \right\} = \bigcup_{\theta \in \Theta \setminus \mathcal{P}_k^\delta} \left\{ \theta \in \widehat{\mathcal{P}}_k^\epsilon \right\} = \bigcup_{\theta \in \Theta \setminus \mathcal{P}_k^\delta} \left\{ \hat{p}_k(\theta) \geq \hat{p}_k\left( \theta_k^{\max} \right) - \epsilon \right\}.$$

Hence, by the union bound, it holds that

$$\mathbb{P}\left(\widehat{\mathcal{P}}_k^\epsilon \nsubseteq \mathcal{P}_k^\delta\right) \leq \sum_{\theta \in \Theta \backslash \mathcal{P}_k^\delta} \mathbb{P}\left(\hat{p}_k(\theta) \geq \hat{p}_k(\theta_k^{\max}) - \epsilon\right).$$

We further bound the probability $\mathbb{P}\left(\{\hat{p}_k(\theta) \geq \hat{p}_k(\theta_k^{\max}) - \epsilon\}\right)$ as follows

$$\mathbb{P}\left(\hat{p}_k(\theta) \geq \hat{p}_k(\theta_k^{\max}) - \epsilon\right)$$

$$\leq \quad \mathbb{P}\left(\left\{\hat{p}_k(\theta) \geq p_k(\theta_k^{\max}) - \frac{\delta + \epsilon}{2}\right\} \cap \left\{\hat{p}_k(\theta_k^{\max}) \leq p_k(\theta_k^{\max}) - \frac{\delta - \epsilon}{2}\right\}\right)$$

$$\leq \quad \mathbb{P}\left(\hat{p}_k(\theta) \geq p_k(\theta_k^{\max}) - \frac{\delta + \epsilon}{2}\right) + \mathbb{P}\left(\hat{p}_k(\theta_k^{\max}) \leq p_k(\theta_k^{\max}) - \frac{\delta - \epsilon}{2}\right). \quad (32)$$

On one hand, the first probability in (32) is solely determined by and increasing in $p_k(\theta) = \mathbb{E}\left[\hat{p}_k(\theta)\right]$. On the other hand, we have $p_k(\theta) < p_k(\theta_k^{\max}) - \delta$ for every $\theta \in \Theta \backslash \mathcal{P}_k^\delta$ by the definition of $\mathcal{P}_k^\delta$. Therefore we can slightly abuse the notation to write

$$\mathbb{P}\left(\hat{p}_k(\theta) \geq \hat{p}_k(\theta_k^{\max}) - \epsilon\right) \quad \leq \quad \mathbb{P}\left(\hat{p}_k(\theta) \geq p_k(\theta_k^{\max}) - \frac{\delta + \epsilon}{2}\Big| p_k(\theta) = p_k(\theta_k^{\max}) - \delta\right)$$

$$+ \mathbb{P}\left(\hat{p}_k(\theta_k^{\max}) \leq p_k(\theta_k^{\max}) - \frac{\delta - \epsilon}{2}\right)$$

$$\leq \quad \mathbb{P}\left(\hat{p}_k(\theta) - p_k(\theta) \geq \frac{\delta - \epsilon}{2}\Big| p_k(\theta) = p_k(\theta_k^{\max}) - \delta\right)$$

$$+ \mathbb{P}\left(\hat{p}_k(\theta_k^{\max}) - p_k(\theta_k^{\max}) \leq -\frac{\delta - \epsilon}{2}\right).$$

Note that, with $\kappa(z_1, \ldots, z_k; \omega) := \mathbf{1}\left(\theta \in \mathbb{A}(z_1, \ldots, z_k; \omega)\right)$, the probability $\hat{p}_k(\theta)$ can be viewed as a U-statistic with the kernel $\kappa^*(z_1, \ldots, z_k) := \mathbb{E}\left[\kappa(z_1, \ldots, z_k; \omega)|z_1, \ldots, z_k\right]$. A similar representation holds for $\hat{p}_k(\theta_k^{\max})$ as well. Therefore, we can apply Lemma 2 to conclude that

$$\mathbb{P}\left(\widehat{\mathcal{P}}_k^\epsilon \nsubseteq \mathcal{P}_k^\delta\right) \leq \sum_{\theta \in \Theta \backslash \mathcal{P}_k^\delta} \mathbb{P}\left(\hat{p}_k(\theta) \geq \hat{p}_k(\theta_k^{\max}) - \epsilon\right)$$

$$\leq \left|\Theta \backslash \mathcal{P}_k^\delta\right| \left[\mathbb{P}\left(\hat{p}_k(\theta) - p_k(\theta) \geq \frac{\delta - \epsilon}{2}\Big| p_k(\theta) = p_k(\theta_k^{\max}) - \delta\right)\right.$$

$$\left. + \mathbb{P}\left(p_k(\theta_k^{\max}) - \hat{p}_k(\theta_k^{\max}) \leq -\frac{\delta - \epsilon}{2}\right)\right]$$

$$\leq |\Theta|\left[\exp\left(-\frac{n}{2k} \cdot D_{\text{KL}}\left(p_k(\theta_k^{\max}) - \delta + \frac{\delta - \epsilon}{2}\Big\| p_k(\theta_k^{\max}) - \delta\right)\right)\right.$$

$$\left. + \exp\left(-\frac{n}{2k} \cdot D_{\text{KL}}\left(p_k(\theta_k^{\max}) - \frac{\delta - \epsilon}{2}\Big\| p_k(\theta_k^{\max})\right)\right)\right],$$

which completes the proof of Lemma 4. $\qquad\square$

**Lemma 5** *Consider discrete decision space $\Theta$. For every $\epsilon \in [0, 1]$ it holds for the solution output by Algorithm 3 that*

$$\mathbb{P}_*\left(\hat{\theta}_n \notin \widehat{\mathcal{P}}_k^\epsilon\right) \leq |\Theta| \cdot \exp\left(-\frac{B}{6} \cdot \frac{\epsilon^2}{\min\left(\hat{p}_k(\theta_k^{\max}), 1 - \hat{p}_k(\theta_k^{\max})\right) + \epsilon}\right),$$

*where $|\cdot|$ denotes the cardinality of a set and $\mathbb{P}_*$ denotes the probability with respect to both the resampling randomness conditioned on the observations and the algorithmic randomness.*

*Proof of Lemma 5.* We observe that $\bar{p}_k(\theta)$ is an conditionally unbiased estimator for $\hat{p}_k(\theta)$, i.e., $\mathbb{E}_*\left[\bar{p}_k(\theta)\right] = \hat{p}_k(\theta)$. We can express the difference between $\bar{p}_k(\theta)$ and $\bar{p}_k(\theta_k^{\max})$ as the sample average

$$\bar{p}_k(\theta) - \bar{p}_k(\theta_k^{\max}) = \frac{1}{B}\sum_{b=1}^{B}\left[\mathbb{1}(\theta \in \mathbb{A}(z_1^b, \ldots, z_k^b)) - \mathbb{1}(\theta_k^{\max} \in \mathbb{A}(z_1^b, \ldots, z_k^b))\right],$$

whose expectation is equal to $\hat{p}_k(\theta) - \hat{p}_k(\theta_k^{\max})$. We denote by

$$\mathbb{1}_\theta^* := \mathbb{1}(\theta \in \mathbb{A}(z_1^*, \ldots, z_k^*)) \text{ for } \theta \in \Theta$$

for convenience, where $(z_1^*, \ldots, z_k^*)$ represents a random subsample. Then by Bernstein's inequality, we have every $t \geq 0$ that

$$\mathbb{P}_* \left( \bar{p}_k(\theta) - \bar{p}_k(\hat{\theta}_k^{\max}) - (\hat{p}_k(\theta) - \hat{p}_k(\theta_k^{\max})) \geq t \right)$$

$$\leq \exp\left( -B \cdot \frac{t^2}{2\mathrm{Var}_*(\mathbb{1}_\theta^* - \mathbb{1}_{\theta_k^{\max}}^*) + 4/3 \cdot t} \right). \tag{33}$$

Since

$$\mathrm{Var}_*(\mathbb{1}_\theta^* - \mathbb{1}_{\theta_k^{\max}}^*) \leq \mathbb{E}_* \left[ (\mathbb{1}_\theta^* - \mathbb{1}_{\theta_k^{\max}}^*)^2 \right]$$
$$\leq \hat{p}_k(\theta) + \hat{p}_k(\theta_k^{\max}) \leq 2\hat{p}_k(\theta_k^{\max}),$$

and

$$\mathrm{Var}_*(\mathbb{1}_\theta^* - \mathbb{1}_{\theta_k^{\max}}^*) \leq \mathrm{Var}_*(1 - \mathbb{1}_\theta^* - 1 + \mathbb{1}_{\theta_k^{\max}}^*)$$
$$\leq \mathbb{E}_* \left[ (1 - \mathbb{1}_\theta^* - 1 + \mathbb{1}_{\theta_k^{\max}}^*)^2 \right]$$
$$\leq 1 - \hat{p}_k(\theta) + 1 - \hat{p}_k(\theta_k^{\max}) \leq 2(1 - \hat{p}_k(\theta)),$$

we have $\mathrm{Var}_*(\mathbb{1}_\theta^* - \mathbb{1}_{\theta_k^{\max}}^*) \leq 2\min(\hat{p}_k(\theta_k^{\max}), 1 - \hat{p}_k(\theta))$. Substituting this bound to (33) and taking $t = \hat{p}_k(\theta_k^{\max}) - \hat{p}_k(\theta)$ lead to

$$\mathbb{P}_* \left( \bar{p}_k(\theta) - \bar{p}_k(\hat{\theta}_k^{\max}) \geq 0 \right) \leq \exp\left( -B \cdot \frac{(\hat{p}_k(\theta_k^{\max}) - \hat{p}_k(\theta))^2}{4\min(\hat{p}_k(\theta_k^{\max}), 1 - \hat{p}_k(\theta)) + 4/3 \cdot (\hat{p}_k(\theta_k^{\max}) - \hat{p}_k(\theta))} \right)$$

$$\leq \exp\left( -B \cdot \frac{(\hat{p}_k(\theta_k^{\max}) - \hat{p}_k(\theta))^2}{4\min(\hat{p}_k(\theta_k^{\max}), 1 - \hat{p}_k(\theta_k^{\max})) + 16/3 \cdot (\hat{p}_k(\theta_k^{\max}) - \hat{p}_k(\theta))} \right)$$

$$\leq \exp\left( -\frac{B}{6} \cdot \frac{(\hat{p}_k(\theta_k^{\max}) - \hat{p}_k(\theta))^2}{\min(\hat{p}_k(\theta_k^{\max}), 1 - \hat{p}_k(\theta_k^{\max})) + \hat{p}_k(\theta_k^{\max}) - \hat{p}_k(\theta)} \right).$$

Therefore, we have that

$$\mathbb{P}_* \left( \hat{\theta}_n \notin \widehat{\mathcal{P}}_k^\epsilon \right) = \mathbb{P}_* \left( \bigcup_{\theta \in \Theta \backslash \widehat{\mathcal{P}}_k^\epsilon} \left\{ \bar{p}_k(\theta) = \max_{\theta' \in \Theta} \bar{p}_k(\theta') \right\} \right)$$

$$\leq \sum_{\theta \in \Theta \backslash \widehat{\mathcal{P}}_k^\epsilon} \mathbb{P}_* \left( \bar{p}_k(\theta) = \max_{\theta' \in \Theta} \bar{p}_k(\theta') \right)$$

$$\leq \sum_{\theta \in \Theta \backslash \widehat{\mathcal{P}}_k^\epsilon} \mathbb{P}_* \left( \bar{p}_k(\theta) \geq \bar{p}_k(\theta_k^{\max}) \right)$$

$$\leq \sum_{\theta \in \Theta \backslash \widehat{\mathcal{P}}_k^\epsilon} \exp\left( -\frac{B}{6} \cdot \frac{(\hat{p}_k(\theta_k^{\max}) - \hat{p}_k(\theta))^2}{\min(\hat{p}_k(\theta_k^{\max}), 1 - \hat{p}_k(\theta_k^{\max})) + \hat{p}_k(\theta_k^{\max}) - \hat{p}_k(\theta)} \right).$$

Note that the function $x^2/(\min(\hat{p}_k(\theta_k^{\max}), 1 - \hat{p}_k(\theta_k^{\max})) + x)$ in $x \in [0,1]$ is monotonically increasing and that $\hat{p}_k(\theta_k^{\max}) - \hat{p}_k(\theta) > \epsilon$ for all $\theta \in \Theta \backslash \widehat{\mathcal{P}}_k^\epsilon$. Therefore, we can further bound the probability as

$$\mathbb{P}_* \left( \hat{\theta}_n \notin \widehat{\mathcal{P}}_k^\epsilon \right) \leq \left| \Theta \backslash \widehat{\mathcal{P}}_k^\epsilon \right| \cdot \exp\left( -\frac{B}{6} \cdot \frac{\epsilon^2}{\min(\hat{p}_k^{\max}, 1 - \hat{p}_k^{\max}) + \epsilon} \right).$$

Noting that $\left| \Theta \backslash \widehat{\mathcal{P}}_k^\epsilon \right| \leq |\Theta|$ completes the proof of Lemma 5. $\qquad \square$

We are now ready for the proof of Theorem 3. We first note that, if $\bar{\eta}_{k,\delta} > 0$, it follows from Definition 2 that

$$\mathcal{P}_k^\eta \subseteq \Theta^\delta \text{ for any } \eta \in (0, \bar{\eta}_{k,\delta}).$$

Therefore, for any $\eta \in (0, \ \bar{\eta}_{k,\delta})$, we can write that

$$
\begin{aligned}
\mathbb{P}\left(\hat{\theta}_n \notin \Theta^\delta\right) \leq \mathbb{P}\left(\hat{\theta}_n \notin \mathcal{P}_k^\eta\right) \leq \ & \mathbb{P}\left(\left\{\hat{\theta}_n \notin \widehat{\mathcal{P}}_k^{\eta/2}\right\} \cup \left\{\widehat{\mathcal{P}}_k^{\eta/2} \not\subseteq \mathcal{P}_k^\eta\right\}\right) \\
\leq \ & \mathbb{P}\left(\hat{\theta}_n \notin \widehat{\mathcal{P}}_k^{\eta/2}\right) + \mathbb{P}\left(\widehat{\mathcal{P}}_k^{\eta/2} \not\subseteq \mathcal{P}_k^\eta\right).
\end{aligned}
\tag{34}
$$

We first evaluate the second probability on the right-hand side of (34). Lemma 4 gives that

$$
\begin{aligned}
\mathbb{P}\left(\widehat{\mathcal{P}}_k^{\eta/2} \not\subseteq \mathcal{P}_k^\eta\right) \leq |\Theta| \Bigg[ & \exp\left(-\frac{n}{2k} \cdot D_{\mathrm{KL}}\left(p_k^{\max} - \frac{3\eta}{4} \Big\| p_k^{\max} - \eta\right)\right) \\
& + \exp\left(-\frac{n}{2k} \cdot D_{\mathrm{KL}}\left(p_k^{\max} - \frac{\eta}{4} \Big\| p_k^{\max}\right)\right) \Bigg].
\end{aligned}
\tag{35}
$$

Next, by applying Lemma 5 with $\epsilon = \eta/2$, we can bound the first probability on the right-hand side of (34) as

$$
\mathbb{P}\left(\hat{\theta}_n \notin \widehat{\mathcal{P}}_k^{\eta/2}\right) \leq |\Theta| \cdot \mathbb{E}\left[\exp\left(-\frac{B}{24} \cdot \frac{\eta^2}{\min\left(\hat{p}_k(\theta_k^{\max}), 1 - \hat{p}_k(\theta_k^{\max})\right) + \eta/2}\right)\right].
\tag{36}
$$

Conditioned on the value of $\hat{p}_k(\theta_k^{\max})$, we can further upper-bound the right-hand side of (36) as follows

$$
\begin{aligned}
& \mathbb{E}\left[\exp\left(-\frac{B}{24} \cdot \frac{\eta^2}{\min\left(\hat{p}_k(\theta_k^{\max}), 1 - \hat{p}_k(\theta_k^{\max})\right) + \eta/2}\right)\right] \\
\leq \ & \mathbb{P}\left(\hat{p}_k(\theta_k^{\max}) \leq p_k^{\max} - \frac{\eta}{4}\right) \cdot \exp\left(-\frac{B}{24} \cdot \frac{\eta^2}{p_k^{\max} + \eta/4}\right) + \\
& \mathbb{P}\left(\left|\hat{p}_k(\theta_k^{\max}) - p_k^{\max}\right| < \frac{\eta}{4}\right) \cdot \exp\left(-\frac{B}{24} \cdot \frac{\eta^2}{\min\left(p_k^{\max}, 1 - p_k^{\max}\right) + 3\eta/4}\right) + \\
& \mathbb{P}\left(\hat{p}_k(\theta_k^{\max}) \geq p_k^{\max} + \frac{\eta}{4}\right) \cdot \exp\left(-\frac{B}{24} \cdot \frac{\eta^2}{1 - p_k^{\max} + \eta/4}\right) \\
\leq \ & \mathbb{P}\left(\hat{p}_k(\theta_k^{\max}) \leq p_k^{\max} - \frac{\eta}{4}\right) + \exp\left(-\frac{B}{24} \cdot \frac{\eta^2}{\min\left(p_k^{\max}, 1 - p_k^{\max}\right) + 3\eta/4}\right) + \\
& \mathbb{P}\left(\hat{p}_k(\theta_k^{\max}) \geq p_k^{\max} + \frac{\eta}{4}\right) \cdot \exp\left(-\frac{B}{24} \cdot \frac{\eta^2}{1 - p_k^{\max} + \eta/4}\right) \\
\overset{(i)}{\leq} \ & \exp\left(-\frac{n}{2k} \cdot D_{\mathrm{KL}}\left(p_k^{\max} - \frac{\eta}{4} \Big\| p_k^{\max}\right)\right) + \\
& \exp\left(-\frac{B}{24} \cdot \frac{\eta^2}{\min\left(p_k^{\max}, 1 - p_k^{\max}\right) + 3\eta/4}\right) + \\
& \mathbb{1}\left(p_k^{\max} + \frac{\eta}{4} \leq 1\right) \cdot \exp\left(-\frac{n}{2k} \cdot D_{\mathrm{KL}}\left(p_k^{\max} + \frac{\eta}{4} \Big\| p_k^{\max}\right)\right) \cdot \exp\left(-\frac{B}{24} \cdot \frac{\eta^2}{1 - p_k^{\max} + \eta/4}\right)
\end{aligned}
$$

where inequality $(i)$ results from applying Lemma 2 with $\hat{p}_k(\theta_k^{\max})$, the U-statistic estimate for $p_k^{\max}$. Together, the above equations imply that

$$
\begin{aligned}
& \mathbb{P}\left(\hat{\theta}_n \notin \Theta^\delta\right) \\
\leq \ & |\Theta| \Bigg[ \exp\left(-\frac{n}{2k} \cdot D_{\mathrm{KL}}\left(p_k^{\max} - \frac{3\eta}{4} \Big\| p_k^{\max} - \eta\right)\right) + \\
& 2\exp\left(-\frac{n}{2k} \cdot D_{\mathrm{KL}}\left(p_k^{\max} - \frac{\eta}{4} \Big\| p_k^{\max}\right)\right) + \\
& \exp\left(-\frac{B}{24} \cdot \frac{\eta^2}{\min\left(p_k^{\max}, 1 - p_k^{\max}\right) + 3\eta/4}\right) + \\
& \mathbb{1}\left(p_k^{\max} + \frac{\eta}{4} \leq 1\right) \cdot \exp\left(-\frac{n}{2k} \cdot D_{\mathrm{KL}}\left(p_k^{\max} + \frac{\eta}{4} \Big\| p_k^{\max}\right) - \frac{B}{24} \cdot \frac{\eta^2}{1 - p_k^{\max} + \eta/4}\right) \Bigg].
\end{aligned}
$$

Since the above probability bound is left-continuous in $\eta$ and $\eta$ can be arbitrarily chosen from $(0, \bar{\eta}_{k,\delta})$, the validity of the case $\eta = \bar{\eta}_{k,\delta}$ follows from pushing $\eta$ to the limit $\bar{\eta}_{k,\delta}$. This gives (30).

To simplify the bound in the case $\bar{\eta}_{k,\delta} > 4/5$. Consider the bound (30) with $\eta = \bar{\eta}_{k,\delta}$. Since $p_k^{\max} \geq \bar{\eta}_{k,\delta}$ by the definition of $\bar{\eta}_{k,\delta}$, it must hold that $p_k^{\max} + \bar{\eta}_{k,\delta}/4 > 4/5 + 1/5 = 1$, therefore the last term in the finite-sample bound (30) vanishes. To simplify the first two terms in the finite-sample bound, we note that

$$p_k^{\max} - \frac{3\bar{\eta}_{k,\delta}}{4} \leq 1 - \frac{3}{4} \cdot \frac{4}{5} = \frac{2}{5},$$

$$p_k^{\max} - \frac{3\bar{\eta}_{k,\delta}}{4} \geq \bar{\eta}_{k,\delta} - \frac{3\bar{\eta}_{k,\delta}}{4} \geq \frac{1}{5},$$

$$p_k^{\max} - \frac{\bar{\eta}_{k,\delta}}{4} \leq 1 - \frac{1}{4} \cdot \frac{4}{5} = \frac{4}{5},$$

$$p_k^{\max} - \frac{\bar{\eta}_{k,\delta}}{4} \geq \bar{\eta}_{k,\delta} - \frac{\bar{\eta}_{k,\delta}}{4} \geq \frac{3}{5},$$

and that $p_k^{\max} - \bar{\eta}_{k,\delta} \leq 1 - \bar{\eta}_{k,\delta} \leq 1/5$, therefore by the bound (18) from Lemma 3, we can bound the first two terms as

$$\exp\left(-\frac{n}{2k} \cdot D_{\mathrm{KL}}\left(p_k^{\max} - \frac{3\bar{\eta}_{k,\delta}}{4} \middle\| p_k^{\max} - \bar{\eta}_{k,\delta}\right)\right)$$

$$\leq \exp\left(\frac{n}{2k} \ln\left(2((p_k^{\max} - \bar{\eta}_{k,\delta})(1 - p_k^{\max} + \bar{\eta}_{k,\delta}))^{1/5}\right)\right)$$

$$= \left(2((p_k^{\max} - \bar{\eta}_{k,\delta})(1 - p_k^{\max} + \bar{\eta}_{k,\delta}))^{1/5}\right)^{n/(2k)}$$

$$\leq \left(2(p_k^{\max} - \bar{\eta}_{k,\delta})^{1/5}\right)^{n/(2k)}$$

$$= \left(2^5(p_k^{\max} - \bar{\eta}_{k,\delta})\right)^{n/(10k)},$$

and similarly

$$\exp\left(-\frac{n}{2k} \cdot D_{\mathrm{KL}}\left(p_k^{\max} - \frac{\bar{\eta}_{k,\delta}}{4} \middle\| p_k^{\max}\right)\right)$$

$$\leq \exp\left(\frac{n}{2k} \ln\left(2(p_k^{\max}(1 - p_k^{\max}))^{1/5}\right)\right)$$

$$= \left(2(p_k^{\max}(1 - p_k^{\max}))^{1/5}\right)^{n/(2k)}$$

$$\leq \left(2(1 - p_k^{\max})^{1/5}\right)^{n/(2k)}$$

$$= \left(2^5(1 - p_k^{\max})\right)^{n/(10k)}.$$

On the other hand, by Lemma 3 both $D_{\mathrm{KL}}\left(p_k^{\max} - 3\bar{\eta}_{k,\delta}/4 \| p_k^{\max} - \bar{\eta}_{k,\delta}\right)$ and $D_{\mathrm{KL}}\left(p_k^{\max} - \bar{\eta}_{k,\delta}/4 \| p_k^{\max}\right)$ are bounded below by $\bar{\eta}_{k,\delta}^2/8$, therefore

$$\exp\left(-\frac{n}{2k} \cdot D_{\mathrm{KL}}\left(p_k^{\max} - \frac{3\bar{\eta}_{k,\delta}}{4} \middle\| p_k^{\max} - \bar{\eta}_{k,\delta}\right)\right) \leq \exp\left(-\frac{n}{2k} \cdot \frac{\bar{\eta}_{k,\delta}^2}{8}\right) \leq \exp\left(-\frac{n}{25k}\right),$$

and the same holds for $\exp\left(-n/(2k) \cdot D_{\mathrm{KL}}\left(p_k^{\max} - \bar{\eta}_{k,\delta}/4 \| p_k^{\max}\right)\right)$. For the third term in the bound (30) we have

$$\frac{\bar{\eta}_{k,\delta}^2}{\min(p_k^{\max}, 1 - p_k^{\max}) + 3\bar{\eta}_{k,\delta}/4} \geq \frac{(4/5)^2}{\min(1, 1/5) + 3/4} \geq \frac{16}{25},$$

and hence

$$\exp\left(-\frac{B}{24} \cdot \frac{\bar{\eta}_{k,\delta}^2}{\min(p_k^{\max}, 1 - p_k^{\max}) + 3\bar{\eta}_{k,\delta}/4}\right) \leq \exp\left(-\frac{B}{75/2}\right).$$

The first desired bound then follows by setting $C_1, C_2, C_3$ to be the appropriate constants. This completes the proof of Theorem 3. □

*Proof of Theorem 1.* Algorithm 1 is a special case of Algorithm 3 with the learning algorithm (23) that outputs a singleton, therefore the results of Theorem 3 automatically apply. Since $\mathcal{E}_{k,\delta} = \sum_{\theta \in \Theta \setminus \Theta^\delta} p_k(\theta) \geq \max_{\theta \in \Theta \setminus \Theta^\delta} p_k(\theta)$, it holds that $\eta_{k,\delta} \leq \bar{\eta}_{k,\delta}$. When $\eta_{k,\delta} > 0$ we also have $\bar{\eta}_{k,\delta} > 0$, therefore (8) follows from setting $\eta$ to be $\eta_{k,\delta}$ in (30), and (9) follows from upper bounding $\max_{\theta \in \Theta \setminus \Theta^\delta} p_k(\theta)$ with $\mathcal{E}_{k,\delta}$ in (31). $\qquad\square$

### C.3 PROOF FOR THEOREM 2

We first present two lemmas to be used in the main proof. The first lemma characterizes the exponentially improving quality of the solution set retrieved in Phase I:

**Lemma 6 (Quality of retrieved solutions in Algorithm 2)** *For every $k$ and $\delta \geq 0$, the set of retrieved solutions $\mathcal{S}$ from Phase I of Algorithm 2 with $k_1 = k$ and without data splitting satisfies that*

$$\mathbb{P}\left(\mathcal{S} \cap \Theta^\delta = \emptyset\right) \leq \min\left(e^{-(1-\mathcal{E}_{k,\delta})/C_4}, C_5 \mathcal{E}_{k,\delta}\right)^{\frac{n}{C_6 k}} + \exp\left(-\frac{B_1}{C_7}(1-\mathcal{E}_{k,\delta})\right), \qquad (37)$$

*where $C_4, C_5, C_6, C_7 > 0$ are universal constants. The same bound with $n$ replaced by $n/2$ holds true for Algorithm 2 with data splitting.*

*Proof of Lemma 6.* Let $(z_1^*, \ldots, z_k^*)$ be a random subsample and $\mathbb{P}_*$ be the probability with respect to the subsampling randomness conditioned on the data and the algorithmic randomness. Consider the two probabilities

$$\mathbb{P}\left(\mathcal{A}(z_1, \ldots, z_k) \in \Theta^\delta\right), \ \mathbb{P}_*\left(\mathcal{A}(z_1^*, \ldots, z_k^*) \in \Theta^\delta\right).$$

We have $1 - \mathcal{E}_{k,\delta} = \mathbb{P}\left(\mathcal{A}(z_1, \ldots, z_k) \in \Theta^\delta\right)$, and the conditional probability

$$\mathbb{P}\left(\mathcal{S} \cap \Theta^\delta = \emptyset \Big| \mathbb{P}_*\left(\mathcal{A}(z_1^*, \ldots, z_k^*) \in \Theta^\delta\right)\right) = \left(1 - \mathbb{P}_*\left(\mathcal{A}(z_1^*, \ldots, z_k^*) \in \Theta^\delta\right)\right)^{B_1}.$$

Therefore we can write

$$
\begin{aligned}
\mathbb{P}\left(\mathcal{S} \cap \Theta^\delta = \emptyset\right) &= \mathbb{E}\left[\left(1 - \mathbb{P}_*\left(\mathcal{A}(z_1^*, \ldots, z_k^*) \in \Theta^\delta\right)\right)^{B_1}\right] \\
&\leq \mathbb{P}\left(\mathbb{P}_*\left(\mathcal{A}(z_1^*, \ldots, z_k^*) \in \Theta^\delta\right) < \frac{1 - \mathcal{E}_{k,\delta}}{e}\right) + \left(1 - \frac{1 - \mathcal{E}_{k,\delta}}{e}\right)^{B_1} \quad (38)
\end{aligned}
$$

where $e$ is the base of the natural logarithm. Applying Lemma 2 with $\kappa(z_1, \ldots, z_k; \omega) := \mathbb{1}\left(\mathcal{A}(z_1, \ldots, z_k; \omega) \in \Theta^\delta\right)$ gives

$$\mathbb{P}\left(\mathbb{P}_*\left(\mathcal{A}(z_1^*, \ldots, z_k^*) \in \Theta^\delta\right) < \frac{1 - \mathcal{E}_{k,\delta}}{e}\right) \leq \exp\left(-\frac{n}{2k} \cdot D_{\mathrm{KL}}\left(\frac{1 - \mathcal{E}_{k,\delta}}{e}\Big\|1 - \mathcal{E}_{k,\delta}\right)\right).$$

Further applying the bound (17) from Lemma 3 to the KL divergence on the right-hand side leads to

$$D_{\mathrm{KL}}\left(\frac{1 - \mathcal{E}_{k,\delta}}{e}\Big\|1 - \mathcal{E}_{k,\delta}\right) \geq \frac{1 - \mathcal{E}_{k,\delta}}{e}\ln\frac{1}{e} + 1 - \mathcal{E}_{k,\delta} - \frac{1 - \mathcal{E}_{k,\delta}}{e} = \left(1 - \frac{2}{e}\right)(1 - \mathcal{E}_{k,\delta})$$

and

$$
\begin{aligned}
&D_{\mathrm{KL}}\left(\frac{1 - \mathcal{E}_{k,\delta}}{e}\Big\|1 - \mathcal{E}_{k,\delta}\right) \\
&= D_{\mathrm{KL}}\left(1 - \frac{1 - \mathcal{E}_{k,\delta}}{e}\Big\|\mathcal{E}_{k,\delta}\right) \\
&\geq \left(1 - \frac{1 - \mathcal{E}_{k,\delta}}{e}\right)\ln\frac{1 - (1 - \mathcal{E}_{k,\delta})/e}{\mathcal{E}_{k,\delta}} - (1 - \mathcal{E}_{k,\delta}) + \frac{1 - \mathcal{E}_{k,\delta}}{e} \quad \text{by bound (17)} \\
&\geq \left(1 - \frac{1 - \mathcal{E}_{k,\delta}}{e}\right)\ln\left(1 - \frac{1 - \mathcal{E}_{k,\delta}}{e}\right) - \left(1 - \frac{1}{e}\right)\ln\mathcal{E}_{k,\delta} - 1 + \frac{1}{e} \\
&\geq \left(1 - \frac{1}{e}\right)\ln\left(1 - \frac{1}{e}\right) - \left(1 - \frac{1}{e}\right)\ln\mathcal{E}_{k,\delta} - 1 + \frac{1}{e} \\
&= \left(1 - \frac{1}{e}\right)\ln\frac{e-1}{e^2\mathcal{E}_{k,\delta}}.
\end{aligned}
$$

Combining the two bounds for the KL divergence we have

$$\mathbb{P}\left(\mathbb{P}_*\left(\mathcal{A}(z_1^*, \ldots, z_k^*) \in \Theta^\delta\right) < \frac{1 - \mathcal{E}_{k,\delta}}{e}\right)$$

$$\leq \quad \min\left(\exp\left(-\frac{n}{2k} \cdot \left(1 - \frac{2}{e}\right)(1 - \mathcal{E}_{k,\delta})\right), \left(\frac{e^2 \mathcal{E}_{k,\delta}}{e-1}\right)^{(1-1/e)\frac{n}{2k}}\right).$$

Note that the second term on the right-hand side of (38) satisfies that $(1 - (1 - \mathcal{E}_{k,\delta})/e)^{B_1} \leq \exp\left(-B_1(1 - \mathcal{E}_{k,\delta})/e\right)$. Thus, we derive that

$$\mathbb{P}\left(\mathcal{S} \cap \Theta^\delta = \emptyset\right)$$

$$\leq \min\left(\exp\left(-\frac{n}{2k} \cdot \left(1 - \frac{2}{e}\right)(1 - \mathcal{E}_{k,\delta})\right), \left(\frac{e^2 \mathcal{E}_{k,\delta}}{e-1}\right)^{(1-1/e)\frac{n}{2k}}\right) + \exp\left(-\frac{B_1(1 - \mathcal{E}_{k,\delta})}{e}\right)$$

$$\leq \min\left(\exp\left(-\frac{1 - 2/e}{1 - 1/e} \cdot (1 - \mathcal{E}_{k,\delta})\right), \frac{e^2 \mathcal{E}_{k,\delta}}{e-1}\right)^{(1-1/e)\frac{n}{2k}} + \exp\left(-\frac{B_1(1 - \mathcal{E}_{k,\delta})}{e}\right).$$

The conclusion then follows by setting $C_4, C_5, C_6, C_7$ to be the appropriate constants. $\qquad\square$

The second lemma gives bounds for the excess risk sensitivity $\bar{\eta}_{k,\delta}$ in the case of the set-valued learning algorithm (24):

**Lemma 7 (Bounds of $\bar{\eta}_{k,\delta}$ for the set-valued learning algorithm (24))** *Consider discrete decision space $\Theta$. If the set-valued learning algorithm*

$$\mathbb{A}(z_1, \ldots, z_k; \omega) := \left\{\theta \in \Theta : \frac{1}{k}\sum_{i=1}^{k} l(\theta, z_i) \leq \min_{\theta' \in \Theta} \frac{1}{k}\sum_{i=1}^{k} l(\theta', z_i) + \epsilon\right\}$$

*is used with $\epsilon \geq 0$, it holds that*

$$p_k^{\max} = \max_{\theta \in \Theta} p_k(\theta) \geq 1 - T_k\left(\frac{\epsilon}{2}\right), \tag{39}$$

$$\max_{\theta \in \Theta \backslash \Theta^\delta} p_k(\theta) \leq T_k\left(\frac{\delta - \epsilon}{2}\right), \tag{40}$$

*and hence*

$$\bar{\eta}_{k,\delta} \geq 1 - T_k\left(\frac{\epsilon}{2}\right) - T_k\left(\frac{\delta - \epsilon}{2}\right), \tag{41}$$

*where $T_k$ is the tail probability defined in Theorem 2.*

*Proof of Lemma 7.* Let $\hat{L}_k(\theta) := \frac{1}{k}\sum_{i=1}^{k} l(\theta, z_i)$. Let $\theta^*$ be an optimal solution of (1). We have

$$\max_{\theta \in \Theta} p_k(\theta) \geq p_k(\theta^*) = \mathbb{P}\left(\theta^* \in \widehat{\Theta}_k^\epsilon\right) \geq \mathbb{P}\left(\Theta^0 \subseteq \widehat{\Theta}_k^\epsilon\right).$$

To bound the probability on the right hand side, we write

$$\left\{\Theta^0 \not\subseteq \widehat{\Theta}_k^\epsilon\right\} \quad \subseteq \quad \bigcup_{\theta \in \Theta^0, \theta' \in \Theta} \left\{\hat{L}_k(\theta) > \hat{L}_k(\theta') + \epsilon\right\}$$

$$= \quad \bigcup_{\theta \in \Theta^0, \theta' \in \Theta} \left\{\hat{L}_k(\theta) - L(\theta) > \hat{L}_k(\theta') - L(\theta') + L(\theta') - L(\theta) + \epsilon\right\}$$

$$\subseteq \quad \bigcup_{\theta \in \Theta^0, \theta' \in \Theta} \left\{\hat{L}_k(\theta) - L(\theta) > \hat{L}_k(\theta') - L(\theta') + \epsilon\right\}$$

$$\subseteq \quad \bigcup_{\theta \in \Theta^0, \theta' \in \Theta} \left\{\hat{L}_k(\theta) - L(\theta) > \frac{\epsilon}{2} \text{ or } \hat{L}_k(\theta') - L(\theta') < -\frac{\epsilon}{2}\right\}$$

$$\subseteq \quad \bigcup_{\theta \in \Theta} \left\{\left|\hat{L}_k(\theta) - L(\theta)\right| > \frac{\epsilon}{2}\right\}$$

$$= \quad \left\{\max_{\theta \in \Theta}\left|\hat{L}_k(\theta) - L(\theta)\right| > \frac{\epsilon}{2}\right\},$$

therefore

$$\max_{\theta \in \Theta} p_k(\theta) \geq \mathbb{P}\left(\max_{\theta \in \Theta}\left|\hat{L}_k(\theta) - L(\theta)\right| \leq \frac{\epsilon}{2}\right) \geq 1 - T_k\left(\frac{\epsilon}{2}\right). \tag{42}$$

This proves (39). To bound the other term $\max_{\theta \in \Theta \setminus \Theta^\delta} p_k(\theta)$, for any $\theta \in \Theta \setminus \Theta^\delta$ it holds that

$$p_k(\theta) = \mathbb{P}\left(\theta \in \widehat{\Theta}_k^\epsilon\right) \leq \mathbb{P}\left(\widehat{\Theta}_k^\epsilon \not\subseteq \Theta^\delta\right), \tag{43}$$

and hence $\max_{\theta \in \Theta \setminus \Theta^\delta} p_k(\theta) \leq \mathbb{P}\left(\widehat{\Theta}_k^\epsilon \not\subseteq \Theta^\delta\right)$. To bound the latter, we have

$$
\begin{aligned}
\left\{\widehat{\Theta}_k^\epsilon \not\subseteq \Theta^\delta\right\} \quad &\subseteq \quad \bigcup_{\theta,\theta' \in \Theta \text{ s.t. } L(\theta')-L(\theta)>\delta} \left\{\hat{L}_k(\theta') \leq \hat{L}_k(\theta) + \epsilon\right\} \\
&= \quad \bigcup_{\theta,\theta' \in \Theta \text{ s.t. } L(\theta')-L(\theta)>\delta} \left\{\hat{L}_k(\theta') - L(\theta') + L(\theta') - L(\theta) \leq \hat{L}_k(\theta) - L(\theta) + \epsilon\right\} \\
&\subseteq \quad \bigcup_{\theta,\theta' \in \Theta \text{ s.t. } L(\theta')-L(\theta)>\delta} \left\{\hat{L}_k(\theta') - L(\theta') + \delta < \hat{L}_k(\theta) - L(\theta) + \epsilon\right\} \\
&\subseteq \quad \bigcup_{\theta,\theta' \in \Theta \text{ s.t. } L(\theta')-L(\theta)>\delta} \left\{\hat{L}_k(\theta') - L(\theta') < -\frac{\delta - \epsilon}{2} \text{ or } \hat{L}_k(\theta) - L(\theta) > \frac{\delta - \epsilon}{2}\right\} \\
&\subseteq \quad \bigcup_{\theta \in \Theta} \left\{\left|\hat{L}_k(\theta) - L(\theta)\right| > \frac{\delta - \epsilon}{2}\right\} \\
&= \quad \left\{\max_{\theta \in \Theta}\left|\hat{L}_k(\theta) - L(\theta)\right| > \frac{\delta - \epsilon}{2}\right\},
\end{aligned}
$$

therefore

$$\max_{\theta \in \Theta \setminus \Theta^\delta} p_k(\theta) \leq \mathbb{P}\left(\max_{\theta \in \Theta}\left|\hat{L}_k(\theta) - L(\theta)\right| > \frac{\delta - \epsilon}{2}\right) \leq T_k\left(\frac{\delta - \epsilon}{2}\right). \tag{44}$$

This immediately gives (40). (41) is obvious given (39) and (40). $\qquad\square$

To prove Theorem 2, we introduce some notation. For every non-empty subset $\mathcal{W} \subseteq \Theta$, we use the following counterpart of Definition 3. Let

$$\mathcal{W}^\delta := \left\{\theta \in \mathcal{W} : L(\theta) \leq \min_{\theta' \in \mathcal{W}} L(\theta') + \delta\right\} \tag{45}$$

be the set of $\delta$-optimal solutions in the restricted decision space $\mathcal{W}$, and

$$\widehat{\mathcal{W}}_k^\delta := \left\{\theta \in \mathcal{W} : \frac{1}{k}\sum_{i=1}^k l(\theta, z_i) = \min_{\theta' \in \mathcal{W}} \frac{1}{k}\sum_{i=1}^k l(\theta', z_i) + \delta\right\} \tag{46}$$

be the set of $\delta$-optimal solutions of the SAA with an i.i.d. data set of size $k$.

*Proof of Theorem 2 for* ROVEs. Given the retrieved solution set $\mathcal{S}$ and the chosen $\epsilon$, the rest of Phase II of Algorithm 2 exactly performs Algorithm 3 on the restricted problem $\min_{\theta \in \mathcal{S}} \mathbb{E}\left[l(\theta, z)\right]$ to obtain $\hat{\theta}_n$ with the data $z_{\lfloor n/2 \rfloor + 1:n}$, the set-valued learning algorithm (24), the chosen $\epsilon$ value and $k = k_2, B = B_2$.

To show the upper bound for the unconditional convergence probability $\mathbb{P}\left(\hat{\theta}_n \notin \Theta^{2\delta}\right)$, note that

$$\left\{\mathcal{S} \cap \Theta^\delta \neq \emptyset\right\} \cap \left\{L(\hat{\theta}_n) \leq \min_{\theta \in \mathcal{S}} L(\theta) + \delta\right\} \subseteq \left\{\hat{\theta}_n \in \Theta^{2\delta}\right\},$$

and hence by union bound we can write

$$\mathbb{P}\left(\hat{\theta}_n \notin \Theta^{2\delta}\right) \leq \mathbb{P}\left(\mathcal{S} \cap \Theta^\delta = \emptyset\right) + \mathbb{P}\left(L(\hat{\theta}_n) > \min_{\theta \in \mathcal{S}} L(\theta) + \delta\right). \tag{47}$$

$\mathbb{P}\left(\mathcal{S} \cap \Theta^{\delta} = \emptyset\right)$ has a bound from Lemma 6. We focus on the second probability.

For a fixed retrieved subset $\mathcal{S} \subseteq \Theta$, define the tail of the maximum deviation on $\mathcal{S}$

$$T_k^{\mathcal{S}}(\cdot) := \mathbb{P}\left(\sup_{\theta \in \mathcal{S}} \left|\frac{1}{k} \sum_{i=1}^{k} l(\theta, z_i) - L(\theta)\right| > \cdot\right).$$

It is straightforward that $T_k^{\mathcal{S}}(\cdot) \leq T_k(\cdot)$ where $T_k$ is the tail of the maximum deviation over the whole space $\Theta$. Since $\mathbb{P}\left(\epsilon \in [\underline{\epsilon}, \bar{\epsilon}]\right) = 1$, we have

$$1 - T_{k_2}^{\mathcal{S}}\left(\frac{\epsilon}{2}\right) - T_{k_2}^{\mathcal{S}}\left(\frac{\delta - \epsilon}{2}\right) \geq 1 - T_{k_2}^{\mathcal{S}}\left(\frac{\underline{\epsilon}}{2}\right) - T_{k_2}^{\mathcal{S}}\left(\frac{\delta - \bar{\epsilon}}{2}\right).$$

If $T_{k_2}\left((\delta - \bar{\epsilon})/2\right) + T_{k_2}\left(\underline{\epsilon}/2\right) < 1/5$, we have $T_{k_2}^{\mathcal{S}}\left((\delta - \bar{\epsilon})/2\right) + T_{k_2}^{\mathcal{S}}\left(\underline{\epsilon}/2\right) < 1/5$ and subsequently $1 - T_{k_2}^{\mathcal{S}}\left((\delta - \epsilon)/2\right) - T_{k_2}^{\mathcal{S}}\left(\epsilon/2\right) > 4/5$, and hence $\bar{\eta}_{k_2,\eta} \geq 1 - T_{k_2}^{\mathcal{S}}\left((\delta - \epsilon)/2\right) - T_{k_2}^{\mathcal{S}}\left(\epsilon/2\right) > 4/5$ by Lemma 7 for Phase II of ROVEs conditioned on $\mathcal{S}$ and $\epsilon$, therefore the bound (31) from Theorem 3 applies. Using the inequalities (39) and (40) to upper bound the $\min(1 - p_k^{\max}, p_k^{\max} - \bar{\eta}_{k,\delta})$ term in (31) gives

$$\mathbb{P}\left(L(\hat{\theta}_n) > \min_{\theta \in \mathcal{S}} L(\theta) + \delta \big| \mathcal{S}, \epsilon\right)$$

$$\leq |\mathcal{S}|\left(3 \min\left(e^{-2/5}, C_1 \max\left(T_{k_2}^{\mathcal{S}}\left(\frac{\epsilon}{2}\right), T_{k_2}^{\mathcal{S}}\left(\frac{\delta - \bar{\epsilon}}{2}\right)\right)\right)^{\frac{n}{2C_2 k_2}} + \exp\left(-\frac{B_2}{C_3}\right)\right)$$

$$= |\mathcal{S}|\left(3 \min\left(e^{-2/5}, C_1 T_{k_2}^{\mathcal{S}}\left(\frac{\min(\underline{\epsilon}, \delta - \bar{\epsilon})}{2}\right)\right)^{\frac{n}{2C_2 k_2}} + \exp\left(-\frac{B_2}{C_3}\right)\right)$$

$$\leq |\mathcal{S}|\left(3 \min\left(e^{-2/5}, C_1 T_{k_2}\left(\frac{\min(\underline{\epsilon}, \delta - \bar{\epsilon})}{2}\right)\right)^{\frac{n}{2C_2 k_2}} + \exp\left(-\frac{B_2}{C_3}\right)\right).$$

Further relaxing $|\mathcal{S}|$ to $B_1$ and taking full expectation on both sides give

$$\mathbb{P}\left(L(\hat{\theta}_n) > \min_{\theta \in \mathcal{S}} L(\theta) + \delta\right) \leq B_1\left(3 \min\left(e^{-2/5}, C_1 T_{k_2}\left(\frac{\min(\underline{\epsilon}, \delta - \bar{\epsilon})}{2}\right)\right)^{\frac{n}{2C_2 k_2}} + \exp\left(-\frac{B_2}{C_3}\right)\right).$$

This leads to the desired bound (10) after the above bound is plugged into (47) and the bound (37) from Lemma 6 is applied with $k = k_1$. $\qquad\square$

*Proof of Theorem 2 for* ROVE. For every non-empty subset $\mathcal{W} \subseteq \Theta$ and $k_2$, we consider the indicator

$$\mathbb{1}_{k_2}^{\theta, \mathcal{W}, \epsilon}(z_1, \ldots, z_{k_2}) := \mathbb{1}\left(\frac{1}{k_2}\sum_{i=1}^{k_2} l(\theta, z_i) \leq \min_{\theta' \in \mathcal{W}} \frac{1}{k_2}\sum_{i=1}^{k_2} l(\theta', z_i) + \epsilon\right) \quad \text{for } \theta \in \mathcal{W}, \epsilon \in [0, \delta/2],$$

which indicates whether a solution $\theta \in \mathcal{W}$ is $\epsilon$-optimal for the SAA formed by $\{z_1, \ldots, z_{k_2}\}$. Here we add $\epsilon$ and $\mathcal{W}$ to the superscript to emphasize its dependence on them. The counterparts of the solution probabilities $p_k, \hat{p}_k, \bar{p}_k$ for $\mathbb{1}_{k_2}^{\theta, \mathcal{W}, \epsilon}$ are

$$p_{k_2}^{\mathcal{W}, \epsilon}(\theta) := \mathbb{E}\left[\mathbb{1}_{k_2}^{\theta, \mathcal{W}, \epsilon}(z_1, \ldots, z_{k_2})\right],$$

$$\hat{p}_{k_2}^{\mathcal{W}, \epsilon}(\theta) := \mathbb{E}_*\left[\mathbb{1}_{k_2}^{\theta, \mathcal{W}, \epsilon}(z_1^*, \ldots, z_{k_2}^*)\right],$$

$$\bar{p}_{k_2}^{\mathcal{W}, \epsilon}(\theta) := \frac{1}{B_2}\sum_{b=1}^{B_2} \mathbb{1}_{k_2}^{\theta, \mathcal{W}, \epsilon}(z_1^b, \ldots, z_{k_2}^b).$$

We need to show the uniform convergence of these probabilities for $\epsilon \in [0, \delta/2]$. To do so, we define a slighted modified version of $\mathbb{1}_{k_2}^{\theta, \mathcal{W}, \epsilon}$

$$\mathbb{1}_{k_2}^{\theta, \mathcal{W}, \epsilon-}(z_1, \ldots, z_{k_2}) := \mathbb{1}\left(\frac{1}{k_2}\sum_{i=1}^{k_2} l(\theta, z_i) < \min_{\theta' \in \mathcal{W}} \frac{1}{k_2}\sum_{i=1}^{k_2} l(\theta', z_i) + \epsilon\right) \quad \text{for } \theta \in \mathcal{W}, \epsilon \in [0, \delta/2],$$

which indicates a strict $\epsilon$-optimal solution, and let $p_{k_2}^{\mathcal{W},\epsilon-}, \hat{p}_{k_2}^{\mathcal{W},\epsilon-}, \bar{p}_{k_2}^{\mathcal{W},\epsilon-}$ be the corresponding counterparts of solution probabilities. For any integer $m > 1$ we construct brackets of size at most $1/m$ to cover the family of indicator functions $\{\mathbb{1}_{k_2}^{\theta,\mathcal{W},\epsilon} : \epsilon \in [0, \delta/2]\}$, i.e., let $m' = \lfloor p_{k_2}^{\mathcal{W},\delta/2}(\theta)m \rfloor$ and

$$
\begin{aligned}
\epsilon_0 &:= 0, \\
\epsilon_i &:= \inf\left\{\epsilon \in [0, \delta/2] : p_{k_2}^{\mathcal{W},\epsilon}(\theta) \geq i/m\right\} \quad \text{for } 1 \leq i \leq m', \\
\epsilon_{m'+1} &:= \frac{\delta}{2},
\end{aligned}
$$

where we assume that $\epsilon_i, i = 0, \dots, m'+1$ are strictly increasing without loss of generality (otherwise we can delete duplicated values). Then for any $\epsilon \in [\epsilon_i, \epsilon_{i+1})$, we have that

$$
\begin{aligned}
\bar{p}_{k_2}^{\mathcal{W},\epsilon}(\theta) - p_{k_2}^{\mathcal{W},\epsilon}(\theta) &\leq \bar{p}_{k_2}^{\mathcal{W},\epsilon_{i+1}-}(\theta) - p_{k_2}^{\mathcal{W},\epsilon_i}(\theta) \\
&\leq \bar{p}_{k_2}^{\mathcal{W},\epsilon_{i+1}-}(\theta) - p_{k_2}^{\mathcal{W},\epsilon_{i+1}-}(\theta) + p_{k_2}^{\mathcal{W},\epsilon_{i+1}-}(\theta) - p_{k_2}^{\mathcal{W},\epsilon_i}(\theta) \\
&\leq \bar{p}_{k_2}^{\mathcal{W},\epsilon_{i+1}-}(\theta) - p_{k_2}^{\mathcal{W},\epsilon_{i+1}-}(\theta) + \frac{1}{m}
\end{aligned}
$$

and that

$$
\begin{aligned}
\bar{p}_{k_2}^{\mathcal{W},\epsilon}(\theta) - p_{k_2}^{\mathcal{W},\epsilon}(\theta) &\geq \bar{p}_{k_2}^{\mathcal{W},\epsilon_i}(\theta) - p_{k_2}^{\mathcal{W},\epsilon_{i+1}-}(\theta) \\
&\geq \bar{p}_{k_2}^{\mathcal{W},\epsilon_i}(\theta) - p_{k_2}^{\mathcal{W},\epsilon_i}(\theta) + p_{k_2}^{\mathcal{W},\epsilon_i}(\theta) - p_{k_2}^{\mathcal{W},\epsilon_{i+1}-}(\theta) \\
&\geq \bar{p}_{k_2}^{\mathcal{W},\epsilon_i}(\theta) - p_{k_2}^{\mathcal{W},\epsilon_i}(\theta) - \frac{1}{m}.
\end{aligned}
$$

Therefore

$$
\begin{aligned}
&\sup_{\epsilon \in [0,\delta/2]} \left| \bar{p}_{k_2}^{\mathcal{W},\epsilon}(\theta) - p_{k_2}^{\mathcal{W},\epsilon}(\theta) \right| \\
&\leq \max_{0 \leq i \leq m'+1} \max\left( \left| \bar{p}_{k_2}^{\mathcal{W},\epsilon_i}(\theta) - p_{k_2}^{\mathcal{W},\epsilon_i}(\theta) \right|, \left| \bar{p}_{k_2}^{\mathcal{W},\epsilon_i-}(\theta) - p_{k_2}^{\mathcal{W},\epsilon_i-}(\theta) \right| \right) + \frac{1}{m}. \quad (48)
\end{aligned}
$$

To show that the random variable in (48) converges to 0 in probability, we note that the U-statistic has the minimum variance among all unbiased estimators, in particular the following simple sample average estimators based on the first $\lfloor n/k_2 \rfloor \cdot k_2$ data

$$
\begin{aligned}
\tilde{p}_{k_2}^{\mathcal{W},\epsilon}(\theta) &:= \frac{1}{\lfloor n/k_2 \rfloor} \sum_{i=1}^{\lfloor n/k_2 \rfloor} \mathbb{1}_{k_2}^{\theta,\mathcal{W},\epsilon}(z_{k_2(i-1)+1}, \dots, z_{k_2 i}), \\
\tilde{p}_{k_2}^{\mathcal{W},\epsilon-}(\theta) &:= \frac{1}{\lfloor n/k_2 \rfloor} \sum_{i=1}^{\lfloor n/k_2 \rfloor} \mathbb{1}_{k_2}^{\theta,\mathcal{W},\epsilon-}(z_{k_2(i-1)+1}, \dots, z_{k_2 i}).
\end{aligned}
$$

Therefore we can write

$$
\mathbb{E}\left[\left(\max_{0 \le i \le m'+1} \max\left(\left|\bar{p}_{k_2}^{\mathcal{W},\epsilon_i}(\theta) - p_{k_2}^{\mathcal{W},\epsilon_i}(\theta)\right|, \left|\bar{p}_{k_2}^{\mathcal{W},\epsilon_i-}(\theta) - p_{k_2}^{\mathcal{W},\epsilon_i-}(\theta)\right|\right)\right)^2\right]
$$

$$
\le \sum_{0 \le i \le m'+1} \left(\mathbb{E}\left[\left(\bar{p}_{k_2}^{\mathcal{W},\epsilon_i}(\theta) - p_{k_2}^{\mathcal{W},\epsilon_i}(\theta)\right)^2\right] + \mathbb{E}\left[\left(\bar{p}_{k_2}^{\mathcal{W},\epsilon_i-}(\theta) - p_{k_2}^{\mathcal{W},\epsilon_i-}(\theta)\right)^2\right]\right)
$$

$$
\le \sum_{0 \le i \le m'+1} \left(\mathbb{E}\left[\left(\bar{p}_{k_2}^{\mathcal{W},\epsilon_i}(\theta) - \hat{p}_{k_2}^{\mathcal{W},\epsilon_i}(\theta)\right)^2\right] + \mathbb{E}\left[\left(\hat{p}_{k_2}^{\mathcal{W},\epsilon_i}(\theta) - p_{k_2}^{\mathcal{W},\epsilon_i}(\theta)\right)^2\right]\right) +
$$

$$
\sum_{0 \le i \le m'+1} \left(\mathbb{E}\left[\left(\bar{p}_{k_2}^{\mathcal{W},\epsilon_i-}(\theta) - \hat{p}_{k_2}^{\mathcal{W},\epsilon_i-}(\theta)\right)^2\right] + \mathbb{E}\left[\left(\hat{p}_{k_2}^{\mathcal{W},\epsilon_i-}(\theta) - p_{k_2}^{\mathcal{W},\epsilon_i-}(\theta)\right)^2\right]\right)
$$

since $\bar{p}_{k_2}^{\mathcal{W},\epsilon_i}(\theta)$ and $\bar{p}_{k_2}^{\mathcal{W},\epsilon_i-}(\theta)$ are conditionally unbiased for $\hat{p}_{k_2}^{\mathcal{W},\epsilon_i}(\theta)$ and $\hat{p}_{k_2}^{\mathcal{W},\epsilon_i-}(\theta)$

$$
\le \sum_{0 \le i \le m'+1} \left(\mathbb{E}\left[\mathbb{E}_*\left[\left(\bar{p}_{k_2}^{\mathcal{W},\epsilon_i}(\theta) - \hat{p}_{k_2}^{\mathcal{W},\epsilon_i}(\theta)\right)^2\right]\right] + \mathbb{E}\left[\left(\tilde{p}_{k_2}^{\mathcal{W},\epsilon_i}(\theta) - p_{k_2}^{\mathcal{W},\epsilon_i}(\theta)\right)^2\right]\right) +
$$

$$
\sum_{0 \le i \le m'+1} \left(\mathbb{E}\left[\mathbb{E}_*\left[\left(\bar{p}_{k_2}^{\mathcal{W},\epsilon_i-}(\theta) - \hat{p}_{k_2}^{\mathcal{W},\epsilon_i-}(\theta)\right)^2\right]\right] + \mathbb{E}\left[\left(\tilde{p}_{k_2}^{\mathcal{W},\epsilon_i-}(\theta) - p_k^{\mathcal{W},\epsilon_i-}(\theta)\right)^2\right]\right)
$$

$$
\le (m'+2)\left(\frac{2}{B_2} + \frac{2}{\lfloor n/k_2 \rfloor}\right) \le (m+2)\left(\frac{2}{B_2} + \frac{4}{n/k_2}\right).
$$

By Minkowski inequality, the supremum satisfies

$$
\mathbb{E}\left[\sup_{\epsilon \in [0,\delta/2]} \left|\bar{p}_{k_2}^{\mathcal{W},\epsilon}(\theta) - p_{k_2}^{\mathcal{W},\epsilon}(\theta)\right|\right] \le \sqrt{(m+2)\left(\frac{2}{B_2} + \frac{4}{n/k_2}\right)} + \frac{1}{m}.
$$

Choosing $m$ such that $m \to \infty$, $m/B_2 \to 0$ and $mk_2/n \to 0$ leads to the convergence $\sup_{\epsilon \in [0,\delta/2]} \left|\bar{p}_{k_2}^{\mathcal{W},\epsilon}(\theta) - p_{k_2}^{\mathcal{W},\epsilon}(\theta)\right| \to 0$ in probability. Since $\Theta$ has finite cardinality and has a finite number of subsets, it also holds that

$$
\sup_{\mathcal{W} \subseteq \Theta, \theta \in \mathcal{W}, \epsilon \in [0,\delta/2]} \left|\bar{p}_{k_2}^{\mathcal{W},\epsilon}(\theta) - p_{k_2}^{\mathcal{W},\epsilon}(\theta)\right| \to 0 \text{ in probability.} \tag{49}
$$

Recall the bound (43) from the proof of Lemma 7. Here we have the similar bound $\max_{\theta \in \mathcal{W}\setminus\mathcal{W}^\delta} p_{k_2}^{\mathcal{W},\epsilon}(\theta) \le \mathbb{P}\left(\widehat{\mathcal{W}}_{k_2}^\epsilon \not\subseteq \mathcal{W}^\delta\right)$, and hence

$$
\sup_{\epsilon \in [0,\delta/2]} \max_{\theta \in \mathcal{W}\setminus\mathcal{W}^\delta} p_k^{\mathcal{W},\epsilon}(\theta) \le \sup_{\epsilon \in [0,\delta/2]} \mathbb{P}\left(\widehat{\mathcal{W}}_{k_2}^\epsilon \not\subseteq \mathcal{W}^\delta\right) = \mathbb{P}\left(\widehat{\mathcal{W}}_{k_2}^{\delta/2} \not\subseteq \mathcal{W}^\delta\right).
$$

We bound the probability $\mathbb{P}\left(\widehat{\mathcal{W}}_{k_2}^{\delta/2} \not\subseteq \mathcal{W}^\delta\right)$ more carefully. We let

$$
\Delta_o := \min\left\{L(\theta') - L(\theta) : \theta, \theta' \in \Theta, L(\theta') > L(\theta)\right\} > 0,
$$

$$
\hat{L}_{k_2}(\theta) := \frac{1}{k_2} \sum_{i=1}^{k_2} l(\theta, z_i),
$$

and have

$$\left\{ \widehat{\mathcal{W}}_{k_2}^{\delta/2} \not\subseteq \mathcal{W}^\delta \right\}$$

$$\subseteq \bigcup_{\theta,\theta' \in \mathcal{W} \text{ s.t. } L(\theta')-L(\theta)>\delta} \left\{ \hat{L}_{k_2}(\theta') \leq \hat{L}_{k_2}(\theta) + \frac{\delta}{2} \right\}$$

$$\subseteq \bigcup_{\theta,\theta' \in \Theta \text{ s.t. } L(\theta')-L(\theta)>\delta} \left\{ \hat{L}_{k_2}(\theta') - L(\theta') + L(\theta') - L(\theta) \leq \hat{L}_{k_2}(\theta) - L(\theta) + \frac{\delta}{2} \right\}$$

$$\subseteq \bigcup_{\theta,\theta' \in \Theta \text{ s.t. } L(\theta')-L(\theta)>\delta} \left\{ \hat{L}_{k_2}(\theta') - L(\theta') + \max(\Delta,\delta) \leq \hat{L}_{k_2}(\theta) - L(\theta) + \frac{\delta}{2} \right\}$$

by the definition of $\Delta_o$

$$\subseteq \bigcup_{\theta,\theta' \in \Theta} \left\{ \hat{L}_{k_2}(\theta') - L(\theta') + \max\left(\Delta_o - \frac{\delta}{2}, \frac{\delta}{2}\right) \leq \hat{L}_{k_2}(\theta) - L(\theta) \right\}$$

$$\subseteq \bigcup_{\theta,\theta' \in \Theta} \left\{ \hat{L}_{k_2}(\theta') - L(\theta') \leq -\max\left(\frac{\Delta_o}{2} - \frac{\delta}{4}, \frac{\delta}{4}\right) \text{ or } \hat{L}_{k_2}(\theta) - L(\theta) \geq \max\left(\frac{\Delta_o}{2} - \frac{\delta}{4}, \frac{\delta}{4}\right) \right\}$$

$$\subseteq \bigcup_{\theta \in \Theta} \left\{ \left| \hat{L}_{k_2}(\theta) - L(\theta) \right| \geq \max\left(\frac{\Delta_o}{2} - \frac{\delta}{4}, \frac{\delta}{4}\right) \right\}$$

$$\subseteq \bigcup_{\theta \in \Theta} \left\{ \left| \hat{L}_{k_2}(\theta) - L(\theta) \right| \geq \frac{\Delta_o}{4} \right\}$$

$$\subseteq \left\{ \sup_{\theta \in \Theta} \left| \hat{L}_{k_2}(\theta) - L(\theta) \right| \geq \frac{\Delta_o}{4} \right\},$$

where the last line holds because $\max\left(\Delta_o/2 - \delta/4, \delta/4\right) \geq \Delta_o/4$. This gives

$$\sup_{\epsilon \in [0,\delta/2]} \max_{\theta \in \mathcal{W}\setminus\mathcal{W}^\delta} p_{k_2}^{\mathcal{W},\epsilon}(\theta) \leq T_{k_2}\left(\frac{\Delta_o}{4}\right) \to 0 \text{ as } k_2 \to \infty.$$

We also have the trivial bound $\inf_{\epsilon \in [0,\delta/2]} \max_{\theta \in \mathcal{W}} p_{k_2}^{\mathcal{W},\epsilon}(\theta) = \max_{\theta \in \mathcal{W}} p_{k_2}^{\mathcal{W},0}(\theta) \geq 1/|\mathcal{W}|$, where the inequality comes from the fact that $\sum_{\theta \in \mathcal{W}} p_{k_2}^{\mathcal{W},0}(\theta) \geq 1$. Now choose a $\underline{k} < \infty$ such that

$$T_{k_2}\left(\frac{\Delta_o}{4}\right) \leq \frac{1}{2|\Theta|} \text{ for all } k_2 \geq \underline{k}$$

and we have for all $k_2 \geq \underline{k}$ and all non-empty $\mathcal{W} \subseteq \Theta$ that

$$\inf_{\epsilon \in [0,\delta/2]} \left( \max_{\theta \in \mathcal{W}} p_{k_2}^{\mathcal{W},\epsilon}(\theta) - \max_{\theta \in \mathcal{W}\setminus\mathcal{W}^\delta} p_{k_2}^{\mathcal{W},\epsilon}(\theta) \right) \geq \inf_{\epsilon \in [0,\delta/2]} \max_{\theta \in \mathcal{W}} p_{k_2}^{\mathcal{W},\epsilon}(\theta) - \sup_{\epsilon \in [0,\delta/2]} \max_{\theta \in \mathcal{W}\setminus\mathcal{W}^\delta} p_{k_2}^{\mathcal{W},\epsilon}(\theta)$$

$$\geq \frac{1}{|\mathcal{W}|} - \frac{1}{2|\Theta|} \geq \frac{1}{2|\Theta|}.$$

Due to the uniform convergence (49), we have

$$\min_{\mathcal{W} \subseteq \Theta} \inf_{\epsilon \in [0,\delta/2]} \left( \max_{\theta \in \mathcal{W}} \bar{p}_{k_2}^{\mathcal{W},\epsilon}(\theta) - \max_{\theta \in \mathcal{W}\setminus\mathcal{W}^\delta} \bar{p}_{k_2}^{\mathcal{W},\epsilon}(\theta) \right) \to \min_{\mathcal{W} \subseteq \Theta} \inf_{\epsilon \in [0,\delta/2]} \left( \max_{\theta \in \mathcal{W}} p_{k_2}^{\mathcal{W},\epsilon}(\theta) - \max_{\theta \in \mathcal{W}\setminus\mathcal{W}^\delta} p_{k_2}^{\mathcal{W},\epsilon}(\theta) \right)$$

in probability, and hence

$$\mathbb{P}\left( \min_{\mathcal{W} \subseteq \Theta} \inf_{\epsilon \in [0,\delta/2]} \left( \max_{\theta \in \mathcal{W}} \bar{p}_{k_2}^{\mathcal{W},\epsilon}(\theta) - \max_{\theta \in \mathcal{W}\setminus\mathcal{W}^\delta} \bar{p}_{k_2}^{\mathcal{W},\epsilon}(\theta) \right) \leq 0 \right) \to 0. \tag{50}$$

Finally, we combine all the pieces to get

$$\left\{ \hat{\theta}_n \notin \Theta^{2\delta} \right\}$$

$$\subseteq \quad \left\{ \mathcal{S} \cap \Theta^\delta = \emptyset \right\} \cup \left\{ \hat{\theta}_n \notin \mathcal{S}^\delta \right\}$$

$$\subseteq \quad \left\{ \mathcal{S} \cap \Theta^\delta = \emptyset \right\} \cup \left\{ \max_{\theta \in \mathcal{S}} \bar{p}_{k_2}^{\mathcal{S},\epsilon}(\theta) - \max_{\theta \in \mathcal{S} \setminus \mathcal{S}^\delta} \bar{p}_{k_2}^{\mathcal{S},\epsilon}(\theta) \leq 0 \right\}$$

$$\subseteq \quad \left\{ \mathcal{S} \cap \Theta^\delta = \emptyset \right\} \cup \left\{ \epsilon > \frac{\delta}{2} \right\} \cup \left\{ \inf_{\epsilon \in [0,\delta/2]} \left( \max_{\theta \in \mathcal{S}} \bar{p}_{k_2}^{\mathcal{S},\epsilon}(\theta) - \max_{\theta \in \mathcal{S} \setminus \mathcal{S}^\delta} \bar{p}_{k_2}^{\mathcal{S},\epsilon}(\theta) \right) \leq 0 \right\}$$

$$\subseteq \quad \left\{ \mathcal{S} \cap \Theta^\delta = \emptyset \right\} \cup \left\{ \epsilon > \frac{\delta}{2} \right\} \cup \left\{ \min_{\mathcal{W} \subseteq \Theta} \inf_{\epsilon \in [0,\delta/2]} \left( \max_{\theta \in \mathcal{W}} \bar{p}_{k_2}^{\mathcal{W},\epsilon}(\theta) - \max_{\theta \in \mathcal{W} \setminus \mathcal{W}^\delta} \bar{p}_{k_2}^{\mathcal{W},\epsilon}(\theta) \right) \leq 0 \right\}.$$

By Lemma 6 we have $\mathbb{P}\left( \mathcal{S} \cap \Theta^\delta = \emptyset \right) \to 0$ under the conditions that $\limsup_{k \to \infty} \mathcal{E}_{k,\delta} < 1$ and $k_1, n/k_1, B_1 \to \infty$. Together with the condition $\mathbb{P}\left( \epsilon \geq \delta/2 \right) \to 0$ and (50), we conclude $\mathbb{P}\left( \hat{\theta}_n \notin \Theta^{2\delta} \right) \to 0$ by the union bound. $\qquad \square$

## APPENDIX D  SUPPLEMENTARY MATERIAL FOR NUMERICAL EXPERIMENTS

This section supplements Section 3. We first provide details for the architecture of the neural networks in Section D.1, and the considered stochastic programs in Section D.2. Section D.3 presents a comprehensive profiling of hyperparameters of our methods, and Section D.4 provides additional experimental results.

### D.1  MLP ARCHITECTURE

The input layer of our MLPs has the same number of neurons as the input dimension, and the output layer is a single neuron that gives the final prediction. All activations are ReLU. For experiments on synthetic data, the architecture of hidden layers is as follows under different numbers of hidden layers $H$:

- $H = 2$: Each hidden layer has 50 neurons.
- $H = 4$: There are 50, 300, 300, 50 neurons from the first to the fourth hidden layer.
- $H = 6$: There are 50, 300, 500, 500 300, 50 neurons from the first to the sixth hidden layer.
- $H = 8$: There are 50, 300, 500, 800, 800 500 300, 50 neurons from the first to the eighth hidden layer.

For experiments on real data, the MLP with 4 hidden layers has 100, 300, 300, 100 neurons from the first to the fourth hidden layer.

### D.2  STOCHASTIC PROGRAMMING PROBLEMS

**Resource allocation** (Kleywegt et al., 2002)  The decision maker wants to choose a subset of $m$ projects. A quantity $q$ of low-cost resource is available to be allocated, and any additional resource can be obtained at an incremental unit cost $c$. Each project $i$ has an expected reward $r_i$. The amount of resource required by each project $i$ is a random variable, denoted by $W_i$. We can formulate the problem as

$$\max_{\theta \in \{0,1\}^m} \left\{ \sum_{i=1}^m r_i \theta_i - c \mathbb{E} \left[ \sum_{i=1}^m W_i \theta_i - q \right]^+ \right\}. \tag{51}$$

In the experiment, we consider the three-product scenario, i.e., $m = 3$, and assume that the random variable $W_i$ follows the Pareto distribution.

**Supply chain network design** (Shapiro et al., 2021, Chapter 1.5)  Consider a network of suppliers, processing facilities, and customers, where the goal is to optimize the overall supply chain efficiency.

The supply chain design problem can be formulated as a two-stage stochastic optimization problem

$$\min_{\theta \in \{0,1\}^{|P|}} \sum_{p \in P} c_p \theta_p + \mathbb{E}[Q(\theta, z)], \tag{52}$$

where $P$ is the set of processing facilities, $c_p$ is the cost of opening facility $p$, and $z$ is the vector of (random) parameters, i.e., $(h, q, d, s, R, M)$ in (53). Function $Q(\theta, z)$ represents the total processing and transportation cost, and it is equal to the optimal objective value of the following second-stage problem:

$$
\begin{aligned}
\min_{y \geq 0, z \geq 0} \quad & q^\top y + h^\top z \\
\text{s.t.} \quad & Ny = 0, \\
& Cy + z \geq d, \\
& Sy \leq s, \\
& Ry \leq M\theta,
\end{aligned}
\tag{53}
$$

where $N, C, S$ are appropriate matrices that describe the network flow constraints. More details about this example can be found in (Shapiro et al., 2021, Chapter 1.5). In our experiment, we consider the scenario of 3 suppliers, 2 facilities, 3 consumers, and 5 products. We choose supply $s$ and demand $d$ as random variables that follow the Pareto distribution.

**Maximum weight matching and stochastic linear program**   We explore both the maximum weight matching problem and the linear program that arises from it. Let $G = (V, E)$ be a general graph, where each edge $e \in E$ is associated with a (possibly) random weight $w_e$. For each node $v \in V$, denote $E(v)$ as the set of edges incident to $v$. Based on this setup, we consider the following linear program

$$
\begin{aligned}
\max_{\theta \in [0,1]^{|E|}} \quad & \mathbb{E}\left[\sum_{e \in E} w_e \theta_e\right] \\
\text{subject to} \quad & \sum_{e \in E(v)} a_e \theta_e \leq 1, \qquad \forall v \in V,
\end{aligned}
\tag{54}
$$

where $a_e$ is some positive coefficient. When $a_e = 1$ for all $e \in E$ and $\theta$ is restricted to the discrete set $\{0,1\}^{|E|}$, (54) is equivalent to the maximum weight matching problem. For the maximum weight matching, we consider a complete bipartite graph with 5 nodes on each side (the dimension is 25). The weights of nine edges are Pareto distributed and the remaining are prespecified constants. For the linear programming problem, we consider a 28-dimensional instance (the underlying graph is an 8-node complete graph), where all $w_e$ follows the Pareto distribution.

**Mean-variance portfolio optimization**   Consider constructing a portfolio based on $m$ assets. Each asset $i$ has a rate of return $r_i$ that is random with mean $\mu_i$. The goal is to minimize the variance of the portfolio while ensuring that the expected rate of return surpasses a target level $b$. The problem can be formulated as

$$
\begin{aligned}
\min_{\theta} \quad & \mathbb{E}\left[\left(\sum_{i=1}^m (r_i - \mu_i)\theta_i\right)^2\right] \\
\text{subject to} \quad & \sum_{i=1}^m \mu_i \theta_i \geq b, \\
& \sum_{i=1}^m \theta_i = 1, \\
& \theta_i \geq 0 \; \forall i = 1, \ldots, m
\end{aligned}
\tag{55}
$$

where $\theta$ is the decision variable and each $\mu_i$ is assumed known. In the experiment, we consider a scenario with 10 assets, i.e., $m = 10$, where each rate of return $r_i$ is a linear combination of the rates of return of 100 underlying assets in the form $r_i = \tilde{r}_{10(i-1)+1}/2 + \sum_{j=1}^{100} \tilde{r}_j/200$. Each of these underlying assets has a Pareto rate of return $\tilde{r}_j, j = 1, \ldots, 100$.

### D.3   HYPERPARAMETER PROFILING

We test the effect of different hyperparameters in our ensemble methods, including subsample sizes $k, k_1, k_2$, ensemble sizes $B, B_1, B_2$, and threshold $\epsilon$. Throughout this profiling stage, we use the sample average approximation (SAA) as the base algorithm. To profile the effect of subsample sizes and ensemble sizes, we consider the resource allocation problem.

**Subsample size**   We explored scenarios where $k$ (equivalently $k_1$ and $k_2$) is both dependent on and independent of the total sample size $n$ (see Figures 5a, 6a, and 6b). The results suggest that a constant $k$ generally suffices, although the optimal $k$ varies by problem instance. For example, Figures 6a and

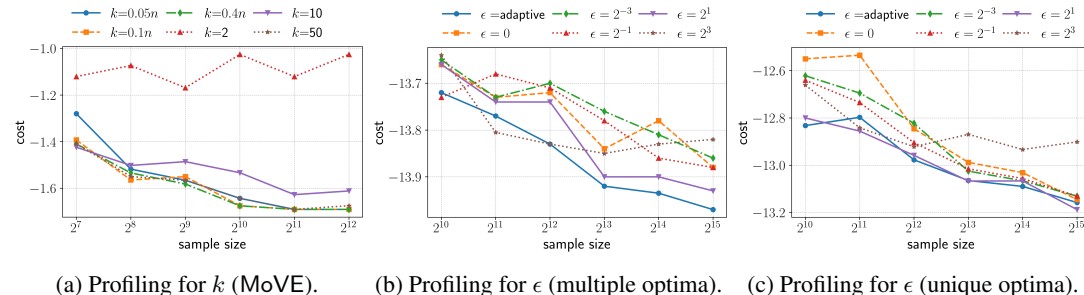

(a) Profiling for $k$ (MoVE).  (b) Profiling for $\epsilon$ (multiple optima).  (c) Profiling for $\epsilon$ (unique optima).

Figure 5: Profiling for subsample size $k$ and threshold $\epsilon$. (a): Resource allocation problem, where $B = 200$; (b) and (c): Linear program, where $k_1 = k_2 = \max(10, 0.005n)$, $B_1 = 20$, and $B_2 = 200$.

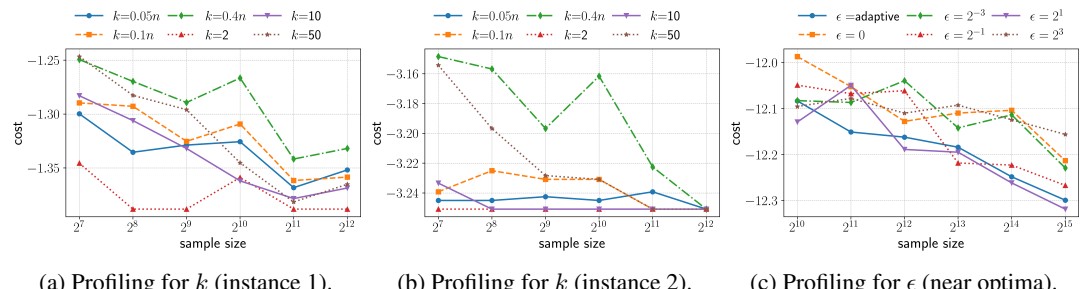

(a) Profiling for $k$ (instance 1).  (b) Profiling for $k$ (instance 2).  (c) Profiling for $\epsilon$ (near optima).

Figure 6: Profiling results for subsample size $k$ and threshold $\epsilon$. (a) and (b): Resource allocation problem using MoVE, where $B = 200$; (c): Linear program with multiple near optima using ROVE, where $k_1 = k_2 = \max(10, 0.005n)$, $B_1 = 20$, and $B_2 = 200$.

6b show that $k = 2$ yields the best performance; increasing $k$ degrades results. Conversely, in Figure 5a, $k = 2$ proves inadequate, with larger $k$ delivering good results. The underlying reason is that the effective performance of MoVE requires $\theta^* \in \arg\max_{\theta \in \Theta} p_k(\theta)$. In the former, this is achieved with only two samples, enabling MoVE to identify $\theta^*$ with a subsample size of 2. For the latter, a higher number of samples is required to meet this condition, explaining the suboptimal performance at $k = 2$. In Figure 7, we simulate $p_k(\theta)$ for the two cases, which further explains the influence of the subsample size.

**Ensemble size**  In Figure 8, we illustrate the performance of MoVE and ROVE under different $B, B_1, B_2$, where we set $k = k_1 = k_2 = 10$ and $\epsilon = 0.005$. From the figure, we find that the performance of our ensemble methods is improving in the ensemble sizes.

**Threshold $\epsilon$**  The optimal choice of $\epsilon$ in ROVE and ROVEs is problem-dependent and related to the number of (near) optimal solutions. This dependence is illustrated by the performance of ROVE shown in Figures 5b and 5c. Hence, we propose an adaptive strategy defined as follows: Let $g(\epsilon) := 1/B_2 \cdot \sum_{b=1}^{B_2} \mathbb{1}(\hat{\theta}_n(\epsilon) \in \widehat{\Theta}_{k_2}^{\epsilon,b})$, where we use $\hat{\theta}_n(\epsilon)$ to emphasize the dependency of $\hat{\theta}_n$ on $\epsilon$. Then, we select $\epsilon^* := \min\{\epsilon : g(\epsilon) \geq 1/2\}$. By definition, $g(\epsilon)$ is the proportion of times that $\hat{\theta}_n(\epsilon)$ is included in the "near optimum set" $\widehat{\Theta}_{k_2}^{\epsilon,b}$. The choice of $\epsilon^*$ makes it more likely for the true optimal solution to be included in the "near optimum set", instead of being ruled out by suboptimal solutions. Practically, $\epsilon^*$ can be efficiently determined using a binary search as an intermediate step between Phases I and II. To prevent data leakage, we compute $\epsilon^*$ using $\mathbf{z}_{1:\lfloor \frac{n}{2} \rfloor}$ (Phase I data) for ROVEs. From Figure 5, we observe that this adaptive strategy exhibits decent performance for all scenarios. Similar behaviors can also be observed for ROVEs in Figure 9.

### D.4 ADDITIONAL EXPERIMENTAL RESULTS

Here, we present additional figures that supplement the experiments and discussions in Section 3. Recall that MoVE refers to Algorithm 1, ROVE refers to Algorithm 2 without data splitting, and

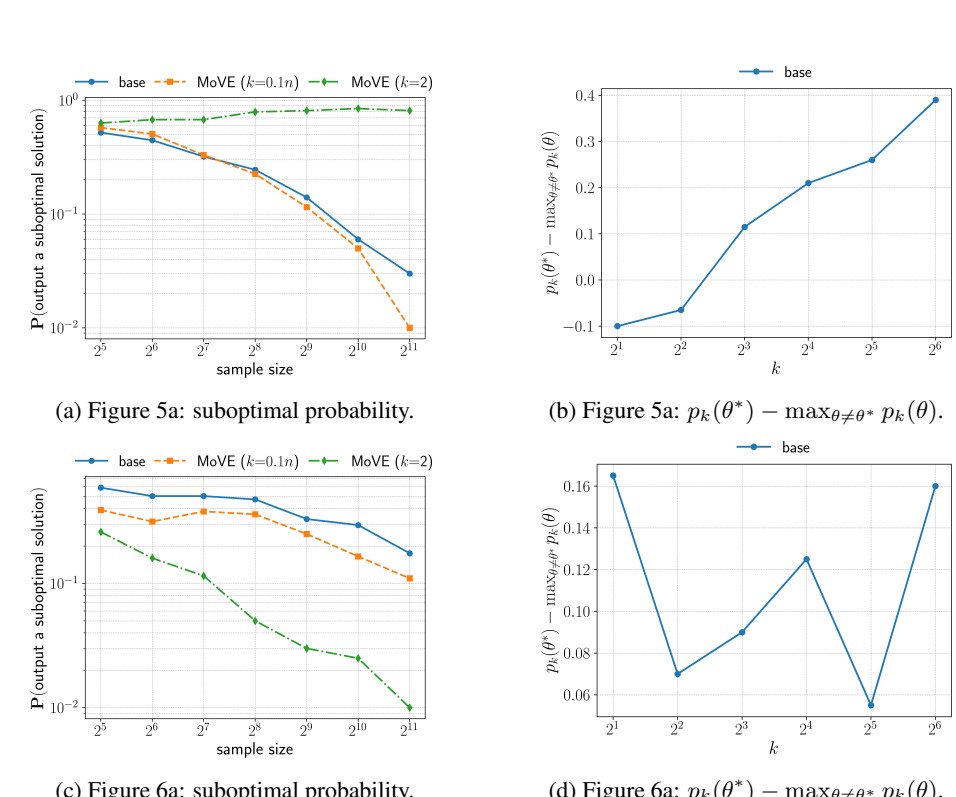

(a) Figure 5a: suboptimal probability.

(b) Figure 5a: $p_k(\theta^*) - \max_{\theta \neq \theta^*} p_k(\theta)$.

(c) Figure 6a: suboptimal probability.

(d) Figure 6a: $p_k(\theta^*) - \max_{\theta \neq \theta^*} p_k(\theta)$.

Figure 7: Performance of MoVE ($B = 200$) in resource allocation, corresponding to the two instances in Figures 5a and 6a. Subfigures (b) and (d) explain the behaviors of MoVE with different subsample sizes $k$: In (b), we find that $p_k(\theta^*) - \max_{\theta \neq \theta^*} p_k(\theta) < 0$ for $k \leq 4$, which results in the poor performance of MoVE with $k = 2$ in Figure 5a; In (d), we have $p_2(\theta^*) - \max_{\theta \neq \theta^*} p_2(\theta) \approx 0.165$, thereby enabling MoVE to distinguish the optimal solution only using subsamples of size two, which results in the good performance of MoVE with $k = 2$ in Figure 6a.

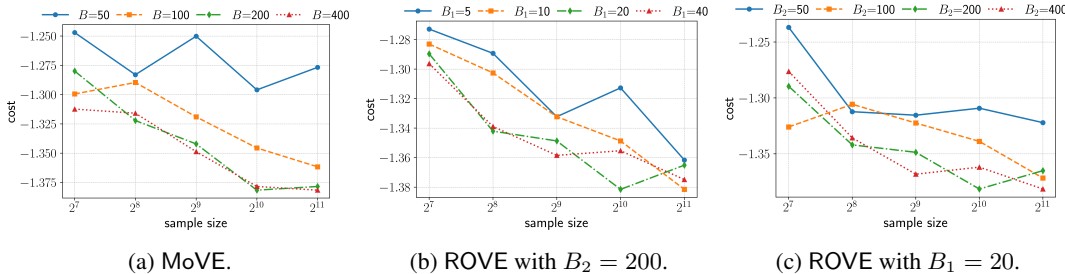

(a) MoVE.

(b) ROVE with $B_2 = 200$.

(c) ROVE with $B_1 = 20$.

Figure 8: Profiling for ensemble sizes $B, B_1, B_2$ in resource allocation. Subsample size is chosen as $k = k_1 = k_2 = 10$.

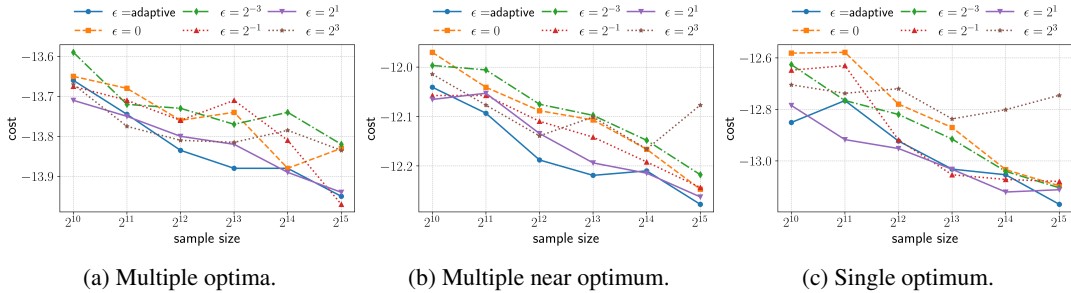

(a) Multiple optima.        (b) Multiple near optimum.        (c) Single optimum.

Figure 9: Performance of ROVEs in three instances of linear programs under different thresholds $\epsilon$. The setting is identical to that of Figures 5b, 5c, and 6c for ROVE. Hyperparameters: $k_1 = k_2 = \max(10, 0.005n)$, $B_1 = 20$, and $B_2 = 200$. Compared with profiling results for ROVE, we observe that the value of $\epsilon$ has similar impacts on the performance of ROVEs. Moreover, the proposed adaptive strategy also behaves well for ROVEs.

ROVEs refers to Algorithm 2 with data splitting. We briefly introduce each figure below and refer the reader to the figure caption for detailed discussions. Figures 10-16 all follow the recommended configuration listed in Section 3.

- Figure 10 supplements the results in Figure 1 with MLPs with $H = 2, 4$ hidden layers.

- Figure 11 shows results for MLP regression on a slightly different synthetic example than in Section 3.1.

- Figures 12 and 13 show results for regression on synthetic data with least squares regression and Ridge regression as the base learning algorithms respectively.

- Figure 14 shows results on the stochastic linear program example with light-tailed uncertainties.

- Figure 15 contains additional results on the supply chain network design example for different choices of hyperparameters and a different problem instance with strong correlation between solutions.

- In Figure 16, we apply our ensemble methods to resource allocation and maximum weight matching using DRO with Wasserstein metric as the base algorithm. This result, together with Figure 3 where the base algorithm is SAA, demonstrates that the benefit of our ensemble methods is agnostic to the underlying base algorithm.

- In Figure 17, we simulate the generalization sensitivity $\bar{\eta}_{k,\delta}$, defined in (29), which explains the superior performance of ROVE and ROVEs in the presence of multiple optimal solutions.

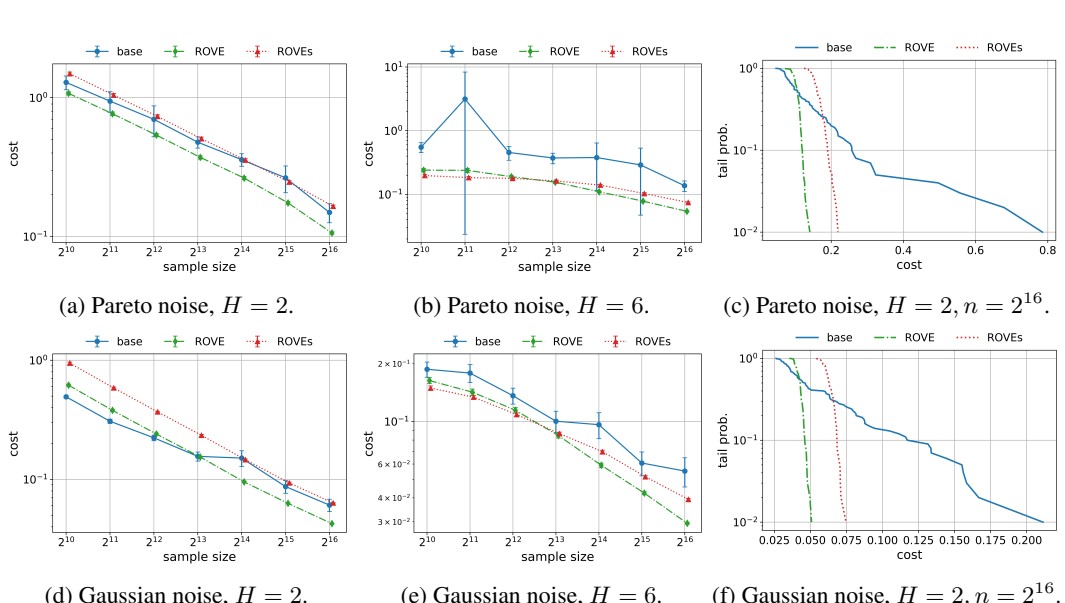

(a) Pareto noise, $H = 2$.    (b) Pareto noise, $H = 6$.    (c) Pareto noise, $H = 2, n = 2^{16}$.

(d) Gaussian noise, $H = 2$.    (e) Gaussian noise, $H = 6$.    (f) Gaussian noise, $H = 2, n = 2^{16}$.

Figure 10: Results of neural networks on synthetic data with the same setup described in Section 3.1. (a)(b)(d)(e): Expected out-of-sample costs (MSE) with $95\%$ confidence intervals under different noise distributions and varying numbers of hidden layers ($H$). (c) and (f): Tail probabilities of out-of-sample costs. In (a), ROVEs slightly underperforms the base learner probably due to the weak expressiveness and hence high bias of the MLP with 2 hidden layers.

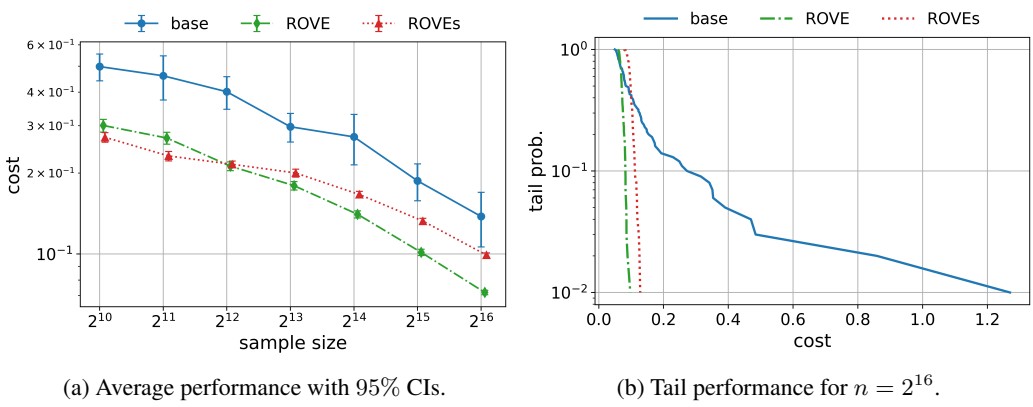

(a) Average performance with $95\%$ CIs.    (b) Tail performance for $n = 2^{16}$.

Figure 11: Results on synthetic data with an MLP of $H = 4$ hidden layers. The setup is the same as in Section 3.1 except that the dimension of $X$ is now 30 and the data generation becomes $Y = (1/30) \cdot \sum_{j=1}^{30} \log(X_j + 1) + \varepsilon$, where each $X_j$ is drawn independently from $\mathrm{Unif}(0, 2 + 198(j-1)/29)$.

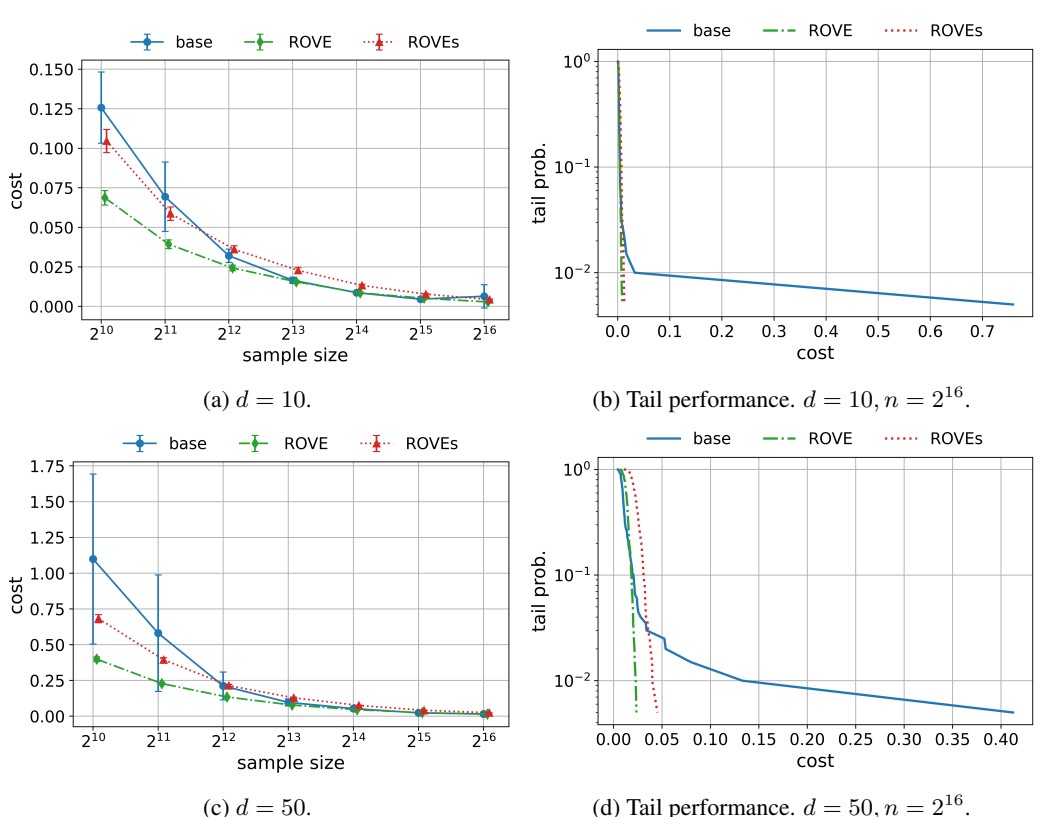

(a) $d = 10$.

(b) Tail performance. $d = 10, n = 2^{16}$.

(c) $d = 50$.

(d) Tail performance. $d = 50, n = 2^{16}$.

Figure 12: Linear regression on synthetic data with least squares regression as the base learning algorithm. Given the input dimension $d$, the data generation is $Y = \sum_{i=1}^{d}(-10 + 20(i - 1)/(d - 1))X_i + \varepsilon_1 - \varepsilon_2$ where each $X_i$ is independent $\text{Unif}(0, 2 + 18(i - 1)/(d - 1))$ and each $\varepsilon_j, j = 1, 2$ is $\text{Pareto}(2.1)$ and independent of $X$. (a) and (c): Expected out-of-sample error with $95\%$ confidence interval. (b) and (d): Tail probabilities of out-of-sample errors.

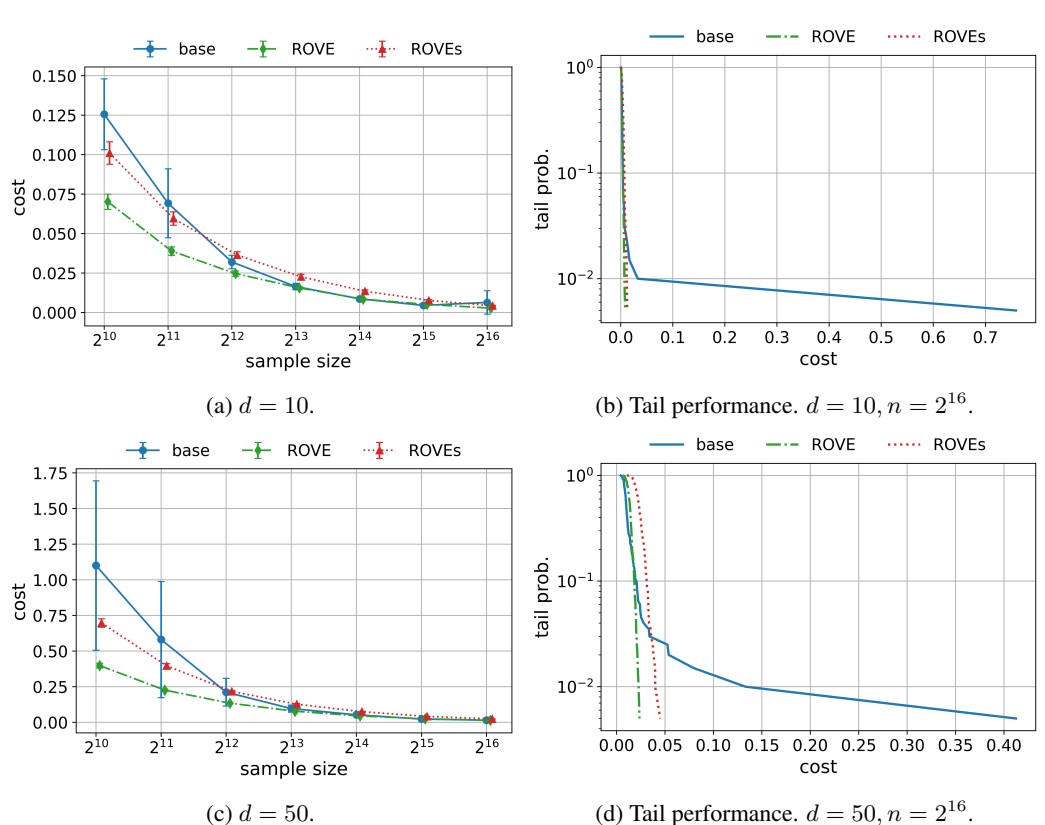

(a) $d = 10$.

(b) Tail performance. $d = 10, n = 2^{16}$.

(c) $d = 50$.

(d) Tail performance. $d = 50, n = 2^{16}$.

Figure 13: Linear regression on synthetic data with Ridge regression as the base learning algorithm. The same data generation as in Figure 12. (a) and (c): Expected out-of-sample error with $95\%$ confidence interval. (b) and (d): Tail probabilities of out-of-sample errors.

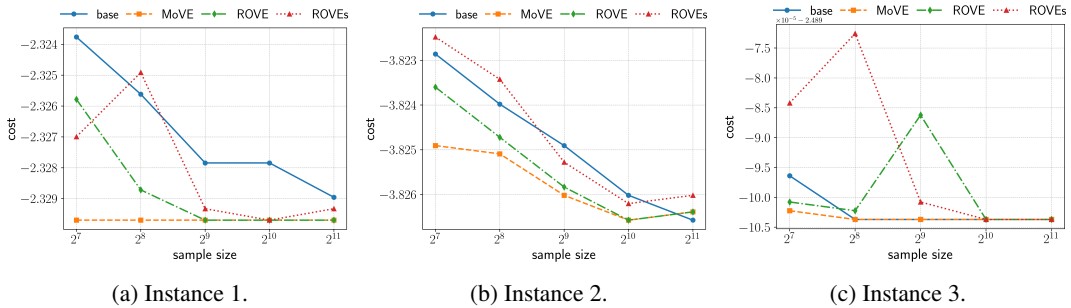

(a) Instance 1.

(b) Instance 2.

(c) Instance 3.

Figure 14: Results for linear programs with light-tailed objectives. The base algorithm is SAA.

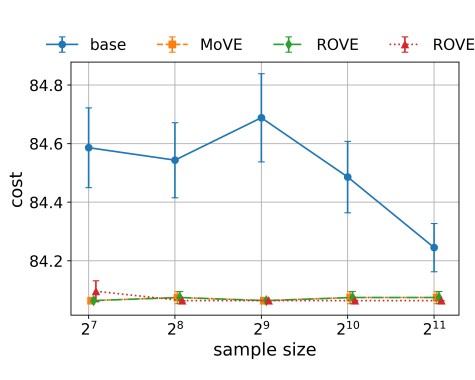 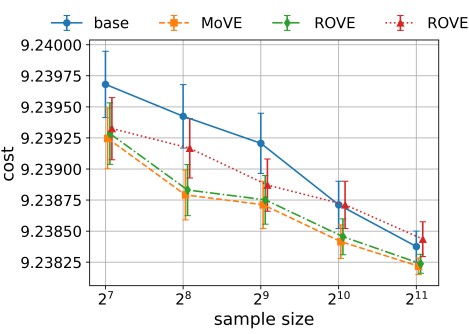

(a) $k = k_1 = k_2 = \max(10, n/10)$.

(b) A different instance with strong correlation.

Figure 15: Results for supply chain network design. (a): The same problem instance as in Section 3.2 under a different hyperparameter choice: $k = \max(10, n/10)$, $B = 200$ for MoVE and $k_1 = k_2 = \max(10, n/10)$, $B_1 = 20$, $B_2 = 200$ for ROVE and ROVEs. (b): The same setup as in Section 3.2 but on a different problem instance for which the objectives under different solutions are strongly correlated. The strong correlation cancels out most of the heavy-tailed noise between solutions, making the base algorithm less susceptible to these noises, thus our ensemble methods appear less effective.

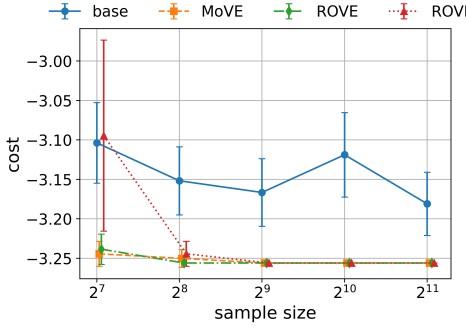 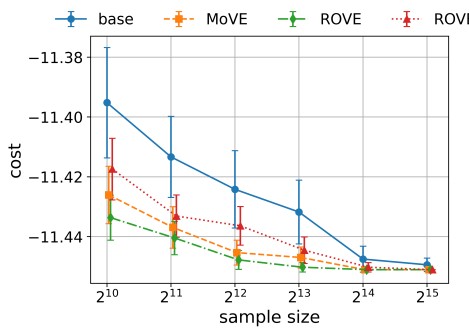

(a) Resource allocation.

(b) Maximum weight matching.

Figure 16: Results for resource allocation and maximum weight matching when the base algorithm is DRO using 1-Wasserstein metric with the $l_\infty$ norm.

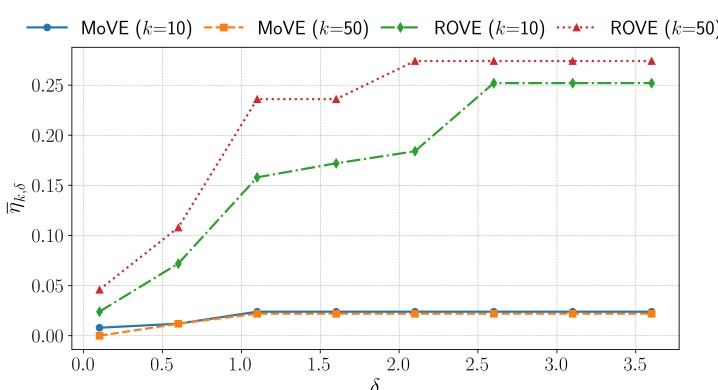

Figure 17: Comparison of $\bar{\eta}_{k,\delta}$ for MoVE and ROVE in a linear program with multiple optima (corresponds to the instance in Figure 3e). Threshold $\epsilon$ is chosen as $\epsilon = 4$ when $k = k_1 = k_2 = 10$ and $\epsilon = 2.5$ when $k = k_1 = k_2 = 50$, according to the adaptive strategy. Note that $\bar{\eta}_{k,\delta} = \max_{\theta \in \Theta} p_k(\theta) - \max_{\theta \in \Theta \setminus \Theta^\delta} p_k(\theta)$ by (29), which measures the generalization sensitivity. For MoVE, we have $p_k(\theta) = \mathbb{P}(\hat{\theta}_k^{SAA} = \theta)$; and for ROVE, we have $p_k(\theta) = \mathbb{P}(\theta \in \widehat{\Theta}_k^\epsilon)$, where $\widehat{\Theta}_k^\epsilon$ is the $\epsilon$-optimal set of SAA defined in (28). From the figure, we can observe that the issue brought by the presence of multiple optimal solutions can be alleviated using the two-phase strategy in ROVE.

