# OpenReview forum: "Subsampled Ensemble Can Improve Generalization Tail Exponentially"
_ICLR.cc/2025/Conference — Submitted to ICLR 2025_

### Official Review · Reviewer_aq9J · 2024-10-28

**Soundness:** 2
**Presentation:** 3
**Contribution:** 3
**Rating:** 8
**Confidence:** 4

**Summary:**

The paper presents an asymptotic theory of ensembling that focuses on the tail distribution of the excess error, that is, the probability that the generalization error exceeds the best possible one by more than a specified constant.  The paper shows that the proposed ensembles can have exponential tails even if the base algorithm has polynomial tails.  Experiments show that ensembles do not only improve the variance, but can have a dramatic effect on the tails.



/* Raised score after discussion */

**Strengths:**

The paper is clearly written.  I learned something from this paper (but maybe not what the authors claim.)

**Weaknesses:**

I believe that the asymptotic viewpoint undermines the theoretical result. Even without going to the refined empirical process literature, we know from Vapnik (Vapnik 1998, Wiley) that if the function L(theta,z) has a finite capacity (VC dim for classification, extensions of VC dim for continuous losses with either bounded range or bounded p-th moments), we have relative uniform convergence results with asymptotically exponential tails.  Hence in that setup, any reasonable base learner will also have exponential tails.  On the other hand, if the VC dim is infinite, we know that it will be very difficult to get any guarantee at all (see also JMLR 11 (2010) 2635-2670).

I can see therefore two ways to interpret the results of the paper:

* We can consider that the author prove that ensembling alone is as good as empirical risk optimization and variants in the sense that it is going to give exponential tails using possibly bad base algorithms.  However this amounts to saying that averaging is optimization, and we know that averaging can be superior to optimization.

* We can also recall that the VC exponential bounds are often meaningless for practical data sizes. Okay they're exponential in the very large sample limit, but for anything practical, they don't look exponential.  Ensembling can be a way to improve dramatically these practical tails (the experiments support this point of view). But that also means that the asymptotic perspective of the theory proposed in the paper is not the right way to look at this.

**Questions:**

1) Can you give a simple self-contained example of a "reasonable" base learner that provably does not have exponential tails but has polynomial ones?  It seems to me that this means that you either have to consider infinite VC cases, or losses with unbounded moments, or algorithms that are far inferior to ERM in the limit.  There might be a way to make a convincing case for the last option, that is, inferior to ERM in the limit, but nevertheless practically reasonable in practical settings.

2) A brief review of the proofs suggests that they might repurposed to argue that ensembling can considerably improve the tails long before the asymptotic regime.  Is this correct?

---

> ### Author Response · Authors · 2024-11-24
>
> We thank the reviewer for the insightful questions. Please find our responses below. We highly appreciate your re-evaluation of our work and a kind reconsideration of the review score.
>
> >Comment 1: I believe that the asymptotic viewpoint undermines the theoretical result. Even without going to the refined empirical process literature, we know from Vapnik (Vapnik 1998 , Wiley) that if the function $L($ theta,z) has a finite capacity (VC dim for classification, extensions of VC dim for continuous losses with either bounded range or bounded p-th moments), we have relative uniform convergence results with asymptotically exponential tails. Hence in that setup, any reasonable base learner will also have exponential tails. On the other hand, if the VC dim is infinite, we know that it will be very difficult to get any guarantee at all (see also JMLR 11 (2010) 2635-2670). I can see therefore two ways to interpret the results of the paper:
> >>We can consider that the author proves that ensembling alone is as good as empirical risk optimization and variants in the sense that it is going to give exponential tails using possibly bad base algorithms. However this amounts to saying that averaging is optimization, and we know that averaging can be superior to optimization.
> >>We can also recall that the VC exponential bounds are often meaningless for practical data sizes. Okay they're exponential in the very large sample limit, but for anything practical, they don't look exponential. Ensembling can be a way to improve dramatically these practical tails (the experiments support this point of view). But that also means that the asymptotic perspective of the theory proposed in the paper is not the right way to look at this.
>
> **Response**: Thank you for your insightful comments. You are correct that for function classes with finite VC dimension, exponential bounds typically require strong moment conditions on the loss function. When only weak moment conditions, such as the finite p-th moment, are assumed, the convergence can indeed become polynomial.
>
> >Comment 2: Can you give a simple self-contained example of a "reasonable" base learner that provably does not have exponential tails but has polynomial ones? It seems to me that this means that you either have to consider infinite VC cases, or losses with unbounded moments, or algorithms that are far inferior to ERM in the limit. There might be a way to make a convincing case for the last option, that is, inferior to ERM in the limit, but nevertheless practically reasonable in practical settings.
>
> **Response**: Thank you for the question. Here is a simple example of a base learner with polynomial decay. Consider the univariate least squares linear regression, formulated as $\min_{\theta\in R}\mathbb E[(x\theta - y)^2]$, where the true coefficient is denoted as $\theta^*$, and the data $(x_i,y_i),i=1,...,n$ are iid samples such that $E[x_i^2]=1$, $y_i=x_i\theta^*+\epsilon_i$, where each $\epsilon_i$ is iid with zero mean. Under this setting, the coefficient estimate is given by $$\hat \theta=\frac{\sum_{i=1}^nx_iy_i}{\sum_{i=1}^nx_i^2}=\frac{\sum_{i=1}^nx_i\epsilon_i}{\sum_{i=1}^nx_i^2}+\theta^*$$, and the excess risk, therefore, is $$\left(\frac{\sum_{i=1}^nx_i\epsilon_i}{\sum_{i=1}^nx_i^2}\right)^2$$. If the distribution of $\epsilon_i$ has a polynomial tail, then the excess risk also has a polynomial tail.
>
> >Comment 3: A brief review of the proofs suggests that they might repurposed to argue that ensembling can considerably improve the tails long before the asymptotic regime. Is this correct?
>
> **Response**: We are not sure what the reviewer means by ``repurpose’’, but our theories are inherently finite-sample rather than asymptotic in that the bounds are valid for all sufficiently large $k$. For example, the bound (8) is valid as long as $k$ is reasonably large such that $\eta_{k,\delta} > 0$. Consequently, these results guarantee a finite-sample exponential tail for the excess risk, even if the base learner’s convergence rate may be slow.

---

> > ### Comment · Reviewer_aq9J · 2024-11-25
> >
> > *You are correct that for function classes with finite VC dimension, exponential bounds typically require strong moment conditions on the loss function. When only weak moment conditions, such as the finite p-th moment, are assumed, the convergence can indeed become polynomial.*
> >
> > The sufficient moment conditions described by Vapnik (1998) in sections 5.7 and section 5.8 have the form
> > $\mathop{sup}_{\theta} \frac{\sqrt[p]{E[ L(z,\theta)^p ]}}{ E[ L(z,\theta)] } < C $ for any $p>2$.
> > This is weaker than you suggest.
> >
> > *If the distribution of  has a polynomial tail, then the excess risk also has a polynomial tail.*
> >
> > Not convinced here. Let's make an innocuous simplification with $x_i=\pm1$ so that the denominator in this excess risk is deterministic.  Then you can certainly show that the expectation $E\left[ \left(\frac{\sum_{i=1}^nx_i\epsilon_i}{n}\right)^2  \right]$ decreases polynomially with $n$, but that does not tell you much about the tails. However, under not-too-crazy polynomial tail assumptions on $\epsilon$, you can also bound some of the moments of $\epsilon^2$ and then use Chernoff bounding to recover an exponential tail bound. This is roughly how the Vapnik result works.
> >
> > Note that I like your results and I believe they tell something really insightful.  I just remain unconvinced that you found the right way to formulate it. Not only I believe necessary to reformulate the theoretical section adequately, but I also believe this work will prove very fruitful.

---

> ### Author Response · Authors · 2024-11-25
>
> Thank you for the prompt reply. We also thank for your kind words on the insight and potential of our work. Here are answers to your further questions:
>
> >The sufficient moment conditions described by Vapnik (1998) in sections 5.7 and section 5.8 have the form $\sup_{\theta}\frac{(E[l(z,\theta)^p])^{1/p}}{E[l(z,\theta)]}<C$ for any $p>2$. This is weaker than you suggest.
>
> We'd like to elaborate on this example from Vapnik (1998) and argue that it actually leads to polynomial tails. In the context of Vapnik (1998), this relative finite p-th moment condition is sufficient for an exponential uniform relative convergence of the empirical loss as correctly pointed out by the reviewer. However, note that the uniform convergence there is one-sided only (e.g., equation 5.43 on page 210 in Vapnik (1998)), and this one-sided exponential convergence is possible only because the loss is assumed non-negative there and hence is effectively light-tailed on the negative side (but still heavy-tailed on the positive side and hence no exponential convergence on that side). To obtain a bound on the excess risk, Vapnik (1998) additionally needs the Bahr-Essen inequality to take care of the other side of the inequality, which introduces a polynomial decay in the tail (at the top of page 213 Vapnik (1998)). The reviewer can verify that the excess risk bound (equation 5.51 in Vapnik (1998)) is in fact polynomial by noticing that the right hand side has a polynomial dependence with an exponent of $-1/2$ on the probability parameter $\eta$. If the bound were exponential, this dependence should be logarithmic.
>
> Back to our example of univariate linear regression, we thank the reviewer for assuming $x_i=\pm1$ to simplify the analysis. Let's further assume that $\epsilon_i$ has a symmetric distribution around zero, then we have that the excess risk $\Big(\frac{\sum_{i=1}^nx_i\epsilon_i}{n}\Big)^2=\Big(\frac{\sum_{i=1}^n\epsilon_i}{n}\Big)^2$ in distribution. The latter is simply the square of sample mean of all $\epsilon_i$'s, and hence has a polynomial tail in general if each $\epsilon_i$ has a polynomial tail (See, e.g., Proposition 6.2 in Catoni, O. (2012). Challenging the empirical mean and empirical variance: a deviation study. In Annales de l'IHP Probabilités et statistiques (Vol. 48, No. 4, pp. 1148-1185), where the variable $M$ is the sample mean of iid variables).

---

> ### Comment · Reviewer_aq9J · 2024-11-25
>
> Thank you for your reply.
>
> ### 1)
>
> Proposition 6.2 in Catoni, O. (2012) is very interesting, but as clear in the proof (section 7.5), this bound is obtained for a distribution that depends on $n$, something that was not completely clear in the text of the proposition. For our purposes, we do not expect the noise distribution to change when we change the training set size.
>
> ### 2)
>
> Vapnik (1998) section 5.8 unhelpfully alternates between discussing the cases $p>2$ (for which one can prove the exponential one-sided tail 5.43) and $1<p\leq2$ for which one can prove only the weaker result (5.45) which I have not analyzed precisely enough to discuss. Note also that the one-sided relative tail of (5.43) is in the correct direction: if one minimizes the empirical risk one is sure that the expectation is not much worse.
>
> You mention (5.51), but this is not a bound on the asymptotic behavior of the tail similar your equations (2) or (3).  Instead this is a non-asymptotic bound that holds for chosen $\eta$. You surely will agree that is is much more important to explain why your ensembles have such a remarkable effect on the non-asymptotic excess error in your UCI experiments than crafting unpractically noisy cases to support the asymptotic statement. This is why I am suggesting that such non-asymoptotic considerations could be a better way to formulate your findings.

---

> ### Author Response · Authors · 2024-11-26
>
> Thank you for your comments. We hope the following responses provide further clarity and address your concerns.
>
> # 1)
> Thanks for pointing out that the distribution in Proposition 6.2 from Catoni (2012) is sample-size dependent, a fact that we overlooked while citing this result, and hence it's not the best example for justifying the polynomial tail for our linear regression example. We now provide a self-contained rigorous proof:
>
> * **Assumption**: There exist constants $C>0$ and $\alpha>0$ such that each $\epsilon_i$ satisfies $P(\epsilon_i>t)>C(t+1)^{-\alpha}$ for every $t>0$, i.e., $\epsilon_i$ has a polynomial tail. We also assume that the distribution of $\epsilon_i$ is symmetric with respect to zero.
>
> * **Theorem**: Denote by $\bar \epsilon_{n}:=\sum_{i=1}^{n}\epsilon_i /n$ the sample average of all the $\epsilon_i$'s. Under the above assumption, we have the following lower bound for the tail of the excess risk $\bar\epsilon_n^2$ of our univariate linear regression problem described in our response
> $$
> P\Big(\bar \epsilon_n^2>\delta\Big)>C(n\sqrt{\delta}+1)^{-\alpha}
> $$
> for every $\delta>0$ and $n>1$.
> * Proof: Denote by $\bar \epsilon_{k}:=\sum_{i=1}^{k}\epsilon_i / k,k\leq n$ the sample average of the first $k$ $\epsilon_i$'s, then $\bar \epsilon_n=\frac{(n-1)\bar \epsilon_{n-1}+\epsilon_n}{n}$, therefore for every $\delta>0$ and $n>1$ we have
> $$
> \begin{aligned}
> P\big(\bar \epsilon_n>\sqrt{\delta}\big)&\geq P\big(\frac{(n-1)\bar \epsilon_{n-1}}{n}\geq 0 \text{ and }\frac{\epsilon_n}{n}>\sqrt{\delta}\big) \\\\
> &=  P\big(\frac{(n-1)\bar \epsilon_{n-1}}{n}\geq 0\big)\cdot P\big(\frac{\epsilon_n}{n}>\sqrt{\delta}\big)\text{ by independence between $\bar\epsilon_{n-1}$ and $\epsilon_n$}\\\\
> &>  P\big(\bar \epsilon_{n-1}\geq 0 \big)\cdot C(n\sqrt{\delta}+1)^{-\alpha}\text{ by the above tail assumption}\\\\
> &\geq \frac{1}{2}C(n\sqrt{\delta}+1)^{-\alpha}\text{ by the symmetry assumption of each $\epsilon_i$}
> \end{aligned}
> $$
> This immediately gives the desired bound by the symmetry of $\bar \epsilon_n$.
>
> We hope this proof clearly demonstrates the fact that the tail is indeed polynomial in this example.
>
> # 2)
> We agree with the reviewer that the one-sided tail (5.43) in Vapnik (1998) is in the correct direction for bounding the expected loss of the empirical estimated model, but the resulting bound (e.g., equation 5.46 in Vapnik (1998)) will still contain heavy-tailed terms such as the empirical loss at the estimated model which ultimately leads to a polynomial tail for generalization as we argued.
>
> We also acknowledge the reviewer’s emphasis on the importance of a finite-sample theoretical framework, and our theories are exactly finite-sample as we argued before. We are concerned that there may be a misunderstanding regarding what constitutes finite-sample versus asymptotic bounds. We consider a bound to be finite-sample if it holds true and is meaningful for an arbitrary sample size or any size above a threshold that can be explicitly characterized, and to be asymptotic if it holds only in the limit as the sample size approaches infinity. By this criterion, our bound is clearly finite-sample as it gives meaningful probability bounds ($<1$) for large enough $k$ and $\delta$ such that $\eta_{k,\delta}>0$. Similarly, the excess risk bound (5.51) in Vapnik (1998) is also finite-sample as the reviewer correctly pointed out. Note that (5.51) is meaningful only for $\eta$ bounded away from zero as the bound is subscripted by $\infty$ and, according to the definition of the $\infty$ subscript right after equation (5.46), takes the trivial value $\infty$ for too small $\eta$. That is, this finite-sample bound is of the same nature as our bound in the sense that both are valid and meaningful only when the sample size or probability exceeds a threshold.

---

> ### Comment · Reviewer_aq9J · 2024-11-26
>
> ### 1)
>
> You're right (and Chernoff does not work here). Now I would be really curious to see how this particular example is cured by ensembles. Sometimes showing first a simple example makes the paper much clearer.
>
> It is also worth nothing that your experiments show ensembles working well for classification problems. For classification losses, ERM has exponential tails without doubt.  So here the argument is indirect: the argument on tails becoming exponential does not really apply directly, but is a good way to attracting the community's attention to what ensembles do to tails...
>
> ### 2)
>
> I finally see your point.  What (5.54) iin Vapnik( 1998) says is that we have exponential tails on the (one-sided relative) difference between expected and empirical errors. But to convert this into a bound on the excess error (the difference between expected error and optimal expected error), one has to bridge(*) inequalities using the empirical error which is itself subject to the high noise conditions.  So the structure of the polynomial tails is quite interesting: the generalization bound has exponential tails, but the excess error bound does not.  I had not realized that before. Thanks.
>
> (*) The bridging itself is quire complicated when dealing with one-sided bounds. See the same Vapnik book, Lemma page 82, that connects strict consistency to simple consistency (this is the part that breaks) and the argument page 91 that links one-sided bounds to strict consistency (I believe this one works).
>
> ### Conclusion
>
> I am raising the score.
>
> I am willing to raise further if you can explain how an ensemble cures the polynomial tails in the little problem above, and if you promise to start the manuscript with such a small example before developing the general argument. Providing an intuition would make the paper useful for a much broader audience (not just me, I believe.)

---

> ### Author Response · Authors · 2024-11-27
>
> Thank you for raising the score and thank you for your continuous involvement in the discussion with us. The bridging argument you referenced from Vapnik (1998) is indeed an excellent demonstration of the technical approach to addressing the other side of the inequality: Upper bound the empirical loss at the estimated model by leveraging its optimality and substituting it with the empirical loss at another (fixed) reference model. Specifically, when bounding the excess risk, this reference model is one that optimizes the expected loss.
>
>
> We are glad to provide further details on the linear regression example, illustrating both the polynomial tail of the base learner and the exponential tail of our ensemble method. Below is a draft for you to review. We will include this in the revised paper as soon as possible if you believe it enhances the clarity of our main results.
>
> For simplicity, we slightly modify the previous example and consider the constrained univariate least squares linear regression. The problem can be formulated as $\min_{\theta\in [-1,1]}\mathbb E[(x\theta - y)^2]$, and the data $(x_i,y_i),i=1,...,n$ are iid samples such that $x_i\in\\{-1,1\\}$, $y_i=x_i\theta^*+\epsilon_i$. The unknown true coefficient is $\theta^*=0$, and each $\epsilon_i$ is iid with zero mean and symmetrically distributed with respect to $0$. Under this setting, the coefficient estimate by least-squares linear regression is
>
> $$
> \hat \theta=\mathcal P_{[-1,1]}\left(\frac{\sum_{i=1}^nx_iy_i}{\sum_{i=1}^nx_i^2}\right)=\mathcal P_{[-1,1]}\left(\frac{\sum_{i=1}^nx_i\epsilon_i}{\sum_{i=1}^nx_i^2}+\theta^*\right)=\mathcal P_{[-1,1]}\left(\frac{\sum_{i=1}^nx_i\epsilon_i}{n}\right)
> $$
> where $\mathcal P_{[-1,1]}$ denotes the projection operator onto the interval $[-1,1]$. The excess risk can thus be expressed as
> $$
> \hat \theta^2=\min\left(\left(\frac{\sum_{i=1}^nx_i\epsilon_i}{n}\right)^2,1\right).
> $$
>
>
> # Polynomial tail for the excess risk
> In the following, we first show that if the distribution of $\epsilon_i$ has a polynomial tail, then the excess risk tail is also polynomial in $n$.
>
> **Assumption**: There exist constants $C>0$ and $\alpha>0$ such that each $\epsilon_i$ satisfies $P(\epsilon_i>t)>C(t+1)^{-\alpha}$ for every $t>0$, i.e., $\epsilon_i$ has a polynomial tail. The distribution of $\epsilon_i$ is symmetric with respect to zero.
>
> **Theorem**: Under the above assumption, we have the following lower bound for the tail of the excess risk
> $$\mathbb P\Big(\hat \theta^2>\delta\Big)>C(n\sqrt{\delta}+1)^{-\alpha},$$
> for every $\delta\in (0,1)$ and $n>1$.
>
> **Proof**: Denote by $\bar \epsilon_{k}:=\sum_{i=1}^{k}\epsilon_i / k,k\leq n$ the sample average of the first $k$ noise terms $\epsilon_i$'s. Then, we have that $\bar \epsilon_n=\frac{(n-1)\bar \epsilon_{n-1}+\epsilon_n}{n}$. Therefore, for every $\delta\in (0,1)$ and $n>1$, we have that
> $$
> \begin{aligned}
> \mathbb P\left(\hat \theta^2>\delta\right)&=\mathbb P\left(\left(\frac{\sum_{i=1}^nx_i\epsilon_i}{n}\right)^2>\delta\right)\text{ since $\delta<1$}\\\\
> &=\mathbb P\left(\left(\frac{\sum_{i=1}^n\epsilon_i}{n}\right)^2>\delta\right)\text{ by the symmetry of $\epsilon_i$ and that $x_i\in\\{-1,1\\}$}\\\\
> &=2\mathbb P\big(\bar \epsilon_n>\sqrt{\delta}\big)\text{ again by the symmetry of $\epsilon_i$}\\\\
> &\geq P\left(\frac{(n-1)\bar \epsilon_{n-1}}{n}\geq 0 \text{ and }\frac{\epsilon_n}{n}>\sqrt{\delta}\right) \\\\
> &= 2P\left(\frac{(n-1)\bar \epsilon_{n-1}}{n}\geq 0\right)\cdot P\left(\frac{\epsilon_n}{n}>\sqrt{\delta}\right)\text{ by independence between $\bar\epsilon_{n-1}$ and $\epsilon_n$}\\\\
> &>  2P\big(\bar \epsilon_{n-1}\geq 0 \big)\cdot C(n\sqrt{\delta}+1)^{-\alpha}\text{ by the above tail assumption}\\\\
> &\geq C(n\sqrt{\delta}+1)^{-\alpha}\text{ by the symmetry of $\epsilon_i$}.
> \end{aligned}
> $$
> This concludes the proof.

---

> ### Author Response · Authors · 2024-11-27
>
> # Exponential tail by our method
> Now, we show that an exponential tail can be achieved under our ensemble method. Denote by $\sigma^2:=\mathbb E[\epsilon_i^2]$ and $\mu:=\mathbb E[\epsilon_i^4]$. Since this is a continuous problem, we can apply our bound in Eq. (10) of the paper. To do so, we need to derive upper bounds for the following two quantities.
>
> - The empirical process tail $T_k$ from Theorem 2. For this problem, the tail can be bounded for $t>0$ as
> $$
> \begin{aligned}
> T_k(t)&=\mathbb P\Big(\sup_{\theta\in [-1,1]} \big| \frac{1}{k}\sum_{i=1}^k(x_i\theta-y_i)^2 - \mathbb E[(x\theta-y)^2]\big|>t \Big)\\\\
> &=\mathbb P\Big(\sup_{\theta\in [-1,1]} \big| \frac{1}{k}\sum_{i=1}^k(x_i\theta-\epsilon_i)^2 - (\theta^2 + \sigma^2)\big|>t \Big)\\\\
> &=\mathbb P\Big(\sup_{\theta\in [-1,1]} \big| \frac{1}{k}\sum_{i=1}^k\epsilon_i^2-\sigma^2 - 2\theta\cdot \frac{1}{k}\sum_{i=1}^k x_i\epsilon_i\big|>t \Big)\\\\
> &\leq \mathbb P\Big(\big| \frac{1}{k}\sum_{i=1}^k\epsilon_i^2-\sigma^2 - \frac{2}{k}\sum_{i=1}^k x_i\epsilon_i\big|>t \Big)+\mathbb P\Big(\big| \frac{1}{k}\sum_{i=1}^k\epsilon_i^2-\sigma^2 + \frac{2}{k}\sum_{i=1}^k x_i\epsilon_i\big|>t \Big)\\\\
> &\ \ \ \ \text{ by union bound and that the maximizing $\theta$ is either $-1$ or $1$}\\\\
> &= 2\mathbb P\Big(\big| \frac{1}{k}\sum_{i=1}^k\epsilon_i^2-\sigma^2 - \frac{2}{k}\sum_{i=1}^k\epsilon_i\big|>t \Big)\text{ by symmetry of $\epsilon_i$ and that $x_i\in\\{-1,1\\}$}\\\\
> &\leq 2\mathbb P\Big(\big| \frac{1}{k}\sum_{i=1}^k\epsilon_i^2-\sigma^2\big| >\frac{t}{2}\Big)+2\mathbb P\Big( \big|\frac{2}{k}\sum_{i=1}^k\epsilon_i\big|>\frac{t}{2} \Big)\text{ by union bound}\\\\
> &\leq \frac{8\mu}{kt^2}+\frac{32\sigma^2}{kt^2}\text{ by Markov's inequality}.\\\\
> \end{aligned}
> $$
> - The excess risk tail of constrained least squares. For $\delta<1$, this can be bounded as
> $$
> \mathcal E_{k,\delta} = \mathbb P\Big(\hat \theta^2>\delta\Big)=\mathbb P\Big(\Big( \frac{\sum_{i=1}^k\epsilon_i}{k}\Big)^2>\delta\Big)\leq \frac{\sigma^2}{k\delta},
> $$
> by Markov's inequality.
>
> Now, assuming $\epsilon_i$ has finite fourth-order moment and instantiating the $T_k$ and $\mathcal E_{k,\delta}$ in Theorem 2 with the above upper bounds, we can apply our bound (10) to obtain the following tail bound for the excess risk of our method in the constrained least squares problem
> $$
> \begin{aligned}
> \mathbb P\Big( \hat \theta^2>\delta \Big)&\leq B_1\Big(3\min\big(e^{-2/5},C_1\frac{32(\mu+4\sigma^2)}{k_2\min(\underline{\epsilon},\delta-\overline{\epsilon})^2}\big)^{\frac{n}{2C_2k_2}}+e^{-B_2/C_3}\Big)\\\\
> &+\min\Big( e^{-\big(1-\frac{\sigma^2}{k_1\delta}\big)/C_4},C_5\frac{\sigma^2}{k_1\delta} \Big)^{\frac{n}{2C_6k_1}}+e^{-B_1(1-\frac{\sigma^2}{k_1\delta})/C_7}
> \end{aligned}
> $$
> for every $\delta\in (0,1)$ and $k_1,k_2$ such that $\delta>\overline{\epsilon}$, $\frac{32(\mu+4\sigma^2)}{k_2(\delta-\overline{\epsilon})^2}+\frac{32(\mu+4\sigma^2)}{k_2\underline{\epsilon}^2}<\frac{1}{5}$ (corresponding to the condition $T_{k_2}((\delta-\overline{\epsilon})/2) + T_{k_2}(\underline{\epsilon}/2)<1/5$ in Theorem 2) and $\frac{\sigma^2}{k_1\delta}<1$ (to make the upper bound of the base risk tail meaningful).

---

> ### Comment · Reviewer_aq9J · 2024-11-29
>
> I must say it does not come as nicely as I hoped.
>
> 1) My intuition was that randomly selecting many subsets of the original data gives a good chance to have a number of subsets whose examples experience benign noise only (that is, not outliers.).  Applying the base estimator to these subsets is more likely to return similar $\theta$s than applying it to subsets with outliers examples.  A heavy-tailed noise problem is a fundamentally a problem with outliers.
>
> 2) Then you also have experiments on classification problems (for which the base estimator has exponential tails already). One could still have a notion of outliers here, such as examples that fall far on the wrong side of the Bayes boundary (e.g. badly mislabeled examples). The same intuition applies it seems.
>
> As you can see, I still believe that the exponential vs non-exponential tail is not the fundamental thing here. It cannot be in the case of classification losses anyway.  I'll raise my score because I value papers that make us think. That said, I also encourage you to keep searching because I am convinced that this setup does not capture the essence yet.

---

> > ### Author Response · Authors · 2024-11-30
> > **Thank you!**
> >
> > Thank you for further raising the score and for sharing your valuable insights. We appreciate your perspective on the broader potential of our method beyond transforming polynomial tails into exponential ones. Indeed, we also believe our method is grounded in more fundamental principles.
> >
> > We would like to share some initial thoughts on interpreting our method from the perspective of **mode estimation**. In the case of a discrete model space, if we consider the base model trained on a random sample as a statistic that takes values in the model space, then our method—which selects the model that occurs most frequently among all subsampled models—amounts to estimating the mode of the sampling distribution via subsampling. Similar observations apply to the continuous case as well. In contrast, other subsampling-based ensemble methods like subagging can be viewed as estimating the mean of the sampling distribution. The tail advantage of our method can then be explained by the fact that **mode estimation relies on data frequency rather than magnitude, making it generally more robust against outliers than mean estimation**.
> >
> > Thank you again for sharing the insights. We are encouraged by your feedback and will continue to explore these fundamental aspects to better understand and articulate the core principles underlying our method.

---

> > > ### Comment · Reviewer_aq9J · 2024-11-30
> > >
> > > Your mode estimation insight is indeed similar to the intuition described in my earlier comment. Not the mode of the data distribution, but the mode of the distribution of the estimated parameter obtained using training set replicas that may be lucky eliminating outliers. The idea being replicas with less outliers will yield more narrowly clustered estimates.
> > >
> > > I regret that this important and intuitive insight does not how in the paper.

---

> > > > ### Author Response · Authors · 2024-12-01
> > > >
> > > > Indeed, our mode estimation insight aligns closely with the intuition you described. This is an important perspective that we should have addressed in the paper. In the revised manuscript, we plan to include this discussion, along with the illustrative example of least squares linear regression, to make the paper more accessible and engaging for a broader audience. Thank you once again for highlighting this point and for your constructive feedback.

---

### Official Review · Reviewer_NNfa · 2024-11-03

**Soundness:** 2
**Presentation:** 2
**Contribution:** 2
**Rating:** 6
**Confidence:** 3

**Summary:**

This paper presents a method for ensemble learning. After training B models on subsampled data from the training set, the authors propose two strategies (MoVE and ROVE / ROVEs) to select the best model from the B ensemble members. The authors proved that their method can achieve an exponentially decaying rate for excess risk. Experiments results on synthetic data and six datasets from UCI repository validate the proposed method’s performance in out-of-sample testing.

**Strengths:**

The authors propose to study an important problem in ensemble learning.

The idea of using majority voting to select the best model in ensemble learning seems interesting.

**Weaknesses:**

=============

After rebuttal:

Thank you to the authors for their further response. I appreciate the linear regression example provided in the discussion with Reviewer aq9J. The derivation on the polynomial tail in the excess risk of the base learner and the exponential tail of the proposed method helps me better understand the technical contribution of the work and addresses my previous concerns about the base learner. Additionally, the discussion on mode estimation in models sampling is insightful. I would recommend including this discussion in the final paper, as it strengthens the rationale behind the proposed ensemble method. Taking all this into account, I believe the theoretical contributions in this work outweigh limitations in experiments. I am happy to raise my score.

=============

I have the following concerns/questions regarding the work:

1. Practicality of Theorem Assumptions: in Theorem 1, Eq. (9) is under the assumption that $n_{k, \delta} > 4/5$, which indicate that $p_k^{\text{max}} > 4/5$. This assumption appears quite strong, as it requires the most frequently subsampled model to have a subsample rate above 80%. Could the authors clarify the rationale behind this requirement?

2. In contrast to MoVE which seems to be based on a strong assumption, ROVE/ROVEs presents a more practical strategy for selecting the best model. However, the practical insights of Theorem 2 regarding the excess risk of ROVE/ROVEs remain somewhat unclear. Could the authors provide more detailed practical insights into Theorem 2?

Regarding the implementation details and evaluation, I have the following additional questions:

3. In line 193, the authors mention that "Therefore, $n_{k, \delta}$ taking large values signals the situation where the base learner already generalizes well." However, this statement depends not only on the base learner but also on the choice of $\delta$, i.e., a larger $\delta$ leads to a smaller $\mathcal{E}\_{k,\delta}$, thus a larger $n_{k, \delta}$. Could the authors provide further clarification on this claim?

4. In line 271, the authors discussed the choice of $\epsilon$ as "In our experiments, we find it a good strategy to choose an ε that leads to a maximum likelihood around 1/2." Could the authors elaborate further on this?

5. Details of the Base Learning Algorithm $\mathcal{A}$: The paper assumes that the ensemble members are trained using a generic learning algorithm $\mathcal{A}$ based on the subsampled data. Could the authors clarify details about $\mathcal{A}$? For instance, would the settings in $\mathcal{A}$ (such as learning rate, batch size in gradient-based methods) impact the performance of the proposed framework?

6. Experimental Validation and Comparisons: The current experimental results may be insufficient to support the claims fully. Firstly, one of the main contributions is the proof of an exponential decay rate of excess risk; could the authors present numerical results to validate this aspect? Secondly, the current experiments focus only on comparisons between the proposed method (MoVE and ROVE / ROVEs) and the base model. Including additional ensemble methods, as discussed in Section 4, would strengthen the experimental comparisons. Finally, the appendix mentions that the ensemble members use MLPs with 2-8 hidden layers; it would be helpful to explore whether the framework is applicable to other architectures, such as CNNs or transformers.

**Questions:**

It would be helpful if the authors could address my questions regarding the practicality, implementation, and evaluation of the proposed method, as detailed above.

---

> ### Author Response · Authors · 2024-11-24
>
> We thank the reviewer for the insightful questions. Please find our responses below. We highly appreciate your re-evaluation of our work and a kind reconsideration of the review score.
>
> >Comment 1: In Theorem 1, Eq. (9) is under the assumption that $\eta_{k, \delta}>4 / 5$, which indicates that $p_k^{\max}>4 / 5$. This assumption appears quite strong, as it requires the most frequently subsampled model to have a subsample rate above $80 \%$. Could the authors clarify the rationale behind this requirement?
>
> **Response**: We would like to clarify that $\eta_{k, \delta}>4 / 5$ is not a fundamental assumption of our theorem but a condition under which our complexity bound in Eq. (8) simplifies to Eq. (9). In particular, it does not affect the exponential tail of our method. Indeed, as long as $\eta_{k, \delta}$ is strictly greater than zero, our bound in Eq. (8) decays exponentially fast in $n/k$.
>
> >Comment 2: In contrast to MoVE which seems to be based on a strong assumption, ROVE/ROVEs presents a more practical strategy for selecting the best model. However, the practical insights of Theorem 2 regarding the excess risk of ROVE/ROVEs remain somewhat unclear. Could the authors provide more detailed practical insights into Theorem 2?
>
> **Response**: We would like to clarify that the main practical implication of Theorem 2 is that ROVE and ROVEs achieve an exponential decay in the generalization tail for general model space—whether it’s discrete, continuous, or a mixture of both. We will emphasize this insight more clearly in the revised paper. Please feel free to let us know if there is any specific aspect that you would like us to address. We would be happy to provide further clarification.
>
> >Comment 3: In line 193, the authors mention that "Therefore, $\eta_{k, \delta}$ taking large values signals the situation where the base learner already generalizes well." However, this statement depends not only on the base learner but also on the choice of $\delta$, i.e., a larger $\delta$ leads to a smaller $\mathcal E_{k,\delta} $, thus a larger $\eta_{k,\delta}$. Could the authors provide further clarification on this claim?
>
> **Response**: We consider $\delta$ as a fixed parameter in this context. Accordingly, for a given $\delta$, a base learner with a larger value of $\eta_{k,\delta}$ is regarded as having better generalization performance than learners with smaller $\eta_{k,\delta}$.
>
> >Comment 4: In line 271, the authors discussed the choice of $\epsilon$ as "In our experiments, we find it a good strategy to choose an $\epsilon$ that leads to a maximum likelihood around $1 / 2$." Could the authors elaborate further on this?
>
> **Response**: We apologize for the confusion. We have provided details on the choice of $\epsilon$ in Appendix D.3 (lines 1821-1831), and we will add a pointer to this discussion around line 271 in the revised paper.
>
> >Comment 5: The paper assumes that the ensemble members are trained using a generic learning algorithm $\mathcal{A}$ based on the subsampled data. Could the authors clarify details about $\mathcal{A}$ ? For instance, would the settings in $\mathcal{A}$ (such as learning rate, batch size in gradient-based methods) impact the performance of the proposed framework?
>
> **Response**: Our theory abstracts away the specific details of the base learner because its influence on our method’s performance is primarily through its own generalization performance—specifically, the probability $p_k^{\max}$ and the generalization tail $\mathcal{E}_{k,\delta}$ as shown in bounds (8) and (9). While factors like learning rate and batch size can affect the ultimate performance, the impact is only through their effect on the base learner's generalization performance.

---

> > ### Author Response · Authors · 2024-11-24
> >
> > >Comment 6: Experimental Validation and Comparisons: The current experimental results may be insufficient to support the claims fully. Firstly, one of the main contributions is the proof of an exponential decay rate of excess risk; could the authors present numerical results to validate this aspect? Secondly, the current experiments focus only on comparisons between the proposed method (MoVE and ROVE / ROVEs) and the base model. Including additional ensemble methods, as discussed in Section 4, would strengthen the experimental comparisons. Finally, the appendix mentions that the ensemble members use MLPs with 2-8 hidden layers; it would be helpful to explore whether the framework is applicable to other architectures, such as CNNs or transformers.
> >
> > **Response**: We’ve already provided rigorous proof for the exponential decay. For the empirical study, directly verifying the exponential decay is challenging due to the nature of the considered generalization tail: Firstly, it’s a stochastic target whose estimation requires repeated runs of our method on independent datasets; secondly, because the value of this target decays exponentially, estimating such rare events becomes challenging. Instead, we demonstrate the superior tail performance through comparison of empirical distribution functions of the out-of-sample costs (e.g., Figure 1(c)(f), Figure 2, and Figure 3(f)).
> >
> > Also, thank you for the suggestions. We will include more experiments in the revised paper. In particular, we have compared our algorithm with the traditional bagging approach, and the results can be found in **https://anonymous.4open.science/r/vote_ensemble/ComparisonWithBagging.pdf**. From the figure, we observe that our methods demonstrate significant advantages as the tail of noises goes heavier.
> >
> > Regarding the suggested advanced architectures, we appreciate the recommendation. Since our approach is a general ensemble method applicable to any base learner, we focused our experiments on the essential aspects of the approach. This includes demonstrating its effectiveness for various problems (regression, stochastic programming) and base learners (MLPs, linear regression, SAA, DRO), as well as analyzing its performance dependence on tail heaviness and the bias and variance of the base learner. While further experiments using the suggested architectures are certainly interesting, we believe they may not provide additional insights into these core aspects.

---

> > > ### Comment · Reviewer_NNfa · 2024-11-25
> > > **Reply to author rebuttal**
> > >
> > > I'd like to thank the authors for your rebuttal. While some of my questions have been addressed, several key concerns still remain: firstly, regarding the base learner $\mathcal{A}$, I didn't quite get the response of "While factors like learning rate and batch size can affect the ultimate performance, the impact is only through their effect on this output distribution." Since the output distribution affects the method's performance, it's important to have a deeper understanding of how the output distributions are affected by variations in the settings of $\mathcal{A}$. This would provide valuable insights into when the proposed method performs well and when it might face limitations. Secondly, regarding the comparison with more recent methods, I still believe it's important especially the authors claim that the proposed method is a general method applicable to any base learner. This concern is also shared by reviewer wArH.
> > >
> > > Considering these concerns, I'll keep my original score.

---

> > > > ### Author Response · Authors · 2024-11-25
> > > >
> > > > Thanks for the prompt reply. Sorry that the phrase "output distribution of the base learner" might be a confusing description of what ultimately affects our method's performance. To be more precise, it is the generalization performance, and nothing else, of the base learner that has a direct impact on the performance since our theoretical bounds (8), (9) and (10) depend on the base learner only through $p_k^{\max}$ and $\mathcal E_{k,\delta}$. We have revised our response above to make this clear. One of the key messages of the paper is that the amount of improvement brought by our method depends on how fast the generalization tail of the base learner decays in the sample size $n$ (i.e., most beneficial for polynomial decay, less beneficial for exponential decay).
> > > >
> > > > Regarding comparison with other ensemble methods, we fully understand the importance. However, there are many ensemble methods out there designed to enhance different aspects of the base model. Our proposal aims at the new aspect of improving the tail of generalization performance that has not been tackled by existing ensemble methods, which is why we didn't include comparison with existing ensemble methods in the manuscript. We decided to include comparison with bagging upon the reviewer's request because 1) procedure-wise it's similar to our method in ensemble construction as explained in our response and 2) it's the ensemble method that is most widely known to reduce variance (a benefit conceptually related to tail improvement but not exactly the same thing mathematically speaking). Therefore, it will be great if the reviewer could explicitly point out what ensemble methods you'd like us to compare with and why, so that we can do a more meaningful comparison.

---

> > > > > ### Author Response · Authors · 2024-11-27
> > > > > **Do our responses provide further clarify?**
> > > > >
> > > > > Thank you again for your insightful questions. We hope our additional clarifications have addressed your concerns, and we are prepared to include these changes in the revised manuscript. Furthermore, as suggested by **Reviewer aq9J**, we have demonstrated the effectiveness of our method in a straightforward linear regression example (please refer to our responses to **Reviewer aq9J**), i.e., the base learner $\mathcal{A}$ is an ordinary least-squares linear regression. We show that while this base learner can have a polynomial tail, our ensemble method is able to achieve an exponential tail. We hope this example more clearly illustrates our results, and we are glad to add this example to the revised paper.
> > > > >
> > > > > We are fully open to discussing any further concerns you might have. If you believe further experiments are necessary, we would appreciate your guidance on what specific aspects you think are crucial for these experiments. We sincerely hope for a reconsideration of the review score and eagerly await your feedback : )

---

### Official Review · Reviewer_wArH · 2024-11-04

**Soundness:** 3
**Presentation:** 3
**Contribution:** 3
**Rating:** 6
**Confidence:** 2

**Summary:**

This paper presents an ensemble learning technique that enhances the generalization performance by selecting the best models trained on subsamples via majority voting, resulting in exponentially decaying tails ($C_{2}\gamma^{n/k}$) for excess risk, even when base laerners have slow (polynomial i.e. $C_{1}n^{-\alpha}$) decay rates. The method is agnostic to the underlying base learner  and offers a stronger improvement than traditional variance reduction, providing rate improvement rather than just a constant factor improvement. Experimetns show that the method also effective in scenarios with heavy-tailed data and slow convergence rates.

**Strengths:**

The paper presents a fresh view on ensemble learning, focusing on exponential improvement in generalization tail bounds rather than the traditional variance reduction approach.

The paper offers valuable theoretical contributions, e.g. developing a sharper concentration result for U-statistics with binary kernels, learners, performing a sensitivity analysis on the regret, and developing a uniform law of large numbers (LLN) for the class of events of being $\epsilon$-optimal.

Theoretical insights are complemented by experiments on both real and synthetic data using neural networks and stochastic programs, demonstrating relevance of the theory.

**Weaknesses:**

**Only one baseline used**: The paper could benefit from including additional baselines beyond just the 'base' model. Since the authors mention variance reduction, it would be useful to incorporate it as a baseline, as well as to provide more detailed comparisons with state-of-the-art ensemble methods across a wider range of datasets and problem types.

**Section 3.1**: The paper only addresses regression problems, using synthetic and real data from UCI. Why not include classification tasks like CIFAR-10, CIFAR-100, or ImageNet? Does the proposed method work for these? If it doesn’t, could you explain why that wasn’t mentioned?

The models used appear to be quite small, with only a few-layer MLPs employed. Exploring more advanced architectures, such as ResNet or WideResNet, could enhance the practical applicability of the findings.

**Real data is too old**:  UCI Machine Learning Repository was launched in 1987 and is too old a benchmark. How about considering latest more challenging regression benchmarks e.g. OpenML Benchmark Datasets (2013), PMLB (Penn Machine Learning Benchmark) (2018) etc.?

**Section 3.2**: The choice using SSA isn’t clearly justified. What is the reason behind this choice? How about doing an abaltion on other methods e.g.  Distributionally Robust Optimization (DRO), RO etc.?

**Computational cost not discussed**: The paper doesn’t clearly address the computational cost of the proposed method, which could be a major concern for large-scale applications.

**Questions:**

**1)** What does "majority vote" mean for the models? What does it imply in practice? How can models trained on different subsets of data end up producing the same outputs that allow for voting, as seen in the final lines of Alg-1 and Alg-2? In standard ensemble methods, majority voting is based on the predictions of the models. I might not be familiar with this specific approach to ensembling, so I would appreciate a clearer explanation.

**2)** In Algorithm 2 line 231-233, how to get $\theta_{\prime}$ in practice?

**3)** For more questions, see Weaknesses.

---

> ### Author Response · Authors · 2024-11-24
>
> We thank the reviewer for the insightful questions. Please find our responses below.
>
> >Comment 1: Additional baseline, such as variance reduction and state-of-the-art ensemble methods. Why not include classification tasks like CIFAR-10, CIFAR-100, or ImageNet? The models used appear to be quite small, with only a few-layer MLPs employed. Exploring more advanced architectures, such as ResNet or WideResNet, could enhance the practical applicability of the findings. How about considering the latest more challenging regression benchmarks e.g. OpenML Benchmark Datasets (2013), PMLB (Penn Machine Learning Benchmark) (2018) etc.?
>
> **Response**: Thank you for the suggestion. We will consider including more real-world datasets in the revised paper and conducting additional comparisons with other benchmarks. In particular, we have compared our algorithm with the traditional bagging approach, and the results can be found in **https://anonymous.4open.science/r/vote_ensemble/ComparisonWithBagging.pdf**. From the figure, we observe that our methods demonstrate significant advantages as the tail of noises become heavier.
>
> We have not included classification tasks in our experiments because classifiers are prevalently trained with the cross-entropy loss, which is unlikely to be heavy-tailed due to the presence of the logarithm in the loss function. Therefore, our approach is anticipated to be more beneficial for regression than for classification. However, we are happy to include some classification tasks if the reviewer believes it would be beneficial.
>
> Regarding the suggested additional datasets and advanced architectures, we appreciate the recommendation. Since our approach is a general ensemble method applicable to any base learner, we focused our experiments on the fundamental aspects of the method. This includes demonstrating its effectiveness across various problems (regression, stochastic programming) and base learners (MLPs, linear regression, SAA, DRO), as well as analyzing its performance dependence on tail heaviness and the bias and variance of the base learner. While further experiments using the suggested datasets and architectures are certainly interesting, we believe they may not provide additional insights into these core aspects. Nonetheless, we are open to including them if it would strengthen the paper.
>
> >Comment 2: The choice using SSA isn’t clearly justified. What is the reason behind this choice? How about doing an ablation on other methods e.g. Distributionally Robust Optimization (DRO), RO etc.?
>
> **Response**: We chose SAA because it is a classical and effective algorithm for data-driven stochastic programs. Additionally, we also conducted experiments using DRO as the base learner (see Figure 16).
>
> >Comment 3: The paper doesn’t clearly address the computational cost of the proposed method, which could be a major concern for large-scale applications.
>
> **Response**: We would like to provide further clarifications. We note that method does not always incur a higher cost than the base learner. To explain, our theory caps the necessary ensemble size $B$ at an order of $O(n/k)$. Let $C(n)$ denote the computational cost of training a base learner with $n$ samples. Then, with $B=O(n/k)$, the total cost of our method is $B*C(k) = O(n\cdot C(k)/k)$, while the base learner applied to the full sample has the cost $C(n)$. Therefore, when $C(\cdot)$ grows superlinearly, our method is computationally cheaper (see Figure 4(b) for such an example with two-stage stochastic programming); When $C(\cdot)$ grows sublinearly, the base learner is cheaper; When the growth is linear, the two are comparable. We will add this discussion to the paper. Additionally, since our ensemble is constructed from independently drawn subsamples, the training can be easily parallelized to accelerate computation as pointed out in line 458.

---

> > ### Author Response · Authors · 2024-11-24
> >
> > >Comment 4: What does "majority vote" mean for the models? What does it imply in practice? How can models trained on different subsets of data end up producing the same outputs that allow for voting, as seen in the final lines of Alg-1 and Alg-2? In standard ensemble methods, majority voting is based on the predictions of the models. I might not be familiar with this specific approach to ensembling, so I would appreciate a clearer explanation.
> >
> > **Response**: The vote in our context applies to models themselves rather than their predictions. We would like to clarify this idea in the context of discrete optimization. Suppose we have a finite decision space $\{x_1, \cdots, x_m\}$. Each time, a base learner outputs (votes for) one of these points as a solution estimate. In the final aggregation step, we simply select the solution that receives the highest number of votes across all base learners. The same concept applies to the continuous case; however, due to the infinite size of the solution space, we include a model retrieval phase to first build a discrete approximation of the region around optimal solutions.
> >
> > >Comment 5: In Algorithm 2 line 231-233, how to get θ′  in practice?
> >
> > **Response**: Note that $\mathcal{S}$ is only a finite space containing the set of retrieved models from Phase I. Thus, solving the program in Phase II is straightforward via enumeration (see lines 215-239).

---

> ### Comment · Reviewer_wArH · 2024-11-25
>
> Thank you for the detailed response. I appreciate clarifications about the method, the setting and computational cost. However, I still believe that using newer and more challenging benchmarks is important, and comparing with more recent and advanced ensemble methods is also crucial.
>
> Therefore, I have decided to maintain my original score.

---

> > ### Author Response · Authors · 2024-11-25
> >
> > Thanks for the prompt reply. We fully understand the importance of comparing with related ensemble methods. However, there are many ensemble methods out there designed to enhance different aspects of the base model. Our proposal aims at the new aspect of improving the tail of generalization performance that has not been tackled by existing ensemble methods, which is why we didn't include comparison with existing ensemble methods in the manuscript. We decided to include comparison with bagging upon the reviewer's request because 1) procedure-wise it's similar to our method in ensemble construction as explained in our response and 2) it's the ensemble method that is most widely known to reduce variance (a benefit conceptually related to tail improvement but not exactly the same thing mathematically speaking). Therefore, it will be great if the reviewer could explicitly point out what ensemble methods you'd like us to compare with and why, so that we can do a more meaningful comparison.

---

### Official Review · Reviewer_GL5r · 2024-11-06

**Soundness:** 2
**Presentation:** 1
**Contribution:** 2
**Rating:** 3
**Confidence:** 3

**Summary:**

This paper presents a novel approach to ensemble learning called MoVE (Majority Vote Ensembling) and ROVE (Retrieval and ϵ-Optimality Vote Ensembling). The approach leverages subsampling to create ensembles that can produce exponentially decaying tails for excess risk, overcoming the polynomial decay limitations typically faced in heavy-tailed distributions. By iteratively training on subsets of the data and combining models through majority voting or likelihood-based selection, the method aims to provide stronger generalization than traditional ensembling techniques, especially in contexts with heavy-tailed data distributions. Empirical evaluations demonstrate improved out-of-sample performance across synthetic and real datasets and various stochastic programming problems.

**Strengths:**

- Provide a new approach to ensemble learning with exponentially decaying tails for excess risk.
- The paper presents experimental evaluations to support the theoretical claims.

**Weaknesses:**

- The theoretical results are quite dense and hard to parse. Need to provide example instantiations to allow the reader to get a better sense of them.
- The proposed algorithms look quite computationally expensive and yet underperform the baseline on real-world tasks.

**Questions:**

- What happens in the case that the data is not heavy-tailed? Do the gains from MoVE and ROVE become constant factor as before? Can you provide the bounds and the correct selection of k for this case?
- It appears that for the method to work, k (sub-sampled data) has to be selected at a lower rate than n which implies that the ensemble size (B = O(n/k)) will be very large. How do existing generalization bounds for ensembles scale with the ensemble size and what are the consequences for them when B is not constant but rather a function of n?
- It seems like in the real-world dataset settings, ROVE is underperforming the base method. Can you detail out why this is the case and what assumptions seem to be breaking?
- Would it be possible to test out this method with slightly more real world datasets (like MNIST, CIFAR and converting them to a regression problem)? I would like to see the convergence rate of ROVE when compared with traditional ensembles as a function of the number of learners in the ensemble.
- Theorems 1 and 2 are quite dense and hard to parse. Would it be possible to instantiate both the theorems for a few example function classes and loss functions and have them as corollaries?
- In the discrete setting, what is the dependence of the generalization error on the size of the parameter class for the traditional ensembling methods?

---

> ### Author Response · Authors · 2024-11-24
>
> We thank the reviewer for the insightful questions. Please find our responses below. We highly appreciate your re-evaluation of our work and a kind reconsideration of the review score.
>
> >Comment 1: Theoretical results are dense. Would it be possible to instantiate both the theorems for a few example function classes and loss functions and have them as corollaries?
>
> **Response**: The key message of our theory is the exponential decay of the excess risk tail in $n/k$ as highlighted in the introduction. This key message is already clearly shown by the theorems as pointed out in line 183 following Theorem 1. While we appreciate the suggestion to exemplify our theory with specific function classes and loss functions, we feel that doing so may not enhance clarity. Substituting the quantities $p_k^{\max}$ and $\mathcal{E}_{k,\delta}$  with their explicit forms for particular base learners would introduce additional complexity to the bounds.
>
> >Comment 2: What happens in the case that the data is not heavy-tailed? Do the gains from MoVE and ROVE become constant factor as before? Can you provide the bounds and the correct selection of k for this case?
>
> **Response**: We would like to clarify that our bounds in Theorems 1 and 2 are applicable regardless of whether the data distribution is heavy-tailed or light-tailed. In the light-tailed case, we have empirically observed improvements of ROVE over the base learner in many instances (see Figures 1d–1f, Figures 10d–10f, and Figure 14), although a formal theoretical comparison is less straightforward than in the heavy-tailed case as the base learner already exhibits exponential decay under light-tailed distributions.
>
> >Comment 3: Test the method with more real world datasets (like MNIST, CIFAR) to see the convergence rate of ROVE when compared with traditional ensembles as a function of the number of learners in the ensemble.
>
> **Response**:  Thank you for your suggestion. We have added an experiment comparing our method with bagging under varying degrees of tail heaviness. In this experiment, both our method and bagging use the same multilayer perceptron as the base learner and have the same number of base learners in the ensemble. The results can be found in **https://anonymous.4open.science/r/vote_ensemble/ComparisonWithBagging.pdf**. From the figure, we observe that our method demonstrates significant advantages as the noise tail becomes heavier. We will also consider including more real-world datasets in the revised paper.
>
> >Comment 4: The proposed algorithms look quite computationally expensive and yet underperform the baseline on real-world tasks. Can you detail out why this is the case and what assumptions seem to be breaking?
>
> **Response**: Regarding the performance of our algorithms, we would like to clarify that our most general and empirically strongest algorithm ROVE never underperforms the baseline by a discernible amount on any of the considered tasks. The algorithm ROVEs does appear to underperform on real-world tasks. This is mainly due to data splitting used in ROVEs, which reduces the amount of data available for training, as explained in lines 378–380. Therefore, as noted in lines 460–462, we recommend ROVE over ROVEs for general use.
>
> As for the computational cost, we note that method does not always incur a higher cost than the base learner. To explain, our theory caps the necessary ensemble size $B$ at an order of $O(n/k)$. Let $C(n)$ denote the computational cost of training a base learner with $n$ samples. Then, with $B=O(n/k)$, the total cost of our method is $B*C(k) = O(n\cdot C(k)/k)$, while the base learner applied to the full sample has the cost $C(n)$. Therefore, when $C(\cdot)$ grows superlinearly, our method is computationally cheaper (see Figure 4(b) for such an example with two-stage stochastic programming); When $C(\cdot)$ grows sublinearly, the base learner is cheaper; When the growth is linear, the two are comparable. We will add this discussion to the paper. Additionally, since our ensemble is constructed from independently drawn subsamples, the training can be easily parallelized to accelerate computation as pointed out in line 458.

---

> ### Author Response · Authors · 2024-11-24
>
> >Comment 5: It appears that k (sub-sampled data) has to be selected at a lower rate than n which implies that the ensemble size (B = O(n/k)) will be very large. How do existing generalization bounds for ensembles scale with the ensemble size and what are the consequences for them when B is not constant but rather a function of n?
>
> **Response**: Choice of $k$: If the base learner has a slow or polynomial decay rate, then $k$ indeed needs to be selected at a lower rate than $n$ for our method to achieve an exponential decay in the generalization tail. On the other hand, if the base learner’s generalization tail already exhibits exponential decay, then $k$ can be any number less than $n$ to retain the exponential tail. This is due to the presence of the tail $\mathcal{E}_{k,\delta}$ of the base learner in our bounds (9) and (10). We have explained this aspect in detail in Appendix B of our paper.
>
> Choice of $B$: The $O(n/k)$ is a theoretical cap for $B$ such that further increasing $B$ does not improve the performance. In practice, choosing $k$ to be a small constant multiple of $n$ (e.g., n/200) and $B$ a large constant (e.g., 200) already deliver satisfactory performance as detailed in lines 292-297. As for other ensemble methods, none of them is proven to achieve an exponential tail decay under our setting as far as we know.
>
> >Comment 6: In the discrete setting, what is the dependence of the generalization error on the size of the parameter class for the traditional ensembling methods?
>
> **Response**: Thank you for your question. To the best of our knowledge, there is currently no established theory on how traditional ensemble methods affect the tail behavior of the generalization error in the discrete setting.

---

> > ### Author Response · Authors · 2024-11-27
> > **Do our responses provide further clarify?**
> >
> > Thank you again for your insightful questions. We hope our further clarifications have addressed your concerns, and we are happy to incorporate the changes in the revised paper.  Besides, as suggested by **Reviewer aq9J**, we demonstrate the effectiveness of our method in a simple linear regression problem (please see our responses to **Reviewer aq9J**). We hope this example provides better illustration of our results, and we are glad to add the example to the revised paper.
> >
> > We are very open discuss any additional concerns you may have. We sincerely hope a kind reconsideration on the review score. Looking forward to your reply : )

---

### Meta-Review · Area_Chair_NDJm · 2024-12-18

**Metareview:**

The paper introduces MoVE (Majority Vote Ensembling) and ROVE (Retrieval and ϵ-Optimality Vote Ensembling), two ensemble learning techniques designed to achieve exponentially decaying tails for excess risk, improving generalization even when base learners exhibit slower polynomial decay rates. By selecting the best model trained on subsampled datasets and combining them using majority voting or likelihood-based selection, the methods enhance performance on out-of-sample tasks, particularly for heavy-tailed data distributions. The paper proves exponential improvement in tail bounds for excess risk, surpassing traditional variance reduction techniques. The authors validate the methods on synthetic and real-world datasets, demonstrating improved generalization over base learners. The paper proposes an innovative approach to ensemble learning with exponential tail improvements, with a rigorous theoretical foundation complemented by experiments.  However, the paper also has some weaknesses, as indicated by the reviewers. It uses outdated datasets, lacks comparisons with state-of-the-art methods, and explores only small model architectures.  Regarding the theoretical aspect, the results are abstract and hard to interpret without concrete examples.   The reviewers give mixed scores (ranging from reject to accept) highlighting the novelty of the theoretical contribution and the shortcomings in experimental validation and practical insights.  Recommendations include testing on more diverse datasets, improving clarity, and explaining computational costs in the paper. Summing up, while the paper introduces promising theoretical advancements in ensemble learning, I believe that it requires significant revisions in experiments, presentation, and practical applications to fully demonstrate its impact.

**Additional Comments On Reviewer Discussion:**

Some of the reviewers were inactive during the discussion period. However, some concerns of the reviewers were addressed. In any case, I still think that this paper needs more work to be accepted for publication.

---

### Decision · Program_Chairs · 2025-01-22

Reject